# Only Train Once: A One-Shot Neural Network Training And Pruning Framework

**Tianyi Chen**[*]
Microsoft
tiachen@microsoft.com

**Bo Ji**
National University of Singapore
jibo@comp.nus.edu.sg

**Tianyu Ding**
Johns Hopkins University
tding1@jhu.edu

**Biyi Fang**
Microsoft
bif@microsoft.com

**Guanyi Wang**
Georgia Institute of Technology
gwang93@gatech.edu

**Zhihui Zhu**
University of Denver
zhihui.zhu@du.edu

**Luming Liang**
Microsoft
lulian@microsoft.com

**Yixin Shi**
Microsoft
yixshi@microsoft.com

**Sheng Yi**
Microsoft
shengyi@microsoft.com

**Xiao Tu**
Microsoft
xiaotu@microsoft.com

## Abstract

Structured pruning is a commonly used technique in deploying deep neural networks (DNNs) onto resource-constrained devices. However, the existing pruning methods are usually heuristic, task-specified, and require an extra fine-tuning procedure. To overcome these limitations, we propose a framework that compresses DNNs into slimmer architectures with competitive performances and significant FLOPs reductions by Only-Train-Once (OTO). OTO contains two keys: $(i)$ we partition the parameters of DNNs into zero-invariant groups, enabling us to prune zero groups without affecting the output; and $(ii)$ to promote zero groups, we then formulate a structured-sparsity optimization problem and propose a novel optimization algorithm, Half-Space Stochastic Projected Gradient (HSPG), to solve it, which outperforms the standard proximal methods on group sparsity exploration and maintains comparable convergence. To demonstrate the effectiveness of OTO, we train and compress full models simultaneously from scratch without fine-tuning for inference speedup and parameter reduction, and achieve state-of-the-art results on VGG16 for CIFAR10, ResNet50 for CIFAR10 and Bert for SQuAD and competitive result on ResNet50 for ImageNet. The source code is available at https://github.com/tianyic/only_train_once.

## 1 Introduction

Deep neural networks (DNNs) have been shown to be effective in various real applications (51; 28). It is widely acknowledged that large-scale DNN models not only learn faster but also outperform their slimmer counterparts. However, such heavy models pose a great challenge to the deployment stage due to their resource-consuming nature. In addressing this issue, many model compression

---

[*]Corresponding author.

35th Conference on Neural Information Processing Systems (NeurIPS 2021).

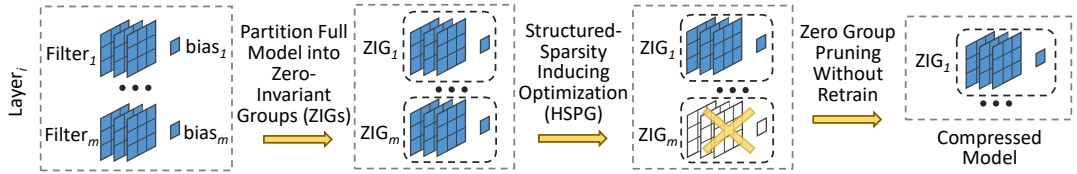

Figure 1: Overview of OTO. Without loss of generality, we illustrate OTO on a model with only vanilla convolutional layers, and for simplicity we only show Layer$_i$ with $m$ 3D filters and their biases. The key to its success is twofold: $(i)$ identify and partition the parameters of the model into zero-invariant groups (ZIGs); and $(ii)$ solve the structured-sparsity regularization problem using HSPG. Finally, we obtain the compressed model by directly pruning the zero groups, $i.e.$, ZIG$_m$.

techniques (5; 11) are proposed in the past decade that aim at compressing those large and complex models into slimmer and simpler ones while suffering negligible loss in performance.

Pruning methods as one of the main categories of model compression, focus on identifying and pruning redundant structures via various mechanisms to achieve a slimmer architecture, and thus improve the interpretability of a DNN model (26; 11; 65). For example, (32; 33) adopt fine-grained pruning via $\ell_1$ or $\ell_2$ regularization, which prune the small-weight connections based on some hard threshold. (36; 57; 60) measure the importance of filters to accelerate the networks by removing insignificant feature maps. (37; 7) utilize reinforcement learning agent to predict compression action.

Nevertheless, many of the existing pruning methods $(i)$ often rely on criteria based on heuristics or empirical cues, $e.g.$, magnitude of a connection weight and importance score of a filter, to identify redundant parameters, which may cause divergence during optimization; $(ii)$ thus require complex multi-stage pipelines that involve either a retraining or fine-tuning procedure to regain the accuracy during constructing a slimmer model, which is time-consuming; and $(iii)$ are specific to certain architectures or applications, and are consequently less applicable to various downstream scenarios. Recently, there have been a few efforts (14; 58; 8) to directly train the network with sparsity inducing regularizers, which provide generality and convergence guarantee. However, these approaches focus on either merely the individual sparsity of the parameters or the group sparsity of the filters, and thus cannot directly remove those zero components (still require subsequent fine-tuning) since the zeros are entangled with other commonly used components, $e.g.$, bias, batch normalization or skip connection. Furthermore, the optimization algorithms used in (14; 58) lack sufficient capability to explore (group) sparsity in DNNs effectively and require a post-processing step to yield exact zeros.

In this paper, we overcome the above limitations of existing pruning methods by proposing a *one-shot* neural network pruning framework, with which we are able to train a full heavy model from scratch only once, and obtain a slim architecture without fine-tuning while maintain high performance. As shown in Figure 1, the key to its success is twofold: $(i)$ we identify and partition the parameters of DNNs into zero-invariant groups (ZIGs), enabling us to prune redundant structures according to zero groups without affecting the output of the network; and $(ii)$ to promote zero groups, we formulate the pruning task as a structured-sparsity optimization problem and propose a novel optimization method, Half-Space Stochastic Projected Gradient (HSPG), to solve it, which outperforms the standard proximal methods on sparsity exploration and maintains comparable convergence. We highlight that both zero-invariant group partition and the novel optimization algorithm in promoting zero group lead to achieve one-shot neural network training and pruning regardless of its architecture.

Our main contributions are summarized as follows.

- **One-Shot Training and Pruning.** We propose OTO, a training and pruning framework that compresses a full neural network with competitive performance by Only-Train-Once, thereby one-shot. OTO dramatically simplifies the complex multi-stage training pipelines of the existing pruning approaches, fits various architectures and applications, and hence is generic and efficient.

- **Zero-Invariant Group.** We define zero-invariant groups for neural networks. If a network is partitioned into ZIGs, it allows us to prune the zero groups without affecting the output, which results in one-shot pruning. Such property is applicable to various popular structures from plain fully connected layers to sophisticated ones such as residual blocks and multi-head attention.

- **Novel Structured-Sparsity Optimization Algorithm.** We propose Half-Space Stochastic Projected Gradient (HSPG), a method that solves structured-sparsity inducing regularization problem. We show and analyze the superiority of HSPG in promoting zero groups of networks than the

standard proximal methods and the competitive objective convergence in practice. The fact that ZIG and HSPG are designed agnostic to networks makes OTO generic to various applications.

- **Experimental Results.** We train and compress full models simultaneously from scratch without fine-tuning for inference speedup and parameter reduction, and achieve state-of-the-art results on compression benchmark VGG for CIFAR10, ResNet50 for CIFAR10/ImageNet, Bert for SQuAD.

## 2   Related Work

Structured pruning focuses on identifying and pruning the redundant structures in a full model to achieve slimmer architectures for efficient model inference and storage (26; 32), where there have been numerous efforts dedicated. For CNN compression, the general procedure can be largely summarized as: (*i*) train a full model; (*ii*) identify and prune the redundant structures to build a slimmer model based on various criteria, including (structured) sparsity (58; 85; 14; 56; 102; 27; 102; 62; 91), Bayesian pruning (101; 65; 59; 81), ranking importance (54; 60; 41; 36; 57; 100), reinforcement learning (37; 7), adversarial robustness (76), scientific control (79), lottery ticket (23; 24; 72), joint quantization learning (80; 90), etc.; (*iii*) retrain or iteratively fine-tune the slimmer model to regain the accuracy regression during pruning. These methods cannot avoid the extra and usually time-consuming fine-tuning step because the identified redundant structures, even parametrized with zeros, actually contribute to the model output, thereby additional fine-tuning step is an absolute necessity.

For pruning Bert (82), knowledge distillation (40) and LayerDropout (21) shorten Bert by reducing the number of layers directly. Other methods (29; 75; 30) build slimmer Berts in the manner of individual sparsity, but require specially designed data structure for storage and computing library to take advantage of sparse data (31; 10), and typically cannot achieve inference speedup against the highly optimized library (16) for dense model due to the discontiguous memory allocation (9).

The structured sparsity for weight pruning is the most relevant to the algorithm described in this paper. The existing structure learning works (58; 85; 14; 56; 102) have the respective disadvantages: (*i*) multiple trainings during the whole procedure since their group partition cannot isolate the impact of pruned structures to the model output; and (*ii*) heuristic post-processing to generate zero groups as the standard proximal methods (19; 87; 88; 12) and ADMM (100; 58; 4) defective on the sparsity exploration for deep learning (8), which may deteriorate the performance of the model significantly.

Avoiding fine-tuning step during the whole pruning procedure is receiving more and more attentions because of its efficiency. In particular, SNIP (52) and GraSP (83) identify redundancy via salience scores at the initialization stage to construct pruned structures, then train the pruned models by the standard optimizers. SCP (48) isolates the impact of batch normalization, while lacks the consideration of more general DNN architectures.

## 3   OTO

In essence, OTO frames the network training and pruning as a structure learning problem. Given a full model $\mathcal{M}$, OTO trains and compresses it simultaneously from scratch *without* fine-tuning, and achieves significant reduction in both FLOPs and parameters. Particularly, as stated in Algorithm 1, the trainable parameters of $\mathcal{M}$ are firstly partitioned into a ZIG set $\mathcal{G}$ (Section 3.1). We then construct and solve a structured-sparsity inducing optimization problem (Section 3.2) by proposed stochastic optimizer (HSPG) to seek a highly group-sparse solution $x^*_{\text{HSPG}}$ (Section 3.3). Lastly, we obtain a compressed model $\mathcal{M}^*$ by directly pruning these zero groups (Section 3.4).

---

**Algorithm 1** Outline of OTO.

---

1: **Input:** Full model $\mathcal{M}$ (no need to be pretrained).
2: **Construct $\mathcal{G}$:** Partition the trainable parameters of $\mathcal{M}$ into a ZIG set $\mathcal{G}$.
3: **Train:** Train the model $\mathcal{M}$ using HSPG (Algorithm. 2) to obtain a group-sparse solution $x^*_{\text{HSPG}}$.
4: **Prune:** Construct a slimmer model architecture $\mathcal{M}^*$ by directly pruning zero groups of $x^*_{\text{HSPG}}$.
5: **Output:** Compressed model $\mathcal{M}^*$.

---

## 3.1 Zero-Invariant Group

The root cause of the existing methods having multi-stage training pipeline is that despite the pruned structure (*e.g.*, 3D filter) being zeros, its associated structures (*e.g.*, non-zero bias) still contribute to its corresponding output to the next layer (*e.g.*, feature map). As a result, the model accuracy regresses, hence an extra step of fine-tuning is necessary. OTO avoids the necessity by partitioning the parameters of DNNs into a set of so-called zero-invariant groups (ZIGs) $\mathcal{G}$ defined as follows.

**Definition 1** (Zero-Invariant Groups (ZIGs))**.** *For a layer-wise DNN, we partition its entire trainable parameters into disjoint groups $\mathcal{G} = \{g\}$. Then we call $\mathcal{G}$ zero-invariant groups (ZIGs) if each group $g \in \mathcal{G}$ is zero-invariant in the sense that all of the parameters in $g$ being zeros results in its corresponding output to the next layer to be zeros as well.*

In effect, if and only if a DNN model is partitioned into a ZIG set $\mathcal{G}$ and one or more of its element $g$ are parameterized by zeros, the entire corresponding structures contribute none to the model outputs and hence can be pruned directly. Such partition is applicable to various structures of DNN models. Without loss of generality, we define and describe ZIG partition for three most popular structures: *(i)* Conv-BN, *(ii)* Residual Block, and *(iii)* Fully Connected and Multi-Head Attention Layer.

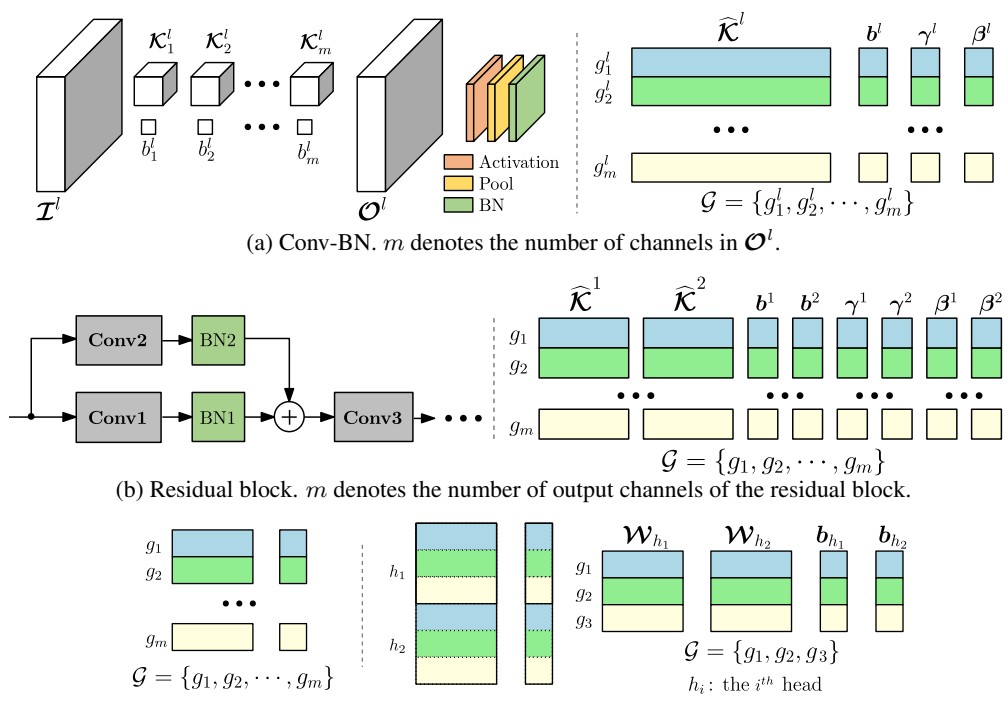

(a) Conv-BN. $m$ denotes the number of channels in $\boldsymbol{\mathcal{O}}^l$.

(b) Residual block. $m$ denotes the number of output channels of the residual block.

(c) Fully connected layer (Left). Multi-head attention layer (Right). $m$ denotes the length of output vector.

Figure 2: Zero-invariant group partition for three popular structures.

**ZIG of Conv-BN.** Convolutional layer (Conv) followed by batch-normalization layer (BN) is extensively used in DNN models. Figure 2a shows the ZIG partition for Conv-BN. The 4D filter tensor $\boldsymbol{\kappa}^l$ is flattened into a filter matrix $\tilde{\boldsymbol{\kappa}}^l$. During the forward pass, the input tensor $\boldsymbol{\mathcal{I}}^l$ is transformed into the output tensor $\boldsymbol{\mathcal{O}}^l$ of Conv and then into the input tensor of the $(l+1)^{th}$ layer $\boldsymbol{\mathcal{I}}^{l+1}$ by

$$\boldsymbol{\mathcal{O}}^l \leftarrow \boldsymbol{\mathcal{I}}^l \otimes \hat{\boldsymbol{\mathcal{K}}}^l + \boldsymbol{b}^l, \ \boldsymbol{\mathcal{I}}^{l+1} \leftarrow \frac{a(\boldsymbol{\mathcal{O}}^l) - \boldsymbol{\mu}^l}{\boldsymbol{\sigma}^l} \odot \boldsymbol{\gamma}^l + \boldsymbol{\beta}^l, \tag{1}$$

where denoted by $\otimes$ the convolutional operation, $\odot$ the element-wise multiplication and $a(\cdot)$ the activation function. BN is parameterized by mean $\boldsymbol{\mu}^l$, standard deviation $\boldsymbol{\sigma}^l$, weight $\boldsymbol{\gamma}^l$ and bias $\boldsymbol{\beta}^l$ respectively. The activation function needs to be zero-invariant, *i.e.*, $a(\mathbf{0}) = \mathbf{0}$, where most instances satisfy, *e.g.*, ReLU (25), PReLU (34), GELU (39) and LeakyReLU (89). Hence, each row of the flattened filter matrix $\tilde{\boldsymbol{\kappa}}^l$ and its bias $\boldsymbol{b}^l$ belong to one ZIG because they being zeros results in their corresponding channel of $\boldsymbol{\mathcal{O}}^l$ (*i.e.*, feature map) to be zeros as well. Subsequently, $\boldsymbol{\gamma}^l$ and $\boldsymbol{\beta}^l$ of this corresponding channel in BN are also included into this ZIG to avoid the value shift (zero to

non-zero) during normalization. Note that grouping these four sets of parameters channel-wisely makes Conv-BN zero-invariant regardless of the value of $\boldsymbol{\mu}^l$ and $\boldsymbol{\sigma}^l$, and hence they are excluded from the ZIG. For illustration, each ZIG is highlighted in the same color (*e.g.*, $g_1^l$ in blue).

**ZIG of Residual Block.** The residual block adds another layer of challenge because its output tensor is the summation of the outputs of two Conv-BNs. Figure 2b shows the ZIG partition for the residual block. As illustrated, before propagated to Conv3, the outputs of Conv1-BN1 and Conv2-BN2 are summarized and hence share the same dimension. As such, to make residual block zero-invariant, we group the four sets of parameters channel-wisely of both Conv1-BN1 and Conv2-BN2 into ZIGs, *i.e.*, each row of $\hat{\boldsymbol{\kappa}}^1, \boldsymbol{b}^1, \boldsymbol{\gamma}^1, \boldsymbol{\beta}^1$ of Conv1-BN1 and each row of $\hat{\boldsymbol{\kappa}}^2, \boldsymbol{b}^2, \boldsymbol{\gamma}^2, \boldsymbol{\beta}^2$ of Conv2-BN2. In Appendix A.1, we describe the zero-invariant group partition of ResNet50 in greater detail.

**ZIG of Fully Connected and Multi-Head Attention Layer.** Figure 2c shows the ZIG partition for fully connected and multi-head attention layer. Particularly, we partition each row of weight matrix and its associated bias into a ZIG, and therefore any input element is turned to zero if that ZIG is parameterized with zeros, making the fully connected layer zero-invariant. Multi-head attention layer is the key building block of the transformer architectures (82). Its trainable parameters contain a weight matrix and bias vector, consisting of the sub-matrix and sub-vector of each head (we use two heads as an example). We form ZIG by grouping each row of every sub-matrix and sub-vector, *i.e.*, each row of $\boldsymbol{\mathcal{W}}_{\boldsymbol{h_1}}, \boldsymbol{b}_{\boldsymbol{h_1}}, \boldsymbol{\mathcal{W}}_{\boldsymbol{h_2}}$ and $\boldsymbol{b}_{\boldsymbol{h_2}}$ of $h_1$ and $h_2$, respectively.

**Automatic ZIG Partition.** Based on the above illustrating examples, we provide prescribed ZIG partition for the tested DNNs in Section 4. Furthermore, given an arbitrary DNN architecture, the procedure of partitioning variables into ZIGs could be automatically proceeded, wherein the key would be identifying the connections among various layers, then performing corresponding group partition. We will leave the automatic ZIG partition for arbitrary DNNs as future work.

## 3.2 Structured-Sparsity Regularization

We now formulate a structured-sparsity regularization problem over the ZIG set $\mathcal{G}$ for the trainable parameters of the full model $\mathcal{M}$ as follows

$$\underset{\boldsymbol{x} \in \mathbb{R}^n}{\text{minimize}}\, \psi(\boldsymbol{x}) := f(\boldsymbol{x}) + \lambda r(\boldsymbol{x}),\; r(\boldsymbol{x}) := \sum_{g \in \mathcal{G}} \|[\boldsymbol{x}]_g\|, \tag{2}$$

where $\lambda > 0$ is a weighting coefficient, $f(\boldsymbol{x})$ is a task-specific loss function, and $r(\boldsymbol{x})$ is an augmented structured-sparsity inducing regularization term encoding the topological structure of $\mathcal{M}$ over $\mathcal{G}$. A larger $\lambda$ typically results in a higher group sparsity while sacrifices more on the bias of model estimation. We aim at computing a local optimum to achieve both low loss and high group sparsity.

To induce group sparsity onto the solution of (2), there exist several candidates for $r(\boldsymbol{x})$, including mixed $\ell_1/\ell_p$ norm ($p > 1$) (1; 20) and group Minmax Concave Penalty (MCP) (96). Among these candidates, the mixed $\ell_1/\ell_2$ norm as defined in (2) is arguably the most popular choice in classical machine learning applications (1; 92), where $\|\cdot\|$ is the $\ell_2$-norm, and each component $g \in \mathcal{G}$ indexes a group of variables. In this paper, we will demonstrate the effectiveness of OTO by selecting $r(\boldsymbol{x})$ as the mixed $\ell_1/\ell_2$ norm. We highlight OTO is applicable for other group sparsity regularizers as well.

## 3.3 Half-Space Stochastic Projected Gradient (HSPG)

To solve the non-smooth regularization problem as (2) in deep learning applications, the standard proximal method and the ADMM lack capability to effectively identify group sparsity; see the discussions later in this Section. Therefore, we propose a novel stochastic optimization algorithm so-called Half-Space Stochastic Projected Gradient (HSPG) to enhance the group sparsity exploration more effectively than the classical methods while maintain a similar convergence property.

**Outline.** We state the outline of HSPG in Algorithm 2. It contains two stages: Initialization Stage and Group-Sparsity Stage. The first Initialization Stage employs Stochastic Gradient Descent (SGD) step to search for a good but usually non-sparse solution estimate. Then the second stage proceeds Half-Space step started with the non-sparse iterate to effectively exploit the group sparsity within a sequence of reduced spaces and converges to the group-sparse solutions. Half-Space step performs SGD update on free non-zero variables along with a novel projection operator so-called Half-Space Projection, which significantly outperforms the standard proximal operators on sparsity exploration.

**Initialization Stage.** The Initialization Stage performs the vanilla SGD to find a good initial point for the subsequent Group-Sparsity Stage. At $k^{th}$ iteration, a stochastic gradient of $\tilde{f}$, *e.g.*, based on a mini-batch, is generated denoted as $\nabla\tilde{f}$. Since the group sparsity inducing regularizer $r(\boldsymbol{x})$ in the form as (2) is non-smooth, we select a subgradient $\zeta(\boldsymbol{x}_k)$ from its subdifferential $\partial r(\boldsymbol{x}_k)$ to form a stochastic subgradient of $\psi(\boldsymbol{x}_k)$ as $\nu(\boldsymbol{x}_k) := \nabla\tilde{f}(\boldsymbol{x}_k) + \lambda\zeta(\boldsymbol{x}_k)$. We then compute the next iterate as $\boldsymbol{x}_{k+1} := \boldsymbol{x}_k - \alpha_k\nu(\boldsymbol{x}_k)$ by subgradient descent update.

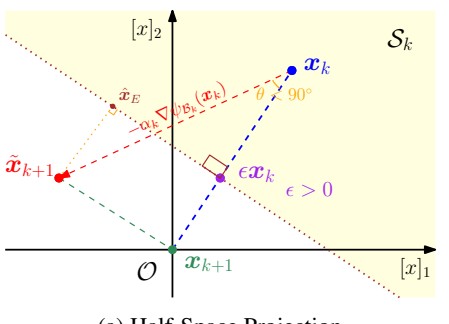
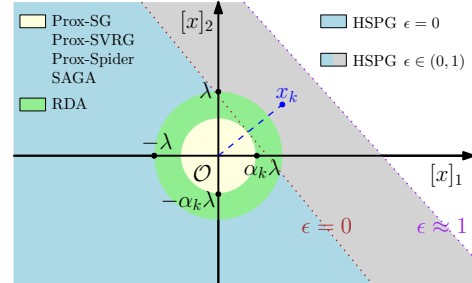

| (a) Half-Space Projection | (b) Projection Region For Mixed $\ell_1/\ell_2$ Regularization |

Figure 3: Illustration of Half-Space Step with projection in (6), where $\mathcal{G} = \{\{1, 2\}\}$.

**Group-Sparsity Stage.** The Group-Sparsity Stage is designed to effectively determine the groups of zero variables and capitalize convergence characteristic, which is in sharp contrast to other heuristic aggressive weight pruning methods that typically lack theoretical guarantees (55; 60). The intuition of Half-Space Step is to project $[\boldsymbol{x}_k]_g$ to zero only if $-[\boldsymbol{x}_k]_g$ serves as a descent step to $\psi(\boldsymbol{x}_k)$, *i.e.*, $-[\boldsymbol{x}_k]_g^\top[\nabla\psi(\boldsymbol{x}_k))]_g < 0$, hence updating $[\boldsymbol{x}_{k+1}]_g \leftarrow [\boldsymbol{x}_k]_g - [\boldsymbol{x}_k]_g = 0$ still results in some progress to the optimality. In particular, we first define the following index sets for any $\boldsymbol{x} \in \mathbb{R}^n$:

$$\mathcal{I}^0(\boldsymbol{x}) := \{g : g \in \mathcal{G}, [\boldsymbol{x}]_g = 0\} \text{ and } \mathcal{I}^{\neq 0}(\boldsymbol{x}) := \{g : g \in \mathcal{G}, [\boldsymbol{x}]_g \neq 0\}, \tag{3}$$

where $\mathcal{I}^0(\boldsymbol{x})$ represents the indices of groups of zero variables at $\boldsymbol{x}$, and $\mathcal{I}^{\neq 0}(\boldsymbol{x})$ indexes the groups of nonzero variables at $\boldsymbol{x}$. To proceed, we further define an artificial set that $\boldsymbol{x}$ lies in:

$$\mathcal{S}(\boldsymbol{x}) := \{\boldsymbol{0}\} \bigcup \{\boldsymbol{z} \in \mathbb{R}^n : [\boldsymbol{z}]_g = \boldsymbol{0} \text{ if } g \in \mathcal{I}^0(\boldsymbol{x}), \text{and } [\boldsymbol{z}]_g^\top[\boldsymbol{x}]_g \geq \epsilon \|[\boldsymbol{x}]_g\|^2 \text{ if } g \in \mathcal{I}^{\neq 0}(\boldsymbol{x})\}, \tag{4}$$

which consists of half-spaces and the origin. Here the parameter $\epsilon \geq 0$ controls how aggressively we promote group sparsity, and is typically fixed as zero in practice. Hence, $\boldsymbol{x} \in \mathcal{S}_k := \mathcal{S}(\boldsymbol{x}_k)$ only if: *(i)* $[\boldsymbol{x}]_g$ lies in the upper half-space for all $g \in \mathcal{I}^{\neq 0}(\boldsymbol{x}_k)$ for some prescribed $\epsilon \in [0, 1)$ as shown in Figure 3a; and *(ii)* $[\boldsymbol{x}]_g$ equals to zero for all $g \in \mathcal{I}^0(\boldsymbol{x}_k)$. Intuitively, $\mathcal{S}_k$ establishes the region where important structures inhabit, thereby redundant structures vanish if falling outside.

Ideally, the Initialization Stage has produced reasonably well but typically non-sparse iterate $\boldsymbol{x}_k$ nearby a group-sparse solution $\boldsymbol{x}^*$ of problem (2), , *i.e.*, the optimal distance $\|\boldsymbol{x}_k - \boldsymbol{x}^*\|$ is sufficiently small. As seen in Appendix B, it further indicates that the group-sparse optimal solution $\boldsymbol{x}^*$ inhabits $\mathcal{S}_k$, and $\mathcal{S}_k$ has already covered the group-support of $\boldsymbol{x}^*$, *i.e.*, $\mathcal{I}^{\neq 0}(\boldsymbol{x}^*) \subseteq \mathcal{I}^{\neq 0}(\boldsymbol{x}_k)$. Our goal now becomes minimizing $\psi(\boldsymbol{x})$ over $\mathcal{S}_k$ to identify the remaining zero groups, *i.e.*, $\mathcal{I}^0(\boldsymbol{x}^*)/\mathcal{I}^0(\boldsymbol{x}_k)$, which is formulated as the following problem:

$$\underset{\boldsymbol{x}\in\mathcal{S}_k}{\text{minimize}} \, \psi(\boldsymbol{x}) = f(\boldsymbol{x}) + \lambda r(\boldsymbol{x}). \tag{5}$$

The next iterate $\boldsymbol{x}_{k+1}$ is computed as an solution estimate of problem (5).

---

**Algorithm 2** Outline of HSPG for solving (2).

1: **Input:** $\boldsymbol{x}_0 \in \mathbb{R}^n$, $\alpha_0 > 0$, $\epsilon \in [0, 1)$, and $N \in \mathbb{Z}^+$.
2: **Output:** a group-sparse solution $\boldsymbol{x}^*_{\text{HSPG}}$ from $\{\boldsymbol{x}_k\}$.
3: **for** $k = 0, 1, 2, \dots$ **do**
4:      Compute a stochastic subgradient $\nu(\boldsymbol{x}_k)$ of $\psi(\boldsymbol{x}_k)$.
5:      **if** $k < N$ **then**
6:          ***Subgradient Descent Update:***
7:          Set $\boldsymbol{x}_{k+1} \leftarrow \boldsymbol{x}_k - \alpha_k\nu(\boldsymbol{x}_k)$.
8:      **else**
9:          ***Half-Space Update:***
10:         Set a trial iterate $\tilde{\boldsymbol{x}}_{k+1}$ as

$$[\tilde{\boldsymbol{x}}_{k+1}]_{\mathcal{I}^{\neq 0}(\boldsymbol{x}_k)} \leftarrow [\boldsymbol{x}_k - \alpha_k\nu(\boldsymbol{x}_k)]_{\mathcal{I}^{\neq 0}(\boldsymbol{x}_k)}$$
$$[\tilde{\boldsymbol{x}}_{k+1}]_{\mathcal{I}^0(\boldsymbol{x}_k)} \leftarrow \boldsymbol{0}.$$

11:         **for** each group $g$ in $\mathcal{G}$ **do**
12:            $[\boldsymbol{x}_{k+1}]_g \leftarrow [\text{Proj}_{\mathcal{S}_k}^{HS}(\tilde{\boldsymbol{x}}_{k+1})]_g$.
13:      Update $\alpha_{k+1}$.

---

Particularly, in Algorithm 2, $[\boldsymbol{x}_{k+1}]_{\mathcal{I}^0(\boldsymbol{x}_k)} \equiv \boldsymbol{0}$ will not be updated, and only the entries in $\mathcal{I}^{\neq 0}(\boldsymbol{x}_k)$ are free to move. Hence $\psi(\boldsymbol{x})$ is smooth on $\mathcal{S}_k$, and (5) is a reduced space optimization problem. A standard way to solve problem (5) would be the stochastic gradient descent equipped with Euclidean projection (68). However, such a projected method rarely produces zero (group) variables, as the dense Euclidean projected point $\hat{\boldsymbol{x}}_E \neq \boldsymbol{0}$ illustrated in Figure 3a. To address, we introduce a novel half-space projection operator to effectively project an entire group of variables to zeros.

As line 4 and 9-12 in Algorithm 2, we first approximate the (sub)gradient of $\psi$ on the free variables by $[\nu(\boldsymbol{x}_k)]_{\mathcal{I}^{\neq 0}(\boldsymbol{x}_k)}$, then employ gradient descent over $\mathcal{I}^{\neq 0}(\boldsymbol{x}_k)$ to compute a trial point $\widetilde{\boldsymbol{x}}_{k+1}$ which is passed into a fresh half-space projection operator $\text{Proj}^{HS}_{\mathcal{S}_k}(\cdot)$ defined as

$$\left[\text{Proj}^{HS}_{\mathcal{S}_k}(\boldsymbol{z})\right]_g := \begin{cases} 0 & \text{if } [\boldsymbol{z}]_g^\top [\boldsymbol{x}_k]_g < \epsilon \left\| [\boldsymbol{x}_k]_g \right\|^2, \\ [\boldsymbol{z}]_g & \text{otherwise.} \end{cases} \tag{6}$$

The above projector of form (6) is not the standard one in Euclidean sense[2], and it has two advantages: *(i)* the actual search direction $\boldsymbol{d}_k := (\text{Proj}^{HS}_{\mathcal{S}_k}(\tilde{\boldsymbol{x}}_{k+1}) - \boldsymbol{x}_k)/\alpha_k$ performs as a descent direction to $\psi(\boldsymbol{x}_k)$, *i.e.*, $[\boldsymbol{d}_k]_g^\top [\nu(\boldsymbol{x}_k)]]_g < 0$ as $\theta < 90°$ in Figure 3a, hence the progress to the optimum is made via the sufficient decrease property drawn as Lemma 1 in Appendix B; then *(ii)* it effectively projects entire groups of variables to zero if the inner product of corresponding entries is sufficiently small. In contrast, the Euclidean projection operator is far away effective to promote group sparsity.

**Superiority of HSPG on Group Sparsity Identification.** We now intuitively illustrate the strength of HSPG on group sparsity exploration. In fact, the half-space projection (6) is a more effective sparsity promotion mechanism compared to the standard proximal methods. Particularly, it benefits from a much larger projection region to map a reference point $\hat{\boldsymbol{x}}_{k+1} := \boldsymbol{x}_k - \alpha_k \nabla \tilde{f}(\boldsymbol{x}_k)$ or its variants to zero. As the 2D case described in Figure 3b, the projection regions of the state-of-the-art Prox-SG (19), Prox-SVRG (88), Prox-Spider (97) and SAGA (12) for (2) are $\ell_2$-balls with radius as $\alpha_k \lambda$. In deep learning applications, the step size $\alpha_k$ is usually selected around $10^{-3}$ to $10^{-4}$ or even smaller for convergence. Together with the common setting of $\lambda \ll 1$, their projection regions would vanish rapidly, resulting in the difficulties to produce group sparsity. As a sharp contrast, even though $\alpha_k \lambda$ is near zero, the projection region of HSPG $\{\boldsymbol{x} : \boldsymbol{x}_k^\top \boldsymbol{x} < (\alpha_k \lambda + \epsilon \left\| \boldsymbol{x}_k \right\|) \left\| \boldsymbol{x}_k \right\| \}$ (seen in Appendix B) is still an open half-space which contains those $\ell_2$ balls as well as RDA (87)'s if $\epsilon$ is large enough. Conversely, vanilla ADMM alone lacks the mechanism to project a group of variables to zero, unless equips with extra post-processing step (100; 58). In Appendix B, we further reveal that HSPG still maintains the convergence to the optimality as drawn in Theorem 1. Moreover, we numerically demonstrate the superiority of HSPG in the sense of optimization in Appendix C.

### 3.4 Pruning Without Fine-Tuning

The group-sparse solution $\boldsymbol{x}^*_{\text{HSPG}}$ over ZIGs to the full model $\mathcal{M}$ is leveraged to construct the slimmer model $\mathcal{M}^*$. Particularly, we prune the redundant structures identified as zero groups $\mathcal{I}^0$ and retain non-zero groups $\mathcal{I}^{\neq 0}$ in $\boldsymbol{x}^*_{\text{HSPG}}$. Because the parameters of full model are partitioned into ZIGs, the pruned structures contribute none to the model output. Therefore, given the same input, the slimmer model $\mathcal{M}^*$ computes the identical output as the full model $\mathcal{M}$ parameterized with $\boldsymbol{x}^*_{\text{HSPG}}$.

## 4 Experiment

In this section, we numerically demonstrate the effectiveness of OTO by one-shot training and pruning without fine-tuning on several benchmark compression tasks for CNNs, *i.e.*, VGG16 (77) for CIFAR10 (49) and ResNet50 (35) for CIFAR10 (49) and ImagetNet (ILSVRC2012) (15). We also verify the scalibility of OTO onto Bert (82) evaluated on SQuAD (69). All datasets are free to academic usage and do not contain personally identifiable information or offensive content. CIFAR10 is under the MIT license, consisting of 50,000 training and 10,000 test images from 10 classes. ImagetNet is a large-scale dataset without license and contains about 1.2 million and 50,000 images in training and validation sets from 1,000 classes. SQuAD is under the CC BY-SA 4.0 license with about 100,000 question/answer pairs splitted into train/dev/test sets as (80/10/10%). We conduct all experiments on a Nvidia RTX8000 GPU and provide implementation details in Appendix A.

---

[2]Note that when $r(\boldsymbol{x}) = \left\| \boldsymbol{x} \right\|_1$ where each $g \in \mathcal{G}$ is singleton, then $\mathcal{S}_k$ becomes an orthant face (8).

Table 1: VGG16 and VGG16-BN for CIFAR10. Convolutional layers are in bold.

| Method | BN | Architecture | FLOPs | # of Params | Top-1 Acc. |
|---|---|---|---|---|---|
| Baseline | ✗ | **64-64-128-128-256-256-256-512-512-512-512-512-512**-512-512 | 100% | 100% | 91.6% |
| SBP (65) | ✗ | **47-50-91-115-227-160-50-72-51-12-34-39-20**-20-272 | 31.1% | 5.9% | **91.0%** |
| BC (59) | ✗ | **51-62-125-128-228-129-38-13-9-6-5-6-6**-6-20 | 38.5% | 5.4% | **91.0%** |
| RBC (101) | ✗ | **43-62-120-120-182-113-40-12-20-11-6-9-10**-10-22 | 32.3% | 3.9% | 90.5% |
| RBP (101) | ✗ | **50-63-123-108-104-57-23-14-9-8-6-7-11**-11-12 | 28.6% | 2.6% | **91.0%** |
| **OTO** | ✗ | **21-45-82-110-109-68-37-13-9-7-3-5-8**-170-344 | **16.3%** | **2.5%** | **91.0%** |
| Baseline | ✓ | **64-64-128-128-256-256-256-512-512-512-512-512-512**-512-512 | 100% | 100% | 93.2% |
| EC (55) | ✓ | **32-64-128-128-256-256-256-256-256-256-256-256-256**-512-512 | 65.8% | 37.0% | 93.1% |
| Hinge (56) | ✓ | – | 60.9% | 20.0% | 93.6% |
| SCP (48) | ✓ | – | 33.8% | 7.0% | **93.8%** |
| **OTO** | ✓ | **22-56-93-123-182-125-95-45-27-21-10-13-19**-244-392 | **26.8%** | **5.5%** | 93.3% |

## 4.1  Deep Convolutional Neural Network

The results on CNN experiments are summarized in Table 1, 2 and 4. In particular, we compare OTO to its state-of-the-art counterparts by Top-1/5 accuracy, remaining FLOPs and parameters against the corresponding baseline (full model). We report the numbers of other methods based on the corresponding literature and leave as '-' if not reported. The best pruning results are marked as bold.

**VGG16 for CIFAR10.** We consider the standard VGG16 and the version with batch normalization layer after each convolutional layer, referred to as VGG16-BN. OTO partitions the parameters into ZIGs following Section 3.1, then trains and prunes the model via HSPG, and finally constructs the slimmer model without fine-tuning. For VGG16, as shown in Table 1, the pruned architecture of OTO indicates that OTO identifies similar redundancy of the intermediate and late convolutional layers compared to other methods, but significantly more of the early convolutional layers. As a result, OTO achieves $83.7\%$ $(1 - 16.3\%)$ FLOPs reduction and $97.5\%$ $(1 - 2.5\%)$ parameter reduction with the best Top-1 accuracy, which outperforms other state-of-the-arts significantly. For VGG16-BN, among all, OTO reduces FLOPs and parameters to the lowest $26.8\%$ and $5.5\%$, respectively. EC (55) and Hinge (56) achieve the same level of Top-1 accuracy as OTO, but are substantially outperformed when it comes to FLOPs and parameter reduction. We further present the FLOPs reductions per layer of OTO in Table 7 of Appendix A.4.

**ResNet50 for CIFAR10.** Since OTO is able to automatically learn a slimmer model of high performance, we compare it with two state-of-the-art automatic neural network compression frameworks, *i.e.*, AMC (37) and ANNC (90). AMC trains a reinforcement learning agent to predict compression action for each layer environment. ANNC jointly proceeds pruning and quantization within energy

Table 2: ResNet50 for CIFAR10.

| Method | FLOPs | # of Params | Top-1 Acc. |
|---|---|---|---|
| Baseline | 100% | 100% | 93.5% |
| AMC (37) | – | 60.0% | 93.6% |
| ANNC (90) | – | 50.0% | **95.0%** |
| PruneTrain (61) | 30.0% | – | 93.1% |
| N2NSkip (78) | – | 10.0% | 94.4% |
| **OTO** | **12.8%** | **8.8%** | 94.4% |

constraint. We conduct OTO on their shared experiment, *i.e.*, ResNet50 on CIFAR10. ResNet50 includes both the standard convolutional layers and the layers with residual connections, which are partitioned into ZIGs following Section 3.1. We report the results in Table 2 along with other competitors from (61; 78). Based on the results, all methods achieve competitive validation accuracies, where most of them are even higher than the baseline reported in (37). OTO outperforms AMC, ANNC without quantization, PruneTrain and N2NSkip by using only $12.8\%$ FLOPs and $8.8\%$ parameters. Note that no FLOPs reduction is reported in (37) and (90). Finally, we highlight that OTO is flexible to incorporate quantization as the two techniques are complementary and will leave to future work.

**Ablation Study on Switching Parameter $N$.** We provide ablation study regarding the impact the switch (parameterized as $N$) between the initialization stage and the group-sparsity stage in Algorithm 1. In theory, as shown in Theorem 1 of Appendix B.4, the projection stage should start when the iterate falls nearby a group sparse local

Table 3: OTO Under Different Switchings ($N = T, 2T, 3T$) for VGG16, VGG16-BN and ResNet50 on CIFAR10

| Backend | FLOPs | # of Params | Top-1 Acc. |
|---|---|---|---|
| VGG16 | $17.0\% \pm 1.4\%$ | $2.6\% \pm 0.4\%$ | $90.9\% \pm 0.3\%$ |
| VGG16-BN | $25.4\% \pm 1.1\%$ | $5.0\% \pm 0.5\%$ | $93.3\% \pm 0.2\%$ |
| ResNet50 | $12.9\% \pm 1.5\%$ | $8.5\% \pm 1.0\%$ | $94.2\% \pm 0.2\%$ |

minimizer. In practice, we relax it to start the group sparsity stage once the iterate falling into some stationary status regarding the validation accuracy. As described in Appendix A.2, throughout all experiments, we periodically decay the learning rate per fixed number of epochs parameterized as $T$.

At the end of each $T$ epochs, we then proceed a statistical test similar to (98) but on the validation accuracy and find that the validation accuracy falls into stationarity near the late epochs of each period. Therefore, in our pruning experiments, we switch to the group-sparsity stage right after the first $T$ epochs. Table 3 describes the performance of OTO under varying switching parameters, from which we observe that OTO is not largely sensitive to the switching parameter if the group-sparsity stage starts after some stationary condition has been numerically satisfied.

**ResNet50 for ImageNet.** We now evaluate OTO on ResNet50 for ImageNet. As shown in Table 4, OTO prunes $64.5\%(1 - 35.5\%)$ parameters to achieve $65.5\%(1-34.5\%)$ FLOPs reduction with only $1.4\%/0.8\%$ Top-1/5 accuracy regression compared to the baseline. OTO consistently outperforms the majority of counterparts especially on the FLOPs reduction and the parameter reduction. We note that Hinge (56) prunes CNNs via structured-sparsity optimization by employing standard stochastic proximal gradient method. It

Table 4: ResNet50 for ImageNet.

| Method | FLOPs | # of Params | Top-1 Acc. | Top-5 Acc. |
|---|---|---|---|---|
| Baseline | 100% | 100% | 76.1% | 92.9% |
| DDS-26 (43) | 57.0% | 61.2% | 71.8% | 91.9% |
| CP (38) | 66.7% | – | 72.3% | 90.8% |
| ThiNet-50 (45) | 44.2% | 48.3% | 71.0% | 90.0% |
| RBP (101) | 43.5% | 48.0% | 71.1% | 90.0% |
| RRBP (101) | 45.4% | – | 73.0% | 91.0% |
| SFP (36) | 41.8% | – | 74.6% | 92.1% |
| Hinge (56) | 46.6% | – | 74.7% | – |
| GBN-50 (94) | 44.9% | 46.6% | 75.2% | 92.4% |
| GBN-60 (94) | 59.5% | 68.2% | 76.2% | 92.8% |
| Group-HS (2e-5) (91) | 32.4% | - | 75.2% | 92.5% |
| Group-HS (1e-5) (91) | 52.9% | - | **76.4%** | **93.1%** |
| ResRep (18) | 45.5% | - | 76.2% | 92.9% |
| SCP (48) | 45.7% | - | 74.2% | 92.0% |
| **OTO** | **34.5%** | **35.5%** | 74.7% | 92.1% |
| **OTO**[*] | **34.5%** | **35.5%** | 75.1% | 92.5% |

requires several trainings including fine-tuning the pruned model, because it partitions the parameters into non-ZIGs and relies on an empirical truncation mechanism to generate zero groups due to the weakness of proximal operator in deep learning applications (8). In contrast, OTO only trains and prunes the full model from scratch once and obtains better pruning results. The comparison between OTO and Hinge stand as evidence of the superiority of OTO due to ZIGs and HSPG. Furthermore, if with more training efforts, OTO reaches higher Top-1/5 accuracy marked as [*] in Table 4 and becomes more competitive to stronger competitors, such as GBN (94), Group-HS (91) and ResRep (48).

**Representation of Deep Features of ImageNet.** It is widely acknowledged that deep neural architectures could be treated as non-linear feature representation extractors. Therefore, we further study the feature representation extracted by OTO to demonstrate its generalizability to other visual applications besides image classification. Figure 4 shows the clustering results of ImageNet validation images using the deep feature extracted by both the baseline ResNet50 and the pruned ResNet50 by OTO. Specifically, we extract the deep features over the validation samples in ImageNet, *i.e.*, the tensors fed into the fully connected layer, and project them onto a 2-dimensional space via PCA (47). For illustration, following the hierarchy of ImageNet (3), two sets of five classes are randomly selected[3]. We observe that the deep features of the pruned ResNet50 by OTO remain structured in the sense that distinct classes are well separated from each other. Over all 1000-class ImageNet validation images, OTO achieves 48.2% clustering accuracy compared to 42.5% of the baseline ResNet50 using k-means. Both observations indicate that with only 35.5% parameters and 34.5% FLOPs, the pruned ResNet50 is still able to extract highly discriminative deep features. We argue that during model compression, OTO not only achieves parameter and FLOPs reduction, but also preserves the ability of capturing perceptual properties (99). This is especially important in training and compressing models for many vision tasks, *e.g.*, object detection (70; 71), frame interpolation (2; 17; 67) and video synthesis (84; 50). We leave the application of OTO to broader tasks to future work.

### 4.2 Large-Scale Transformer

We show the scalability of OTO by pruning the large-scale transformer Bert (82), evaluated on SQuAD, a question-answering benchmark (69). Bert mainly includes embedding layers, fully connected layers and multi-head attention layers. The fully connected layers and the multi-head attention layers are partitioned into ZIGs following Section 3.1. For fair comparisons, we follow the prior Bert compression works (14; 75) and do not prune the embedding layers.

---

[3]Each selected class belongs to a disjoint upper category.

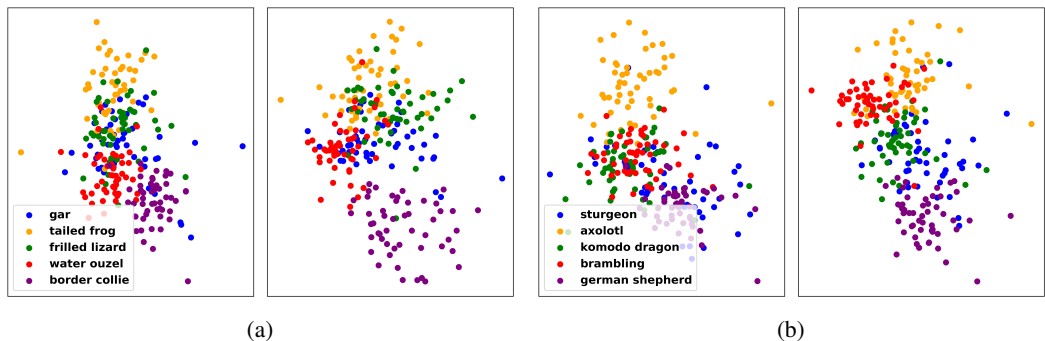

(a)                      (b)

Figure 4: Clustering results of ImageNet validation images using deep features extracted by full ResNet50 (left of a and b) and pruned ResetNet50 by OTO (right of a and b). The points are visualized by projecting deep features onto a two-dimensional space via PCA.

To the best of our knowledge, OTO is the first work that compresses Bert by exploring group sparsity on individual layers and achieves significant parameter reduction and inference speedup[4]. In contrast, the existing works (29; 75; 30) prune individual parameters instead, *i.e.*, the generated sparsity is not structured. Hence, the computed models typically do not have inference speedup (75), unless are executed by specialized hardware and sparse computing library (31; 10). As

Table 5: Pruning Bert on SQuAD

| Method | # of Params | Exact | F1-score | SpeedUp |
|---|---|---|---|---|
| Baseline | 100% | 81.0% | 88.3% | 1× |
| MaP (75) | **10.0%** | 67.7% | 78.5% | 1×* |
| MvP (75) | **10.0%** | 71.9% | 81.7% | 1×* |
| ProxSSI (14) | 83.4%[†] | 72.3% | 82.0% | 1× |
| **OTO** | 91.0% | **75.0%** | **84.1%** | 1.1× |
| **OTO** | 76.2% | 72.3% | 82.1% | 1.2× |
| **OTO** | 66.7% | 71.9% | 82.0% | 1.3× |
| **OTO** | 53.3% | 71.4% | 81.5% | 1.5× |
| **OTO** | 40.0% | 70.9% | 81.1% | **1.8×** |

\* Based on the statement in the official git repository of (75).
[†] Approximate value based on the group sparsity reported in (14).

shown in Table 5, under different group sparsity upper bound constraints, OTO reduces 9% to 60% parameters and achieves up to $1.8\times$ inference speedup based on the average model execution time [5]. In comparison, despite that the pruned model contains $10\%$ parameters, MaP and MvP (75) do not have any inference speedup. On the other hand, the structured sparsity on Bert is studied in (14) (referred to as ProxSSI), where an adaptive proximal method is proposed to yield group-sparse solution. Nonetheless, ProxSSI optimizes over non-ZIGs and relies on proximal operator to identify group sparsity. Therefore, the groups even parameterized with zeros have to be retained in the model rather than pruned. As a consequence, ProxSSI is not competitive to OTO on parameter reduction, and there is no reported inference speedup. Note that all the pruning methods achieve comparable exact match rate and F1-score.

## 5   Conclusion And Future Work

We propose OTO, a one-shot deep neural networks (DNNs) training and pruning framework, that compresses full DNNs into slimmer architectures with competitive performances and significant FLOPs and parameter reduction without fine-tuning. OTO contains two fundamentals: *(i)* partitions the trainable parameters of DNNs into zero-invariant groups (ZIGs), thereby pruning zero groups does not affect the model output, and *(ii)* trains by a novel optimizer, Half-Space Stochastic Projected Gradient (HSPG), which outperforms proximal methods on group sparsity exploration and maintains comparable convergence. We numerically demonstrate OTO on benchmark experiments, *i.e.*, VGG16 for CIFAR10, ResNet50 for CIFAR10/ImageNet and Bert for SQuAD, and achieve state-of-the-art pruning results. We leave automatically generating ZIGs for arbitrary DNNs, incorporating quantization and applying OTO to other tasks to future work.

---

[4]Knowledge distillation (40) and LayerDropout (21) compresses Bert by pruning layers in their entirety.
[5]Run by OnnxRuntime (16)

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
