# A Implementation Details of OTO

## A.1 ZIG for ResNet50

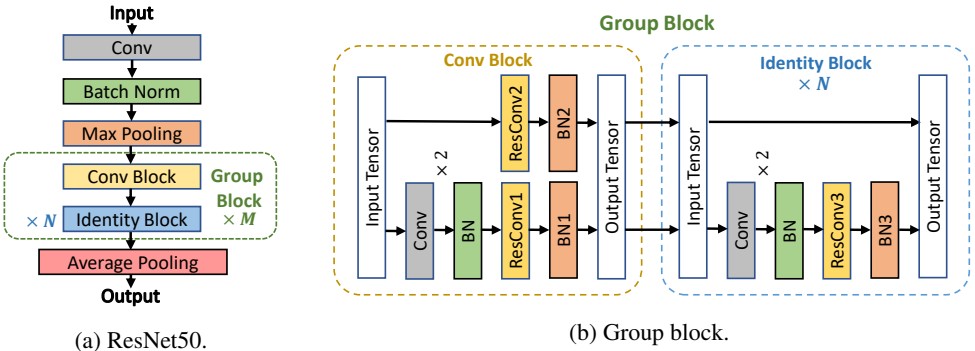

(a) ResNet50.

(b) Group block.

Figure 5: ResNet50 Architecture.

Without loss of generality, we illustrate ZIGs for the general ResNet class with ResNet50. As shown in Figure 5a, ResNet50 begins with a Conv-BN, a pooling layer, and extracts features by $M$ Group Blocks that each contains one Conv Block and $N$ Identity Block. The extracted features are ultimately fed into an average pooling layer for different downstream computations. There exist two types of convolution structures inside ResNet50: *(i)* the regular Conv-BN (see Section 3.1), marked as gray and green blocks in Figure 5b, and *(ii)* the ResConv-BN of which the output shares the same dimension with another ResConv-BN, marked as yellow and brown in Figure 5b.

For ResNet50, we partition regular Conv-BN following Section 3.1. For ResConv-BN, within each Group Block, the intermediate input/output tensors in Conv/Indentity Blocks share the same dimension, and hence all the ResConv-BNs in one Group block share the same number of 3D filters. Consequently, their flattened filter matrices has the same number of rows. Figure 5b breaks down the architecture of a Group Block. The output tensors of ResConv-BN1 and ResConv-BN2 in Conv Block, denoted as $\mathcal{O}^1$ and $\mathcal{O}^2$, are computed by (7) and (8) respectively. They are then summed up as the input tensor of the subsequent identify block $\mathcal{I}^{I_1}$, indicating that $\mathcal{O}^1$ and $\mathcal{O}^2$ have the same shape and their flattened filter matrices $\hat{\mathcal{K}}^1$ and $\hat{\mathcal{K}}^2$ has the same number of rows. As (11), $\mathcal{I}^{I_1}$ later sums the output tensor of ResConv-BN3 $\mathcal{O}^3$ to yield the input tensor to the next Identity Block $\mathcal{I}^{I_2}$, implying the filter matrix of ResConv-BN3 $\hat{\mathcal{K}}^3$ has the same number of rows as $\hat{\mathcal{K}}^1$ and $\hat{\mathcal{K}}^2$.

$$\mathcal{O}^1 \leftarrow \frac{a(\mathcal{I}^1 \otimes \hat{\mathcal{K}}^1 + b^1) - \mu^1}{\sigma^1} \odot \gamma^1 + \beta^1 \tag{7}$$

$$\mathcal{O}^2 \leftarrow \frac{a(\mathcal{I}^2 \otimes \hat{\mathcal{K}}^2 + b^2) - \mu^2}{\sigma^2} \odot \gamma^2 + \beta^2 \tag{8}$$

$$\mathcal{I}^{I_1} \leftarrow \mathcal{O}^1 + \mathcal{O}^2 \tag{9}$$

$$\mathcal{O}^3 \leftarrow \frac{a(\mathcal{I}^3 \otimes \hat{\mathcal{K}}^3 + b^3) - \mu^3}{\sigma^3} \odot \gamma^3 + \beta^3 \tag{10}$$

$$\mathcal{I}^{I_2} \leftarrow \mathcal{I}^{I_1} + \mathcal{O}^3 \tag{11}$$

Therefore, based on (7) to (11), to make the entire Group Block zero-invariant, we group each $i^{th}$ row of the filter matrix for all the Res-Conv-BNs of same group block. In doing so, any one row of parameters being zeros results in the output, *i.e.*, the corresponding channel of feature map, being zeros. Figure 6 shows ZIG for the three ResConv-BN of a Group Block. Regardless of the input, the $i^{th}$ channel-wise matrix of $\mathcal{I}^{I_1}$ are zeros if and only if both $i^{th}$ channel-wise matrices of $\mathcal{O}^1$ and $\mathcal{O}^2$ are equal to zero. This is equivalent to both $i^{th}$ rows of $\hat{\mathcal{K}}^1$ and $\hat{\mathcal{K}}^2$ being zeros. Similarly, $i^{th}$ channel-wise matrix of $\mathcal{I}^{I_2}$ being zeros regardless of the input further requires the $i^{th}$ row of $\hat{\mathcal{K}}^3$ to be grouped in the ZIG.

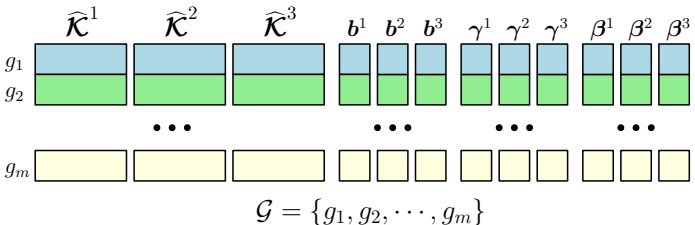

Figure 6: Zero Invariant Groups for the three ResConv-BN of a Group Block.

## A.2  Training Details

We implement OTO in PyTorch. The key ingredient HSPG is packaged as an optimizer class which is flexible to various applications. During the experiment, the trainable parameters of the full model $\mathcal{M}$ are firstly partitioned into a ZIG set $\mathcal{G}$ wherein each group is tagged as its corresponding atomic layer category, *e.g.*, fully-connected layer or convolutional layer. The ZIG set $\mathcal{G}$ is treated as an argument to the HSPG constructor. Then $\mathcal{M}$ is trained from scratch by HSPG where the group-wise variables are updated based on their tagged layer category. In our repository, we provide the prescribed ZIG partitions for the DNNs used in this paper, *i.e.*, VGG16, VGG16-BN, ResNet50 and Bert. For other models, one can easily follow Section 3.1 and Appendix A.1 to construct a ZIG partition and feed it as an argument to the HSPG optimizer. After training, a full group-sparse model with high performance is achieved. Finally, a slimmer pruned model $\mathcal{M}^*$ is constructed following Section 3.4 without fine-tuning and has the identical performance as the full group-sparse model. We provide the implementation in https://github.com/tianyic/only_train_once and the pruned models associated with the corresponding full group sparse models in https://tinyurl.com/otocheckpoints.

**Parameter Settings.**  We conduct all experiments on an Nvidia RTX8000 graphics card with 48 GB memory. For all CNN experiments, the step size (learning rate) $\alpha_k$ is initialized as $10^{-1}$, and decayed by a factor 0.1 periodically $T$ epochs till the minimum value $10^{-4}$. The selection of $T$ depends on the max number of epochs $K$. We follow various benchmark online resources to select $K$. Particularly, for all CIFAR10 experiments, we follow the model pre-training settings in (56) and set $K$ as 300. Note that by using the same number of epochs, OTO achieves both slimmer model and competitive performance simultaneously. For the ImageNet experiment, following (35), we set $T$ as 30 and $K$ as 120. For all Bert experiments, the step size $\alpha_k$ is initialized as $10^{-1}$ and decayed by a factor 0.1 after 4 epochs to be as $10^{-2}$.

We set the mini-batch size as the commonly used 64 for CIFAR10, 256 for ImageNet and 32 for SQuAD experiments. For all experiments, we initialize the regularization coefficient $\lambda$ as $10^{-3}$ to balance between model performance and group sparsity. In particular, $\lambda$ as $10^{-3}$ is the maximum value from the candidate set $\{10^{-2}, 10^{-3}, 10^{-4}\}$ which returns competitive evaluation results to the majority of the tested models trained without regularization. In addition, to favor more on the model performance, if group sparsity becomes stationary, we decay $\lambda$ by a factor 0.1 periodically after stepping into Group-Sparsity Stage. The control parameter $\epsilon \in [0, 1)$ in the half-space projection (6) controls the aggressiveness level of group sparsity promotion, which is typically fixed as 0 since for most of experiments, $\epsilon \equiv 0$ has resulted in sufficiently good experiment results. In case if group sparsity is not sufficiently yielded, we provide an adaptive mechanism to increase $\epsilon$ by 0.1 till the upper bound 0.999. For the setting of $N$ which controls when switching to the Group-Sparsity Stage, we proceed a test on objective value stationarity similarly to (98, Section 2.1) and empirically set $N \equiv T$ for CNN experiments since the validation accuracy values become stationary at the late epochs till $T$. Hence, the Group-Sparsity Stage starts after $T$ epochs and is accompanied with the $\alpha_k$ decaying. For Bert experiment, we empirically set $N$ as 1 since the F1-score and exact match rate becomes stationary after one epoch training.

**Additional Remarks.**  For the ZIG partition of ResNet50 on CIFAR10, we include all trainable variables of ResNet50 and apply the ZIG partition described in Appendix A.1 for ResConv-BN and the ZIG partition described in Section 3.1 for standard Conv-BN. For the ZIG partition of ResNet50 on ImageNet, we construct ZIGs for standard Conv-BN only. This is because we observe that ZIG partition for ResConv-BN lead to accuracy regression in spite of more FLOPs reduction, (15% FLOPs

with up to 71% Top-1 Accuracy). The cause is that it decreases the number of features maps generated by the entire Group Block. Additionally, for Bert experiments, to study the accuracy evolution against different compression rates, we set extra constraints to bound the maximum group sparsity ratio, *e.g.*, 30%, 50%, 70%, and do not yield new zero groups if the upper bound has been exceeded. Note that without any constraint, OTO reaches about 95% group sparsity ratio with 80% F1-score.

### A.3 Error Bar Analysis

In this section, we report the overall statistics of the experiments and analyze the error bar. We note that for fair comparison with others, in the main body of paper, we report the best results in terms of remaining FLOPs/parameters and Top-1/5 accuracy. We conduct all experiments three times with different random seeds.

Table 6: OTO for CNN Experiments (mean $\pm$ std)

| Model | Dataset | FLOPs | # of Params | Top-1 Acc. |
|---|---|---|---|---|
| VGG16 | CIFAR10 | $16.9\% \pm 1.5\%$ | $2.7\% \pm 0.2\%$ | $90.7\% \pm 0.3\%$ |
| VGG16-BN | CIFAR10 | $26.9\% \pm 0.1\%$ | $5.5\% \pm 0.1\%$ | $93.2\% \pm 0.2\%$ |
| ResNet50 | CIFAR10 | $11.9\% \pm 1.7\%$ | $8.8\% \pm 0.4\%$ | $93.9\% \pm 0.5\%$ |
| ResNet50 | ImageNet | $34.8\% \pm 1.8\%$ | $35.9\% \pm 1.7\%$ | $73.3\% \pm 1.1\%$ |

Training neural networks is equivalent to solving a non-convex optimization problem which has numerous local minimizers, thereby training from scratch like OTO may generate solutions close to stationary points with different attributes. As shown in Table 6, we can see that for the CNN experiments, OTO performs reliably to achieve significant FLOPs and parameters reduction and competitive Top-1 accuracy with small fluctuations.

### A.4 FLOPs Reduction Breakdown

We provide the layer-wise FLOPs reduction for VGG16 on CIFAR10. As shown in Table 7, the majority of the FLOPs reduction via OTO comes from a few middle ConvLayers (over 10% to the overall FLOPs reductions) instead of the first ConvLayer (0.45% to the overall FLOPs reduction). In general, the distribution of FLOPs reduction per Layer of OTO is similar to other pruning baselines.

Table 7: FLOPs Reduction Breakdown for the ConvLayers of VGG16 on CIFAR10

| ConvLayer Index | # of Output Channels | | FLOPs Reduction | |
|---|---|---|---|---|
| | Original | Pruned | Quantity (Million) | Percentage (%) |
| 1 | 64 | 21 | 1.19M | 0.45% |
| 2 | 64 | 45 | 29.04M | 11.07% |
| 3 | 128 | 82 | 10.47M | 3.99% |
| 4 | 128 | 110 | 17.22M | 6.57% |
| 5 | 256 | 109 | 11.97M | 4.56% |
| 6 | 256 | 68 | 33.48M | 12.77% |
| 7 | 256 | 37 | 36.30M | 13.84% |
| 8 | 512 | 13 | 18.81M | 7.17% |
| 9 | 512 | 9 | 37.73M | 14.38% |
| 10 | 512 | 7 | 37.74M | 14.39% |
| 11 | 512 | 3 | 9.44M | 3.60% |
| 12 | 512 | 5 | 9.44M | 3.60% |
| 13 | 512 | 8 | 9.44M | 3.60% |

## B Convergence Analysis of HSPG

In this section, we provide theoretical analysis of HSPG. We focus on the most popular setting of optimization problem (2) as follows

$$\underset{\boldsymbol{x}\in\mathbb{R}^n}{\text{minimize}}\ \psi(\boldsymbol{x}) := f(\boldsymbol{x}) + \lambda r(\boldsymbol{x}),\ f(\boldsymbol{x}) := \frac{1}{N}\sum_{i=1}^{N} f_i(\boldsymbol{x}), \tag{12}$$

Here $f(\boldsymbol{x})$ is defined as the average of $N$ task-specific loss functions $f_i : \mathbb{R}^n \mapsto \mathbb{R}, \forall\, i = 1, \dots, N$. The stochastic gradient $\nabla \tilde{f}$ proposed in Section 3.3 can be obtained via a uniformly chosen mini-batch $\mathcal{B} \subseteq [N]$ as follows: for any $\boldsymbol{x} \in \mathbb{R}^n$, given $\mathcal{B}$, we have

$$\nabla \tilde{f}(\boldsymbol{x}) = \nabla \bigg( \underbrace{\frac{1}{|\mathcal{B}|} \sum_{i \in \mathcal{B}} f_i(\boldsymbol{x})}_{=: f_{\mathcal{B}}(\boldsymbol{x})} \bigg), \tag{13}$$

in short, we denote above term as $\nabla f_{\mathcal{B}}(\boldsymbol{x})$ where $f_{\mathcal{B}}(\boldsymbol{x})$ is the average of loss functions with repsect to mini-batch $\mathcal{B}$. Similarly, let $\psi_{\mathcal{B}}(\boldsymbol{x}) := f_{\mathcal{B}}(\boldsymbol{x}) + \lambda r(\boldsymbol{x})$ for all $\boldsymbol{x} \in \mathbb{R}^n$.

**Organization.** The Section B is organized as follows: From Section B.1 to Section B.5, we present the convergence result and the sparse recovery guarantee for Half-Space Step. More specifically,

- In Section B.1, we first presented the existing related work of solving the problem (12).

- In Section B.2, we show the sufficient decrease of Half-Space Step under Assumption 1.

- In Section B.3, we derive the projection region of Half-Space Step and compare this projection region with existing methods.

- In Section B.4, we give the convergence result of Half-Space Step as stated in Theorem 1 under the Assumption (2, 3).

To complete the story, in Section B.5, we show that the "close enough" condition required in Theorem 1 can be achieved by the *Sub-gradient Descent Step* under the Assumption 5. Moreover, we further point out that: (1) the *Sub-gradient Descent Step* we used to achieve a "close enough" solution can be replaced by other methods, and (2) the Assumption 4 is only a sufficient condition that we could use to show the "close enough" condition.

## B.1 Related Work

Problem (12) has been well studied in deterministic optimization with various algorithms that are capable of returning solutions with both low objective value and high group sparsity under proper $\lambda$ (95; 73; 42; 64). Proximal methods are classical approaches to solve the structured non-smooth optimization (12), including the popular proximal gradient method (Prox-FG) which only uses the first-order derivative information. When $N$ is huge, stochastic methods become ubiquitous to operate on a small subset to avoid the costly evaluation over all instances in deterministic methods for large-scale problems. Proximal stochastic gradient method (Prox-SG) (19) is the natural stochastic extension of Prox-FG. Regularized dual-averaging method (RDA) (87; 92) is proposed by extending the dual averaging scheme in (66). To improve the convergence rate, there exists a set of incremental gradient methods inspired by SAG (74) to utilizes the average of accumulated past gradients. For example, proximal stochastic variance-reduced gradient method (Prox-SVRG) (88) and proximal spider (Prox-Spider) (97) are developed to adopt multi-stage schemes based on the well-known variance reduction technique SVRG proposed in (46) and Spider developed in (22) respectively. SAGA (12) stands as the midpoint between SAG and Prox-SVRG.

Compared to deterministic methods, the studies of structured sparsity regularization (12) in stochastic field become somewhat rare and limited. Prox-SG, RDA, Prox-SVRG, Prox-Spider and SAGA are valuable state-of-the-art stochastic algorithms for solving problem (12) but with apparent weakness. Particularly, these existing stochastic algorithms typically meet difficulties to achieve both decent convergence and effective group sparsity identification simultaneously (*e.g.*, small function values but merely dense solutions), because of the randomness and the limited sparsity-promotion mechanisms. In depth, Prox-SG, RDA, Prox-SVRG, Prox-Spider and SAGA derive from proximal gradient method to utilize the proximal operator to produce group of zero variables. Such operator is generic to extensive non-smooth problems, consequently perhaps not sufficiently insightful if the target problems possess certain properties, *e.g.*, the group sparsity structure as problem (12). In fact, in convex setting, the proximal operator suffers from variance of gradient estimate; and in non-convex setting, especially deep learning, the discreet step size (learning rate) further deteriorates its effectiveness on the group sparsity promotion, as shown in Section 3.3 of the main body that the projection region vanishes rapidly except RDA. RDA has superiority on finding manifold structure to others (53), but inferiority on the objective convergence. Besides, the variance reduction techniques are typically required to

measure over a huge mini-batch data points in both theory and practice which is probably prohibitive for large-scale problems, and have been observed as sometimes noneffective for deep learning applications (13). On the other hand, to introduce sparsity, there exist heuristic weight pruning methods (55; 60), whereas they commonly do not equip with theoretical guarantee, so that easily diverge and hurt generalization accuracy.

## B.2 Sufficient Decrease of Half-Space Step

Before we present the convergence result of Half-Space Step to the global group-sparsity solution, in this part, we first show that the sufficient decrease property holds for Half-Space Step under the following Assumption 1.

**Assumption 1.** *Assume the following assumptions hold.*

- *(A1-1). $f : \mathbb{R}^n \mapsto \mathbb{R}$ is differentiable and $L$ smooth.*

- *(A1-2). $r : \mathbb{R}^n \mapsto \mathbb{R}$ is sub-differentiable and convex.*

- *(A1-3). $\psi = f + \lambda r : \mathbb{R}^n \mapsto \mathbb{R}$ is sub-differentiable over all points $\boldsymbol{x} \in \mathbb{R}^n$.*

For any $k > N_{\mathcal{P}}$ (in Half-Space Step of Algorithm 2), recall the next iterate $\boldsymbol{x}_{k+1}$ and the search direction

$$\boldsymbol{d}_k := \frac{\boldsymbol{x}_{k+1} - \boldsymbol{x}_k}{\alpha_k} = \frac{\text{Proj}_{\mathcal{S}_k}^{\text{HS}}(\boldsymbol{x}_k - \alpha_k \nabla\psi_{\mathcal{B}_k}(\boldsymbol{x}_k)) - \boldsymbol{x}_k}{\alpha_k}. \tag{14}$$

Define

$$\hat{\mathcal{G}}_k := \mathcal{I}^{\neq 0}(\boldsymbol{x}_k) \cap \mathcal{I}^0(\boldsymbol{x}_{k+1}) \tag{15}$$

$$\tilde{\mathcal{G}}_k := \mathcal{I}^{\neq 0}(\boldsymbol{x}_k) \cap \mathcal{I}^{\neq 0}(\boldsymbol{x}_{k+1}) \tag{16}$$

be the sets of groups which projects or not onto zero. We claim that the following Lemma 1 holds.

**Lemma 1.** *Under Assumption 1, the search direction $\boldsymbol{d}_k$ is a descent direction for $\psi_{\mathcal{B}_k}(\boldsymbol{x}_k)$, i.e., $\boldsymbol{d}_k^\top \nabla\psi_{\mathcal{B}_k}(\boldsymbol{x}_k) < 0$. Moreover, we have the following sufficient decrease property holds,*

$$\psi_{\mathcal{B}_k}(\boldsymbol{x}_{k+1}) \leq \psi_{\mathcal{B}_k}(\boldsymbol{x}_k) - \left(\alpha_k - \frac{\alpha_k^2 L}{2}\right) \sum_{g \in \tilde{\mathcal{G}}_k} \|[\nabla\psi_{\mathcal{B}_k}(\boldsymbol{x}_k)]_g\|^2 - \left(\frac{1-\epsilon}{\alpha_k} - \frac{L}{2}\right) \sum_{g \in \hat{\mathcal{G}}_k} \|[\boldsymbol{x}_k]_g\|^2. \tag{17}$$

*Proof.* **Proof of Descent Direction.** It follows the Half-Space Step in Algorithm 2 and the definition of $\tilde{\mathcal{G}}_k$ and $\hat{\mathcal{G}}_k$ as (16) and (15) that $\boldsymbol{x}_{k+1} = \boldsymbol{x}_k + \alpha_k \boldsymbol{d}_k$ where $\boldsymbol{d}_k$ is

$$[\boldsymbol{d}_k]_g = \begin{cases} -[\nabla\psi_{\mathcal{B}_k}(\boldsymbol{x}_k)]_g & \text{if } g \in \tilde{\mathcal{G}}_k = \mathcal{I}^{\neq 0}(\boldsymbol{x}_k) \bigcap \mathcal{I}^{\neq 0}(\boldsymbol{x}_{k+1}), \\ -[\boldsymbol{x}_k]_g/\alpha_k & \text{if } g \in \hat{\mathcal{G}}_k = \mathcal{I}^{\neq 0}(\boldsymbol{x}_k) \bigcap \mathcal{I}^0(\boldsymbol{x}_{k+1}), \\ 0 & \text{otherwise.} \end{cases} \tag{18}$$

We also notice that for any $g \in \hat{\mathcal{G}}_k$, the following holds

$$\begin{aligned}
[\boldsymbol{x}_k - \alpha_k \nabla\psi_{\mathcal{B}_k}(\boldsymbol{x}_k)]_g^\top [\boldsymbol{x}_k]_g &< \epsilon \|[\boldsymbol{x}_k]_g\|^2, \\
(1-\epsilon)\|[\boldsymbol{x}_k]_g\|^2 &< \alpha_k [\nabla\psi_{\mathcal{B}_k}(\boldsymbol{x}_k)]_g^\top [\boldsymbol{x}_k]_g.
\end{aligned} \tag{19}$$

For simplicity, let $\mathcal{I}_k^{\neq 0} := \mathcal{I}^{\neq 0}(\boldsymbol{x}_k)$. Since $[\boldsymbol{d}_k]_g = \boldsymbol{0}$ for any $g \in \mathcal{I}^0(\boldsymbol{x}_k)$, then by (18) and (19), we have

$$\begin{aligned}
\boldsymbol{d}_k^\top \nabla\psi_{\mathcal{B}_k}(\boldsymbol{x}_k) &= [\boldsymbol{d}_k]_{\mathcal{I}_k^{\neq 0}}^\top [\nabla\psi_{\mathcal{B}_k}(\boldsymbol{x}_k)]_{\mathcal{I}_k^{\neq 0}} \\
&= -\sum_{g \in \tilde{\mathcal{G}}_k} \|[\nabla\psi_{\mathcal{B}_k}(\boldsymbol{x}_k)]_g\|^2 - \sum_{g \in \hat{\mathcal{G}}_k} \frac{1}{\alpha_k} [\boldsymbol{x}_k]_g^\top [\nabla\psi_{\mathcal{B}_k}(\boldsymbol{x}_k)]_g \\
&\leq -\sum_{g \in \tilde{\mathcal{G}}_k} \|[\nabla\psi_{\mathcal{B}_k}(\boldsymbol{x}_k)]_g\|^2 - \sum_{g \in \hat{\mathcal{G}}_k} \frac{1}{\alpha_k^2}(1-\epsilon)\|[\boldsymbol{x}_k]_g\|^2 < 0,
\end{aligned} \tag{20}$$

holds for any $\epsilon \in [0, 1)$, which implies that $\boldsymbol{d}_k$ is a descent direction for $\psi_{\mathcal{B}_k}(\boldsymbol{x}_k)$.

**Proof of Sufficient Decrease.** Now, we start to prove the suffcient decrease of Half-Space Step. By assumption, $f : \mathbb{R}^n \mapsto \mathbb{R}$ is $L$ smooth and $r : \mathbb{R}^n \mapsto \mathbb{R}$ is convex. Therefore

$$\psi_{\mathcal{B}_k}(\boldsymbol{x}_k + \alpha_k \boldsymbol{d}_k) \tag{21}$$

$$= f_{\mathcal{B}_k}(\boldsymbol{x}_k + \alpha_k \boldsymbol{d}_k) + \lambda r(\boldsymbol{x}_k + \alpha_k \boldsymbol{d}_k) \tag{22}$$

$$\leq f_{\mathcal{B}_k}(\boldsymbol{x}_k) + \alpha_k \boldsymbol{d}_k^\top \nabla f_{\mathcal{B}}(\boldsymbol{x}_k) + \frac{\alpha_k^2 L}{2} \|\boldsymbol{d}_k\|^2 \qquad \text{by Assumption 1} \tag{23}$$

$$+ \lambda r(\boldsymbol{x}_k) + \alpha_k \lambda \boldsymbol{d}_k^\top \zeta(\boldsymbol{x}_k) \tag{24}$$

$$= \psi_{\mathcal{B}_k}(\boldsymbol{x}_k) + \alpha_k \boldsymbol{d}_k^\top \nabla \psi_{\mathcal{B}_k}(\boldsymbol{x}_k) + \frac{\alpha_k^2 L}{2} \|\boldsymbol{d}_k\|^2 \tag{25}$$

$$\leq \psi_{\mathcal{B}_k}(\boldsymbol{x}_k) - \left( \alpha_k - \frac{\alpha_k^2 L}{2} \right) \sum_{g \in \tilde{\mathcal{G}}_k} \|[\nabla \psi_{\mathcal{B}_k}(\boldsymbol{x}_k)]_g\|^2 \quad \text{by inequality (20) \& } \boldsymbol{d}_k \text{ definition} \tag{26}$$

$$- \left( \frac{1 - \epsilon}{\alpha_k} - \frac{L}{2} \right) \sum_{g \in \hat{\mathcal{G}}_k} \|[\boldsymbol{x}_k]_g\|^2 , \tag{27}$$

which completes the proof. $\qquad\square$

According to Lemma 1, the objective value $\psi_{\mathcal{B}}(\boldsymbol{x})$ with $\mathbb{E}[\psi_{\mathcal{B}}(\boldsymbol{x})|\boldsymbol{x}] = \psi(\boldsymbol{x})$ achieves a sufficient decrease in Half-Space Step given $\alpha_k$ is small enough. Taking the expectation over mini-batch $\mathcal{B}$ on both sides, it is straight-forward to obtain the expectation version of the sufficient decrease property.

**Corollary 1.** *Similarly, under Assumption 1, for all $k > N_{\mathcal{P}}$, we have*

$$\psi(\boldsymbol{x}_{k+1}) \leq \psi(\boldsymbol{x}_k) - \sum_{g \in \tilde{\mathcal{G}}_k} \left( \alpha_k - \frac{\alpha_k^2 L}{2} \right) \mathbb{E}\left[ \|[\nabla \psi_{\mathcal{B}_k}(\boldsymbol{x}_k)]_g\|^2 \right] - \left( \frac{1 - \epsilon}{\alpha_k} - \frac{L}{2} \right) \sum_{g \in \hat{\mathcal{G}}_k} \|[\boldsymbol{x}_k]_g\|^2 . \tag{28}$$

### B.3 Projection Region of Half-Space Step

In this part, we derive the projection region of Half-Space Step, and reveal that is a superset of the projection region of existing methods, e.g. Prox-SG, Prox-SVRG and Prox-Spider, under the same $\alpha_k$ and $\lambda$.

**Proposition 1.** *For any $k > N_{\mathcal{P}}$, given $\boldsymbol{x}_k$, the next iterate $\boldsymbol{x}_{k+1}$ obtained by the Half-Space Step satisfies that: for any group $g \in \mathcal{I}^{\neq 0}(\boldsymbol{x}_k)$,*

$$[\boldsymbol{x}_{k+1}]_g = \begin{cases} [\hat{\boldsymbol{x}}_{k+1}]_g - \alpha_k \lambda \frac{[\boldsymbol{x}_k]_g}{\|[\boldsymbol{x}_k]_g\|} & \text{if } [\hat{\boldsymbol{x}}_{k+1}]_g^\top [\boldsymbol{x}_k]_g > (\alpha_k \lambda + \epsilon) \|[\boldsymbol{x}_k]_g\| , \\ 0 & \text{otherwise}, \end{cases} \tag{29}$$

*where $\hat{\boldsymbol{x}}_{k+1} := \boldsymbol{x}_k - \alpha_k \nabla f_{\mathcal{B}_k}(\boldsymbol{x}_k)$. Moreover, we claim that if $\|[\hat{\boldsymbol{x}}_{k+1}]_g\| \leq \alpha_k \lambda$, then $[\boldsymbol{x}_{k+1}]_g = 0$ for any $\epsilon \geq 0$.*

*Proof.* For $g \in \mathcal{I}^{\neq 0}(\boldsymbol{x}_k) \bigcap \mathcal{I}^{\neq 0}(\boldsymbol{x}_{k+1})$, by line 11-12 in Algorithm 2, it is equivalent to

$$\left[ \boldsymbol{x}_k - \alpha_k \nabla f_{\mathcal{B}_k}(\boldsymbol{x}_k) - \alpha_k \lambda \frac{[\boldsymbol{x}_k]_g}{\|[\boldsymbol{x}_k]_g\|} \right]_g^\top [\boldsymbol{x}_k]_g > \epsilon \|[\boldsymbol{x}_k]_g\|^2 ,$$

$$[\hat{\boldsymbol{x}}_{k+1}]_g^\top [\boldsymbol{x}_k]_g - \alpha_k \lambda \|[\boldsymbol{x}_k]_g\| > \epsilon \|[\boldsymbol{x}_k]_g\|^2 , \tag{30}$$

$$[\hat{\boldsymbol{x}}_{k+1}]_g^\top [\boldsymbol{x}_k]_g > (\alpha_k \lambda + \epsilon \|[\boldsymbol{x}_k]_g\|) \|[\boldsymbol{x}_k]_g\| .$$

Similarly, $g \in \mathcal{I}^{\neq 0}(\boldsymbol{x}_k) \bigcap \mathcal{I}^0(\boldsymbol{x}_{k+1})$ is equivalent to

$$\left[ \boldsymbol{x}_k - \alpha_k \nabla f_{\mathcal{B}_k}(\boldsymbol{x}_k) - \alpha_k \lambda \frac{[\boldsymbol{x}_k]_g}{\|[\boldsymbol{x}_k]_g\|} \right]_g^\top [\boldsymbol{x}_k]_g \leq \epsilon \|[\boldsymbol{x}_k]_g\|^2 ,$$

$$[\hat{\boldsymbol{x}}_{k+1}]_g^\top [\boldsymbol{x}_k]_g - \alpha_k \lambda \|[\boldsymbol{x}_k]_g\| \leq \epsilon \|[\boldsymbol{x}_k]_g\|^2 , \tag{31}$$

$$[\hat{\boldsymbol{x}}_{k+1}]_g^\top [\boldsymbol{x}_k]_g \leq (\alpha_k \lambda + \epsilon \|[\boldsymbol{x}_k]_g\|) \|[\boldsymbol{x}_k]_g\| .$$

If $\|[\hat{\boldsymbol{x}}_{k+1}]_g\| \leq \alpha_k \lambda$, then

$$[\hat{\boldsymbol{x}}_{k+1}]_g^\top [\boldsymbol{x}_k]_g \leq \|[\hat{\boldsymbol{x}}_{k+1}]_g\| \, \|[\boldsymbol{x}_k]_g\| \leq \alpha_k \lambda \, \|[\boldsymbol{x}_k]_g\|. \tag{32}$$

Hence $[\boldsymbol{x}_{k+1}]_g = 0$ holds for any $\epsilon \geq 0$ by (31), which implies that the projection region of Prox-SG and its variance reduction variants, *e.g.*, Prox-SVRG, Prox-Spider and SAGA are the subsets of HSPG's. $\qquad \square$

## B.4 Convergence Analysis of Half-Space Step

In this section, we give the convergence result of Half-Space Step under the following Assumptions for the properties of the objective function and the global optimal solution $\boldsymbol{x}^*$ of (2).

**Assumption 2.** *Assume the following assumptions hold.*

- *(A2-1). For $i = 1, 2, \cdots, N$, each $f_i : \mathbb{R}^n \to \mathbb{R}$ is differentiable and bounded below.*

- *(A2-2). For $i = 1, 2, \cdots, N$, each $f_i : \mathbb{R}^n \to \mathbb{R}$ is $L_i$ smooth.*

- *(A2-3). $\psi_{\mathcal{B}} = f_{\mathcal{B}} + \lambda r : \mathbb{R}^n \mapsto \mathbb{R}$ has bounded sub-gradient (i.e., $\mathbb{E}[\|\nabla \psi_{\mathcal{B}}(\boldsymbol{x})\|^2] \leq M^2$ for some universal constant $M$) over all points $\boldsymbol{x} \in \mathbb{R}^n$ with respect to any mini-batch $\mathcal{B} \subseteq [N]$.*

- *(A2-4). The stochastic gradient $\nabla f_{\mathcal{B}}(\boldsymbol{x})$ satisfies $\mathbb{E}_{\mathcal{B}}[\nabla f_{\mathcal{B}}(\boldsymbol{x}) | \boldsymbol{x}] = \nabla f(\boldsymbol{x})$ for all $\boldsymbol{x} \in \mathbb{R}^n$.*

- *(A2-5). The stochastic gradient $\nabla f_{\mathcal{B}}(\boldsymbol{x})$ satisfies $Var_{\mathcal{B}}[\nabla f_{\mathcal{B}}(\boldsymbol{x}) | \boldsymbol{x}] \leq \sigma^2$ for all $\boldsymbol{x} \in \mathbb{R}^n$, where $\sigma^2 > 0$ is a universal constant.*

Notice that this Assumption 2 is a variant of the Assumption 1, to be concise, we set $L$ proposed in Assumption 1 as $L := \max_{i=1}^N \{L_i\}$.

**Assumption 3.** *Assume the following assumptions hold.*

- *(A3-1). $\sum_{k \geq N_{\mathcal{P}}} \alpha_k = \infty$.*

- *(A3-2). $\sum_{k \geq N_{\mathcal{P}}} \alpha_k^2 < \infty$.*

**Assumption 4.** *The least and the largest $\ell_2$-norm of non-zero groups in $\boldsymbol{x}^*$ are lower and upper bounded by some constants,*

$$0 < 2\delta_1 := \min_{g \in \mathcal{I}^{\neq 0}(\boldsymbol{x}^*)} \|[\boldsymbol{x}^*]_g\| \leq \max_{g \in \mathcal{I}^{\neq 0}(\boldsymbol{x}^*)} \|[\boldsymbol{x}^*]_g\| =: 2\delta_2. \tag{33}$$

**Theorem 1.** *Under Assumptions (1, 2, 3, 4), set*

$$R \in \left( 0, \ \min \left\{ \frac{1}{\epsilon} \cdot \left[ -(\delta_1 + 2\epsilon\delta_2) + \sqrt{(\delta_1 + 2\epsilon\delta_2)^2 - 4\epsilon^2\delta_2 + 4\epsilon\delta_1^2} \right], \delta_1 \right\} \right), \tag{34}$$

$$\epsilon \in \left[ 0, \ \min \left\{ \frac{\delta_1^2}{\delta_2}, \frac{2\delta_1 - R}{2\delta_2 + R} \right\} \right), \tag{35}$$

$$\alpha_k \in \left( 0, \ \min \left\{ \frac{2(1-\epsilon)}{L}, \frac{1}{L}, \frac{2\delta_1 - R - \epsilon(2\delta_2 + R)}{M} \right\} \right), \quad \forall k \geq N_{\mathcal{P}}. \tag{36}$$

*If there exists a $K \geq N$ such that*

$$\|\boldsymbol{x}_K - \boldsymbol{x}^*\| \leq \frac{R}{2}. \tag{37}$$

*Given any $\tau \in (0, 1)$, there exists some $\alpha_k = \mathcal{O}(1/(1+\sqrt{\tau})(k-K))$ and $|\mathcal{B}_k| = \mathcal{O}(k-K)$ for all $k \geq K$ such that the sequence $\{\boldsymbol{x}_k\}_{k \geq K}$ obtained from the Algorithm 2 converges to some stationary point with probability at least $1 - \tau$, i.e.,*

$$\liminf_k \mathbb{E}\left[ \|\nabla \psi_{\mathcal{B}_k}(\boldsymbol{x}_k)\| \right] = 0 \quad \text{with probability} \quad 1 - \tau. \tag{38}$$

*Proof.* **Proof Sketch.** We split the proof of showing the convergence to some stationary points into two parts. In the first part, we show the convergence holds for all groups in $\tilde{\mathcal{G}}_k$; and in the second part, we show the convergence also holds in $\hat{\mathcal{G}}_k$.

**Convergence in $\tilde{\mathcal{G}}_k$ part.** For any $t \in \mathbb{N}_+$, applying Corollary 1 yields

$$\psi(\boldsymbol{x}_{N_{\mathcal{P}}}) - \psi(\boldsymbol{x}_{N_{\mathcal{P}}+t}) \tag{39}$$

$$= \sum_{k=N_{\mathcal{P}}}^{N_{\mathcal{P}}+t-1} \psi(\boldsymbol{x}_k) - \psi(\boldsymbol{x}_{k+1}) \tag{40}$$

$$\geq \sum_{k=N_{\mathcal{P}}}^{N_{\mathcal{P}}+t-1} \sum_{g \in \tilde{\mathcal{G}}_k} \left(\alpha_k - \frac{\alpha_k^2 L}{2}\right) \mathbb{E}\left[\|[\nabla\psi_{\mathcal{B}_k}(\boldsymbol{x}_k)]_g\|^2\right] + \sum_{k=N_{\mathcal{P}}}^{N_{\mathcal{P}}+t-1} \left(\frac{1-\epsilon}{\alpha_k} - \frac{L}{2}\right) \sum_{g \in \hat{\mathcal{G}}_k} \|[\boldsymbol{x}_k]_g\|^2. \tag{41}$$

Combining the assumption that $\psi$ is bounded below and letting $t \to \infty$ yield

$$\underbrace{\sum_{k=N_{\mathcal{P}}}^{\infty} \sum_{g \in \tilde{\mathcal{G}}_k} \left(\alpha_k - \frac{\alpha_k^2 L}{2}\right) \mathbb{E}\left[\|[\nabla\psi_{\mathcal{B}_k}(\boldsymbol{x}_k)]_g\|^2\right]}_{=:T_1} + \underbrace{\sum_{k=N_{\mathcal{P}}}^{\infty} \left(\frac{1-\epsilon}{\alpha_k} - \frac{L}{2}\right) \sum_{g \in \hat{\mathcal{G}}_k} \|[\boldsymbol{x}_k]_g\|^2}_{=:T_2} < \infty. \tag{42}$$

Given $\alpha_k \in (0, 2(1-\epsilon)/L)$, we have $T_1 > 0, T_2 > 0$, combining with $T_1 + T_2 < \infty$ implies

$$\sum_{k=N_{\mathcal{P}}}^{\infty} \sum_{g \in \tilde{\mathcal{G}}_k} \left(\alpha_k - \frac{\alpha_k^2 L}{2}\right) \mathbb{E}\left[\|[\nabla\psi_{\mathcal{B}_k}(\boldsymbol{x}_k)]_g\|^2\right] \tag{43}$$

$$= \sum_{k=N_{\mathcal{P}}}^{\infty} \sum_{g \in \tilde{\mathcal{G}}_k} \alpha_k \mathbb{E}\left[\|[\nabla\psi_{\mathcal{B}_k}(\boldsymbol{x}_k)]_g\|^2\right] - \sum_{k=N_{\mathcal{P}}}^{\infty} \sum_{g \in \tilde{\mathcal{G}}_k} \frac{\alpha_k^2 L}{2} \mathbb{E}\left[\|[\nabla\psi_{\mathcal{B}_k}(\boldsymbol{x}_k)]_g\|^2\right]. \tag{44}$$

Based on the boundness of sub-gradient in Assumptions 2 and the choice of stepsize in 3, we have

$$\sum_{k=N_{\mathcal{P}}}^{\infty} \sum_{g \in \tilde{\mathcal{G}}_k} \frac{\alpha_k^2 L}{2} \mathbb{E}\left[\|[\nabla\psi_{\mathcal{B}_k}(\boldsymbol{x}_k)]_g\|^2\right] < \infty, \tag{45}$$

which yields

$$\sum_{k=N_{\mathcal{P}}}^{\infty} \sum_{g \in \tilde{\mathcal{G}}_k} \alpha_k \mathbb{E}\left[\|[\nabla\psi_{\mathcal{B}_k}(\boldsymbol{x}_k)]_g\|^2\right] < \infty \tag{46}$$

$$\Rightarrow \liminf_{k \geq N_{\mathcal{P}}} \sum_{g \in \tilde{\mathcal{G}}_k} \mathbb{E}\left[\|[\nabla\psi_{\mathcal{B}_k}(\boldsymbol{x}_k)]_g\|^2\right] = 0 \tag{47}$$

$$\Rightarrow \lim_{k \geq \mathcal{K}} \sum_{g \in \tilde{\mathcal{G}}_k} \mathbb{E}\left[\|[\nabla\psi_{\mathcal{B}_k}(\boldsymbol{x}_k)]_g\|^2\right] = 0, \quad \exists \mathcal{K} \subseteq \{N_{\mathcal{P}}, \ldots\} \tag{48}$$

**Convergence in $\hat{\mathcal{G}}_k$ part.** Under Assumption 4, Lemma (2, 3, 4) show that if there exists a $K \geq N_{\mathcal{P}}$ such that

$$\|\boldsymbol{x}_K - \boldsymbol{x}^*\| \leq R, \tag{49}$$

then we have the following results hold

$$\mathcal{I}^{\neq 0}(\boldsymbol{x}^*) \subseteq \mathcal{I}^{\neq 0}(\boldsymbol{x}_K), \qquad\qquad \text{non-zero group coverage,} \tag{50}$$

$$\boldsymbol{x}^* \in \mathcal{S}_K, \qquad\qquad \text{correct optimal inclusion } \mathcal{S}_K, \tag{51}$$

$$\mathcal{I}^{\neq 0}(\boldsymbol{x}_K) \cap \mathcal{I}^{=0}(\boldsymbol{x}_{K+1}) \subseteq \mathcal{I}^{=0}(\boldsymbol{x}^*), \qquad\qquad \text{correct zero group projection.} \tag{52}$$

Under Assumption (2, 3, 4), Lemma (5, 6, 7) and Corollary 2 show that: given any $\tau \in (0, 1)$, with probability at least $1 - \tau$, for any $k \geq K$, $\boldsymbol{x}^*$ inhabits $\mathcal{S}_k$. Therefore, for any $k \geq K$, any group $g \in \hat{\mathcal{G}}_k = \mathcal{I}^{\neq 0}(\boldsymbol{x}_k) \cap \mathcal{I}^{=0}(\boldsymbol{x}_{k+1})$ will be projected to zero group correctly with probability at least $1 - \tau$.

**Convergence over the whole space.** Based on the discussion in $\hat{\mathcal{G}}_k$ part, it is sufficient to focus on the subspace of $\tilde{\mathcal{G}}_k$. Hence, (48) naturally implies that the sequence $\{\boldsymbol{x}_k\}_{k\in\mathcal{K}}$ converges to some stationary point with high probability. By the above, we conclude that

$$\mathbb{P}\left(\liminf_k \mathbb{E}\left[\|\nabla\psi_{\mathcal{B}_k}(\boldsymbol{x}_k)\|\right] = 0\right) \geq 1 - \tau. \tag{53}$$

$\square$

### B.4.1 Support Lemma in the Proof of Theorem 1

The Lemma 2 shows that if the optimal distance from the current iterate $\boldsymbol{x}_k$ to any local minimizer $\boldsymbol{x}^*$ is sufficiently small, then HSPG already covers the supports of $\boldsymbol{x}^*$, *i.e.*, $\mathcal{I}^{\neq 0}(\boldsymbol{x}^*) \subseteq \mathcal{I}^{\neq 0}(\boldsymbol{x}_k)$.

**Lemma 2.** *Under Assumption 4, given any $R \leq \delta_1$, for any $k \geq N_{\mathcal{P}}$, if $\|\boldsymbol{x}_k - \boldsymbol{x}^*\| \leq R$, then we have $\mathcal{I}^{\neq 0}(\boldsymbol{x}^*) \subseteq \mathcal{I}^{\neq 0}(\boldsymbol{x}_k)$.*

*Proof.* For any $g \in \mathcal{I}^{\neq 0}(\boldsymbol{x}^*)$, we have that

$$\begin{aligned}
\|[\boldsymbol{x}^*]_g\| - \|[\boldsymbol{x}_k]_g\| &\leq \|[\boldsymbol{x}_k - \boldsymbol{x}^*]_g\| \leq \|\boldsymbol{x}_k - \boldsymbol{x}^*\| \leq R \leq \delta_1 \\
\|[\boldsymbol{x}_k]_g\| &\geq \|[\boldsymbol{x}^*]_g\| - \delta_1 \geq 2\delta_1 - \delta_1 = \delta_1 > 0
\end{aligned} \tag{54}$$

Hence $\|[\boldsymbol{x}_k]_g\| \neq 0$, *i.e.*, $g \in \mathcal{I}^{\neq 0}(\boldsymbol{x}_k)$. Therefore, $\mathcal{I}^{\neq 0}(\boldsymbol{x}^*) \subseteq \mathcal{I}^{\neq 0}(\boldsymbol{x}_k)$. $\square$

The Lemma 3 shows that if the distance between the current iterate $\boldsymbol{x}_k$ and $\boldsymbol{x}^*$, *i.e.*, $\|\boldsymbol{x}_k - \boldsymbol{x}^*\|$ is sufficiently small, then $\boldsymbol{x}^*$ inhabits the reduced space $\mathcal{S}_k := \mathcal{S}(\boldsymbol{x}_k)$.

**Lemma 3.** *Under Assumption 4, for any $k \geq N_{\mathcal{P}}$, given $\epsilon \in [0, \delta_1^2/\delta_2)$ and*

$$R \leq R^* := \frac{1}{\epsilon} \cdot \left[-(\delta_1 + 2\epsilon\delta_2) + \sqrt{(\delta_1 + 2\epsilon\delta_2)^2 - 4\epsilon^2\delta_2 + 4\epsilon\delta_1^2}\right], \tag{55}$$

*if $\|\boldsymbol{x}_k - \boldsymbol{x}^*\| \leq R$, we have*

$$[\boldsymbol{x}_k]_g^\top[\boldsymbol{x}^*]_g \geq \epsilon\|[\boldsymbol{x}_k]_g\|^2, \quad g \in \mathcal{I}^{\neq 0}(\boldsymbol{x}^*). \tag{56}$$

*Consequently, it implies $\boldsymbol{x}^* \in \mathcal{S}_k$ by the definition as (4).*

*Proof.* For any $g \in \mathcal{I}^{\neq 0}(\boldsymbol{x}^*)$,

$$\|[\boldsymbol{x}_k]_g\| \leq \|[\boldsymbol{x}^*]_g\| + R \leq 2\delta_2 + R, \tag{57}$$

and the $R^*$ defined in (55) is one of the roots of the quadratic $\epsilon z^2 + (4\epsilon\delta_2 + 2\delta_1)z + 4\epsilon\delta_2^2 - 4\delta_1^2 = 0$ regarding $z \in \mathbb{R}$. Thus

$$\begin{aligned}
[\boldsymbol{x}_k]_g^\top[\boldsymbol{x}^*]_g &= [\boldsymbol{x}_k - \boldsymbol{x}^* + \boldsymbol{x}^*]_g^\top[\boldsymbol{x}^*]_g \\
&= [\boldsymbol{x}_k - \boldsymbol{x}^*]_g^\top[\boldsymbol{x}^*]_g + \|[\boldsymbol{x}^*]_g\|^2 \\
&\geq \|[\boldsymbol{x}^*]_g\|^2 - \|[\boldsymbol{x}_k - \boldsymbol{x}^*]_g\|\|[\boldsymbol{x}^*]_g\| \\
&= \|[\boldsymbol{x}^*]_g\|(\|[\boldsymbol{x}^*]_g\| - \|[\boldsymbol{x}_k - \boldsymbol{x}^*]_g\|) \\
&\geq 2\delta_1(2\delta_1 - R) \geq \epsilon(2\delta_2 + R)^2 \\
&\geq \epsilon\|[\boldsymbol{x}_k]_g\|^2
\end{aligned} \tag{58}$$

holds for any $g \in \mathcal{I}^{\neq 0}(\boldsymbol{x}^*)$, where the second last inequality holds because that $2\delta_1(2\delta_1 - R) = \epsilon(2\delta_2 + R)^2$ as $R = R^*$. Now combing with the definition of $\mathcal{S}_k$ as (4), we have $\boldsymbol{x}^*$ inhabits $\mathcal{S}_k$, which completes the proof. $\square$

The Lemma 4 shows that if $\|\boldsymbol{x}_k - \boldsymbol{x}^*\|$ is small enough and the step size is selected properly, every recovery of group sparsity by Half-Space Step can be guaranteed as successful as stated in the following lemma.

**Lemma 4.** *Under Assumption 4, for any* $k \geq N_\mathcal{P}$, *given* $\epsilon \in \left[0, \frac{2\delta_1-R}{2\delta_2+R}\right)$, $\alpha_k \in \left(0, \frac{2\delta_1-R-\epsilon(2\delta_2+R)}{M}\right)$ *and* $R \in (0, \min\{R^*, \delta_1\})$, *if* $\|\boldsymbol{x}_k - \boldsymbol{x}^*\| \leq R$, *then for any* $g \in \hat{\mathcal{G}}_k = \mathcal{I}^{\neq 0}(\boldsymbol{x}_k) \bigcap \mathcal{I}^0(\boldsymbol{x}_{k+1})$, *we have* $g \in \mathcal{I}^0(\boldsymbol{x}^*)$.

*Proof.* To prove it by contradiction, suppose there exists some $g \in \hat{\mathcal{G}}_k$ such that $g \in \mathcal{I}^{\neq 0}(\boldsymbol{x}^*)$. Since $g \in \hat{\mathcal{G}}_k = \mathcal{I}^{\neq 0}(\boldsymbol{x}_k) \bigcap \mathcal{I}^0(\boldsymbol{x}_{k+1})$, then the group projection (6) is trigerred at $g$ such that

$$
\begin{aligned}
[\tilde{\boldsymbol{x}}_{k+1}]_g^\top [\boldsymbol{x}_k]_g &= [\boldsymbol{x}_k - \alpha \nabla \psi_{\mathcal{B}_k}(\boldsymbol{x}_k)]_g^\top [\boldsymbol{x}_k]_g \\
&= \|[\boldsymbol{x}_k]_g\|^2 - \alpha_k [\nabla \psi_{\mathcal{B}_k}(\boldsymbol{x}_k)]_g^\top [\boldsymbol{x}_k]_g < \epsilon \|[\boldsymbol{x}_k]_g\|^2 .
\end{aligned}
\tag{59}
$$

On the other hand, it follows the assumption of this lemma and $g \in \mathcal{I}^{\neq 0}(\boldsymbol{x}^*)$ that

$$
\|[\boldsymbol{x}_k - \boldsymbol{x}^*]_g\| \leq \|\boldsymbol{x}_k - \boldsymbol{x}^*\| \leq R
\tag{60}
$$

Combining the definition of $\delta_1$ and $\delta_2$ in Assumption 4, we have that

$$
\begin{aligned}
\|[\boldsymbol{x}_k]_g\| &\geq \|[\boldsymbol{x}^*]_g\| - R \geq 2\delta_1 - R \\
\|[\boldsymbol{x}_k]_g\| &\leq \|[\boldsymbol{x}^*]_g\| + R \leq 2\delta_2 + R
\end{aligned}
\tag{61}
$$

It then follows $0 < \alpha_k \leq \frac{2\delta_1-R-\epsilon(2\delta_2+R)}{M}$, where note $2\delta_1 - R - \epsilon(2\delta_2 + R) > 0$ as $R \leq \delta_1$ and $\epsilon < \frac{2\delta_1-R}{2\delta_2+R}$, that

$$
\begin{aligned}
[\tilde{\boldsymbol{x}}_{k+1}]_g^\top [\boldsymbol{x}_k]_g &= \|[\boldsymbol{x}_k]_g\|^2 - \alpha_k [\nabla \psi_{\mathcal{B}_k}(\boldsymbol{x}_k)]_g^\top [\boldsymbol{x}_k]_g \\
&\geq \|[\boldsymbol{x}_k]_g\|^2 - \alpha_k \|[\nabla \psi_{\mathcal{B}_k}(\boldsymbol{x}_k)]_g\| \|[\boldsymbol{x}_k]_g\| \\
&= \|[\boldsymbol{x}_k]_g\| \left(\|[\boldsymbol{x}_k]_g\| - \alpha_k \|[\nabla \psi_{\mathcal{B}_k}(\boldsymbol{x}_k)]_g\|\right) \\
&\geq \|[\boldsymbol{x}_k]_g\| \left(\|[\boldsymbol{x}_k]_g\| - \alpha_k M\right) \\
&\geq \|[\boldsymbol{x}_k]_g\| \left[(2\delta_1 - R) - \alpha_k M\right] \\
&\geq \|[\boldsymbol{x}_k]_g\| \left[(2\delta_1 - R) - \frac{2\delta_1 - R - \epsilon(2\delta_2 + R)}{M} M\right] \\
&\geq \|[\boldsymbol{x}_k]_g\| \left[(2\delta_1 - R) - 2\delta_1 + R + \epsilon(2\delta_2 + R)\right] \\
&\geq \epsilon \|[\boldsymbol{x}_k]_g\| (2\delta_2 + R) \\
&\geq \epsilon \|[\boldsymbol{x}_k]_g\|^2
\end{aligned}
\tag{62}
$$

which contradicts with (59). Hence, we conclude that any $g$ of variables projected to zero, *i.e.*, $g \in \hat{\mathcal{G}}_k = \mathcal{I}^{\neq 0}(\boldsymbol{x}_k) \bigcap \mathcal{I}^0(\boldsymbol{x}_{k+1})$ are exactly also the zeros on the optimal solution $\boldsymbol{x}^*$, *i.e.*, $g \in \mathcal{I}^0(\boldsymbol{x}^*)$. □

We next present that if the iterate of Half-Space Step is close enough to the optimal solution $\boldsymbol{x}^*$, then $\boldsymbol{x}^*$ inhabits all reduced spaces constructed by the subsequent iterates of Half-Space Step with high probability.

To establish this results, we require the following two lemmas (Lemma 5 and Lemma 6). The Lemma 5 bounds the accumulated error because of random sampling. Here we introduce the error of gradient estimator on $\mathcal{I}^{\neq 0}(\boldsymbol{x})$ for $\psi$ on mini-batch $\mathcal{B}$ as

$$
\boldsymbol{e}_\mathcal{B}(\boldsymbol{x}) := [\nabla \psi_\mathcal{B}(\boldsymbol{x}) - \nabla \psi(\boldsymbol{x})]_{\mathcal{I}^{\neq 0}(\boldsymbol{x})},
\tag{63}
$$

where by the definition of $r$ in problem (12), we have $\boldsymbol{e}_\mathcal{B}(\boldsymbol{x})$ also equals to the error of estimation for $\nabla f$, i.e., $\boldsymbol{e}_\mathcal{B}(\boldsymbol{x}) = [\nabla f_\mathcal{B}(\boldsymbol{x}) - \nabla f(\boldsymbol{x})]_{\mathcal{I}^{\neq 0}(\boldsymbol{x})}$.

**Lemma 5.** *Under Assumption 2, given any* $\theta > 1$, $K \geq N_\mathcal{P}$, *let* $k := K + t$ *with* $t \in \mathbb{Z}_{\geq 0}$, *then there exists a sequence of stepsize* $\alpha_k = \mathcal{O}(1/(1 + \theta)t)$ *and corresponding size of mini-batch* $|\mathcal{B}_k| = \mathcal{O}(t)$, *such that for any* $y_t \in \mathbb{R}^n$,

$$
\max_{\{\boldsymbol{y}_t\}_{t=0}^\infty \in \mathcal{X}^\infty} \sum_{t=0}^\infty \alpha_k \|e_{\mathcal{B}_k}(\boldsymbol{y}_t)\|_2 \leq \frac{3R^2}{8(4R+1)}
$$

*holds with probability at least* $1 - \frac{1}{\theta^2}$.

*Proof.* Define random variable $Y_t := \alpha_{K+t} \|e_{\mathcal{B}_{K+t}}(\boldsymbol{y}_t)\|_2$ for all $t \geq 0$. Since $\{\boldsymbol{y}_t\}_{t=0}^{\infty}$ are arbitrarily chosen, then the random variables $\{Y_t\}_{t=0}^{\infty}$ are independent. Let $Y := \sum_{t=0}^{\infty} Y_t$. Using Chebshev's inequality, we obtain

$$\mathbb{P}\left(Y \geq \mathbb{E}[Y] + \theta\sqrt{\text{Var}[Y]}\right) \leq \mathbb{P}\left(|Y - \mathbb{E}[Y]| \geq \theta\sqrt{\text{Var}[Y]}\right) \leq \frac{1}{\theta^2}. \tag{64}$$

And based on the Assumption 2, there exists an upper bound $\sigma^2 > 0$ for the variance of random noise $e_{\mathcal{B}}(\boldsymbol{x})$ generated from the one-point mini-batch, *i.e.*, $\mathcal{B} = \{i\}, i = 1, \ldots, N$. Consequently, for each $t \geq 0$, we have $\mathbb{E}[Y_t] \leq \frac{\alpha_{K+t}\sigma}{\sqrt{|\mathcal{B}_{K+t}|}}$ and $\text{Var}[Y_t] \leq \frac{\alpha_{K+t}^2 \sigma^2}{|\mathcal{B}_{K+t}|}$, then combining with (64), we have

$$Y \leq \mathbb{E}[Y] + \theta\sqrt{\text{Var}[Y]} \tag{65}$$

$$\leq \sum_{t=0}^{\infty} \frac{\alpha_{K+t}\sigma}{\sqrt{|\mathcal{B}_{k+t}|}} + \theta \cdot \sum_{t=0}^{\infty} \frac{\alpha_{K+t}^2 \sigma^2}{|\mathcal{B}_{K+t}|} \tag{66}$$

$$\leq \sum_{t=0}^{\infty} \frac{\alpha_{K+t}\sigma}{\sqrt{|\mathcal{B}_{k+t}|}} + \theta \cdot \sum_{t=0}^{\infty} \frac{\alpha_{K+t}\sigma}{\sqrt{|\mathcal{B}_{K+t}|}} = (1+\theta)\sum_{t=0}^{\infty} \frac{\alpha_{K+t}\sigma}{\sqrt{|\mathcal{B}_{K+t}|}} \tag{67}$$

holds with probability at least $1 - \frac{1}{\theta^2}$. Here, for the second inequality, we use the property that the equality $\mathbb{E}[\sum_{t=0}^{\infty} Y_i] = \sum_{t=0}^{\infty} \mathbb{E}[Y_i]$ holds whenever $\sum_{t=0}^{\infty} \mathbb{E}[|Y_i|]$ convergences, see Section 2.1 in (63); and for the third inequality, we use $\frac{\alpha_{K+t}\sigma}{\sqrt{|\mathcal{B}_{K+t}|}} \leq 1$ without loss of generality as the common setting of large mini-batch size and small step size.

Given any $\theta > 1$, there exists some $\alpha_k = \mathcal{O}(1/(1+\theta)t)$ and $|\mathcal{B}_k| = \mathcal{O}(t)$, the above series converges and satisfies that

$$(1+\theta)\sum_{t=0}^{\infty} \frac{\alpha_{K+t}\sigma}{\sqrt{|\mathcal{B}_{K+t}|}} \leq \frac{3R^2}{8(4R+1)} \tag{68}$$

holds. Notice that the above proof holds for any given sequence $\{\boldsymbol{y}_t\}_{t=0}^{\infty} \in \mathcal{X}^{\infty}$, thus

$$\max_{\{\boldsymbol{y}_t\}_{t=0}^{\infty} \in \mathcal{X}^{\infty}} \sum_{t=0}^{\infty} \alpha_k \|e_{\mathcal{B}_k}(\boldsymbol{y}_t)\|_2 \leq \frac{3R^2}{8(4R+1)}$$

holds with probability at least $1 - \frac{1}{\theta^2}$. $\qquad \square$

The Lemma 6 draws if previous iterate of Half-Space Step falls into the neighbor of $\boldsymbol{x}^*$, then under appropriate step size and mini-batch setting, the current iterate also inhabits the neighbor with high probability.

**Lemma 6.** *Under the assumptions of Lemma 5, suppose $\|\boldsymbol{x}_K - \boldsymbol{x}^*\| \leq R/2$; for any $\ell$ satisfying $K \leq \ell < K + t$, $0 < \alpha_\ell \leq \min\{\frac{1}{L}, \frac{2\delta_1 - R - \epsilon(2\delta_2 + R)}{M}\}$, $|B_\ell| \geq N - \frac{N}{2M}$ and $\|\boldsymbol{x}_\ell - \boldsymbol{x}^*\| \leq R$ holds, then*

$$\|\boldsymbol{x}_{K+t} - \boldsymbol{x}^*\| \leq R. \tag{69}$$

*holds with probability at least $1 - \frac{1}{\theta^2}$.*

*Proof.* It follows the assumptions of this lemma, Lemma 4, (15) and (16) that for any $\ell$ satisfying $K \leq \ell < K + t$

$$\|[\boldsymbol{x}^*]_g\| = 0, \text{ for any } g \in \hat{\mathcal{G}}_\ell. \tag{70}$$

Hence we have that for $K \leq \ell < K + t$,

$$
\begin{aligned}
&\|\boldsymbol{x}_{\ell+1} - \boldsymbol{x}^*\|^2 \\
&= \sum_{g \in \tilde{\mathcal{G}}_\ell} \|[\boldsymbol{x}_\ell - \boldsymbol{x}^* - \alpha_\ell \nabla\Psi(\boldsymbol{x}_\ell) - \alpha_\ell \boldsymbol{e}_{\mathcal{B}_\ell}(\boldsymbol{x}_\ell)]_g\|^2 + \sum_{g \in \hat{\mathcal{G}}_k} \|[\boldsymbol{x}_\ell - \boldsymbol{x}^* - \boldsymbol{x}_\ell]_g\|^2 \\
&= \sum_{g \in \tilde{\mathcal{G}}_\ell} \left\{ \|[\boldsymbol{x}_\ell - \boldsymbol{x}^*]_g\|^2 - 2\alpha_\ell [\boldsymbol{x}_\ell - \boldsymbol{x}^*]_g^\top [\nabla\Psi(\boldsymbol{x}_\ell) + \boldsymbol{e}_{\mathcal{B}_\ell}(\boldsymbol{x}_\ell)]_g + \alpha_\ell^2 \|[\nabla\Psi(\boldsymbol{x}_\ell) + \boldsymbol{e}_{\mathcal{B}_\ell}(\boldsymbol{x}_\ell)]_g\|^2 \right\} + \sum_{g \in \hat{\mathcal{G}}_\ell} \|[\boldsymbol{x}^*]_g\|^2 \\
&= \sum_{g \in \tilde{\mathcal{G}}_\ell} \left\{ \|[\boldsymbol{x}_\ell - \boldsymbol{x}^*]_g\|^2 - 2\alpha_\ell [\boldsymbol{x}_\ell - \boldsymbol{x}^*]_g^\top [\nabla\Psi(\boldsymbol{x}_\ell)]_g - 2\alpha_\ell [\boldsymbol{x}_\ell - \boldsymbol{x}^*]_g^\top [\boldsymbol{e}_{\mathcal{B}_\ell}(\boldsymbol{x}_\ell)]_g + \alpha_\ell^2 \|[\nabla\Psi(\boldsymbol{x}_\ell) + \boldsymbol{e}_{\mathcal{B}_\ell}(\boldsymbol{x}_\ell)]_g\|^2 \right\} \\
&\leq \sum_{g \in \tilde{\mathcal{G}}_\ell} \|[\boldsymbol{x}_\ell - \boldsymbol{x}^*]_g\|^2 - \|[\nabla\Psi(\boldsymbol{x}_\ell)]_g\|^2 \left( 2\frac{\alpha_\ell}{L} - \alpha_\ell^2 \right) - 2\alpha_\ell [\boldsymbol{x}_\ell - \boldsymbol{x}^*]_g^\top [\boldsymbol{e}_{\mathcal{B}_\ell}(\boldsymbol{x}_\ell)]_g + \alpha_\ell^2 \|[\boldsymbol{e}_{\mathcal{B}_\ell}(\boldsymbol{x}_\ell)]_g\|^2 \\
&\quad + 2\alpha_\ell^2 [\nabla\Psi(\boldsymbol{x}_\ell)]_g^\top [\boldsymbol{e}_{\mathcal{B}_\ell}(\boldsymbol{x}_\ell)]_g \\
&\leq \sum_{g \in \tilde{\mathcal{G}}_\ell} \|[\boldsymbol{x}_\ell - \boldsymbol{x}^*]_g\|^2 - \|[\nabla\Psi(\boldsymbol{x}_\ell)]_g\|^2 \left( 2\frac{\alpha_\ell}{L} - \alpha_\ell^2 \right) + 2\alpha_\ell \|[\boldsymbol{x}_\ell - \boldsymbol{x}^*]_g\| \|[\boldsymbol{e}_{\mathcal{B}_\ell}(\boldsymbol{x}_\ell)]_g\| + \alpha_\ell^2 \|[\boldsymbol{e}_{\mathcal{B}_\ell}(\boldsymbol{x}_\ell)]_g\|^2 \\
&\quad + 2\alpha_\ell^2 \|[\nabla\Psi(\boldsymbol{x}_\ell)]_g\| \|[\boldsymbol{e}_{\mathcal{B}_\ell}(\boldsymbol{x}_\ell)]_g\| \\
&\leq \sum_{g \in \tilde{\mathcal{G}}_\ell} \|[\boldsymbol{x}_\ell - \boldsymbol{x}^*]_g\|^2 - \|[\nabla\Psi(\boldsymbol{x}_\ell)]_g\|^2 \left( 2\frac{\alpha_\ell}{L} - \alpha_\ell^2 \right) + (2\alpha_\ell + 2\alpha_\ell^2 L) \|[\boldsymbol{x}_k - \boldsymbol{x}^*]_g\| \|[\boldsymbol{e}_{\mathcal{B}_\ell}(\boldsymbol{x}_\ell)]_g\| + \alpha_\ell^2 \|[\boldsymbol{e}_{\mathcal{B}_\ell}(\boldsymbol{x}_\ell)]_g\|^2 \\
&\leq \sum_{g \in \tilde{\mathcal{G}}_\ell} \left\{ \|[\boldsymbol{x}_\ell - \boldsymbol{x}^*]_g\|^2 - \|[\nabla\Psi(\boldsymbol{x}_\ell)]_g\|^2 \left( 2\frac{\alpha_\ell}{L} - \alpha_\ell^2 \right) \right\} + (2\alpha_\ell + 2\alpha_\ell^2 L) \|\boldsymbol{x}_\ell - \boldsymbol{x}^*\| \|\boldsymbol{e}_{\mathcal{B}_\ell}(\boldsymbol{x}_\ell)\| + \alpha_\ell^2 \|\boldsymbol{e}_{\mathcal{B}_\ell}(\boldsymbol{x}_\ell)\|^2
\end{aligned}
\tag{71}
$$

On the other hand, by the definition of $\boldsymbol{e}_{\mathcal{B}}(\boldsymbol{x})$ as (63), we have that

$$
\begin{aligned}
\boldsymbol{e}_{\mathcal{B}}(\boldsymbol{x}) &= [\nabla\Psi_{\mathcal{B}}(\boldsymbol{x}) - \nabla\Psi(\boldsymbol{x})]_{\mathcal{I}^{\neq 0}(\boldsymbol{x})} = [\nabla f_{\mathcal{B}}(\boldsymbol{x}) - \nabla f(\boldsymbol{x})]_{\mathcal{I}^{\neq 0}(\boldsymbol{x})} \\
&= \frac{1}{|\mathcal{B}|} \sum_{j \in \mathcal{B}} [\nabla f_j(\boldsymbol{x})]_{\mathcal{I}^{\neq 0}(\boldsymbol{x})} - \frac{1}{N} \sum_{i=1}^{N} [\nabla f_i(\boldsymbol{x})]_{\mathcal{I}^{\neq 0}(\boldsymbol{x})} \\
&= \frac{1}{N} \sum_{j \in \mathcal{B}} \left[ \frac{N}{|\mathcal{B}|} [\nabla f_j(\boldsymbol{x})]_{\mathcal{I}^{\neq 0}(\boldsymbol{x})} - [\nabla f_j(\boldsymbol{x})]_{\mathcal{I}^{\neq 0}(\boldsymbol{x})} \right] - \frac{1}{N} \sum_{\substack{i=1 \\ i \notin \mathcal{B}}}^{N} [\nabla f_i(\boldsymbol{x})]_{\mathcal{I}^{\neq 0}(\boldsymbol{x})} \\
&= \frac{1}{N} \sum_{j \in \mathcal{B}} \left[ \frac{N - |\mathcal{B}|}{|\mathcal{B}|} [\nabla f_j(\boldsymbol{x})]_{\mathcal{I}^{\neq 0}(\boldsymbol{x})} \right] - \frac{1}{N} \sum_{\substack{i=1 \\ i \notin \mathcal{B}}}^{N} [\nabla f_i(\boldsymbol{x})]_{\mathcal{I}^{\neq 0}(\boldsymbol{x})}
\end{aligned}
\tag{72}
$$

Thus taking the norm on both side of (72) and using triangle inequality results in the following:

$$
\begin{aligned}
\|\boldsymbol{e}_{\mathcal{B}}(\boldsymbol{x})\| &\leq \frac{1}{N} \sum_{j \in \mathcal{B}} \left[ \frac{N - |\mathcal{B}|}{|\mathcal{B}|} \|[\nabla f_j(\boldsymbol{x})]_{\mathcal{I}^{\neq 0}(\boldsymbol{x})}\| \right] + \frac{1}{N} \sum_{\substack{i=1 \\ i \notin \mathcal{B}}}^{N} \|[\nabla f_i(\boldsymbol{x})]_{\mathcal{I}^{\neq 0}(\boldsymbol{x})}\| \\
&\leq \frac{1}{N} \frac{N - |\mathcal{B}|}{|\mathcal{B}|} |\mathcal{B}_k| M + \frac{1}{N} (N - |\mathcal{B}|) M \leq \frac{2(N - |\mathcal{B}|)M}{N}.
\end{aligned}
\tag{73}
$$

Since $\alpha_\ell \leq 1$, and $|B_\ell| \geq N - \frac{N}{2M}$ hence $\alpha_\ell \|e_{\mathcal{B}_\ell}(x_\ell)\| \leq 1$. Then combining with $\alpha_\ell \leq 1/L$, (71) can be further simplified as

$$
\begin{aligned}
&\|x_{\ell+1} - x^*\|^2 \\
&\leq \sum_{g \in \tilde{\mathcal{G}}_\ell} \left\{ \|[x_\ell - x^*]_g\|^2 - \|[\nabla\Psi(x_\ell)]_g\|^2 \left(2\frac{\alpha_\ell}{L} - \alpha_\ell^2\right)\right\} + (2\alpha_\ell + 2\alpha_\ell^2 L)\|x_\ell - x^*\|\, \|e_{\mathcal{B}_\ell}(x_\ell)\| + \alpha_\ell^2 \|e_{\mathcal{B}_\ell}(x_\ell)\|^2 \\
&\leq \sum_{g \in \tilde{\mathcal{G}}_\ell} \left\{ \|[x_\ell - x^*]_g\|^2 - \frac{1}{L^2}\|[\nabla\Psi(x_\ell)]_g\|^2 \right\} + 4\alpha_\ell \|x_\ell - x^*\|\, \|e_{\mathcal{B}_\ell}(x_\ell)\| + \alpha_\ell^2 \|e_{\mathcal{B}_\ell}(x_\ell)\|^2 \\
&\leq \|x_\ell - x^*\|^2 + 4\alpha_\ell \|x_\ell - x^*\|\, \|e_{\mathcal{B}_\ell}(x_\ell)\| + \alpha_\ell \|e_{\mathcal{B}_\ell}(x_\ell)\|
\end{aligned}
\tag{74}
$$

Following from the assumption that $\|x_\ell - x^*\| \leq R$, then (74) can be further simplified as

$$
\begin{aligned}
\|x_{\ell+1} - x^*\|^2 &\leq \|x_\ell - x^*\|^2 + 4\alpha_\ell R \|e_{\mathcal{B}_\ell}(x_\ell)\| + \alpha_k \|e_{\mathcal{B}_\ell}(x_\ell)\| \\
&\leq \|x_\ell - x^*\|^2 + (4R+1)\alpha_\ell \|e_{\mathcal{B}_\ell}(x_\ell)\|
\end{aligned}
\tag{75}
$$

Summing the the both side of (75) from $\ell = K$ to $\ell = K + t - 1$ results in

$$
\|x_{K+t} - x^*\|^2 \leq \|x_K - x^*\|^2 + (4R+1)\sum_{\ell=K}^{K+t-1} \alpha_\ell \|e_{\mathcal{B}_\ell}(x_\ell)\|
\tag{76}
$$

It follows Lemma 5 that the followng holds with probability at least $1 - \frac{1}{\theta^2}$,

$$
\sum_{\ell=K}^{\infty} \alpha_\ell \|e_{\mathcal{B}_\ell}(x_\ell)\| \leq \frac{3R^2}{4(4R+1)}.
\tag{77}
$$

Thus we have that

$$
\begin{aligned}
\|x_{K+t} - x^*\|^2 &\leq \|x_K - x^*\|^2 + (4R+1)\sum_{\ell=K}^{K+t-1} \alpha_\ell \|e_{\mathcal{B}_\ell}(x_\ell)\| \\
&\leq \|x_K - x^*\|^2 + (4R+1)\sum_{\ell=K}^{\infty} \alpha_\ell \|e_{\mathcal{B}_\ell}(x_\ell)\| \\
&\leq \frac{R^2}{4} + (4R+1)\frac{3R^2}{4(4R+1)} \leq \frac{R^2}{4} + \frac{3R^2}{4} \leq R^2,
\end{aligned}
\tag{78}
$$

holds with probability at least $1 - \frac{1}{\theta^2}$, which completes the proof. $\square$

Based on the above lemmas, the Lemma 7 shows if initial iterate of Half-Space Step locates closely enough to $x^*$, step size $\alpha_k$ polynomially decreases, and mini-batch size $\mathcal{B}_k$ polynomially increases, then $x^*$ inhabits all subsequent reduced space $\{S_k\}_{k=K}^{\infty}$ constructed in Half-Space Step with high probability.

**Lemma 7.** *If $\|x_K - x^*\| \leq \frac{R}{2}$, $K \geq N_\mathcal{P}$, $k = K + t$, $t \in \mathbb{Z}^+$, $0 < \alpha_k = \mathcal{O}(1/(\sqrt{N}t)) \leq \min\{\frac{2(1-\epsilon)}{L}, \frac{1}{L}, \frac{2\delta_1 - R - \epsilon(2\delta_2 + R)}{M}\}$ and $|\mathcal{B}_k| = \mathcal{O}(t) \geq N - \frac{N}{2M}$. Then for any constant $\tau \in (0,1)$, $\|x_k - x^*\| \leq R$ with probability at least $1 - \tau$ for any $k \geq K$.*

*Proof.* It follows Lemma 3 and the assumption of this lemma that $x^* \in S_K$. Moreover, it follows the assumptions of Lemma (5, 6, 7), the definition of finite-sum $f(x)$ in (12), and the bound of error as (73) that

$$
\mathbb{P}(\{x_k\}_{k=K}^{\infty} \in \{x : \|x - x^*\| \leq R\}^{\infty}) \geq \left(1 - \frac{1}{\theta^2}\right)^{\mathcal{O}(N-K)} \geq 1 - \tau,
\tag{79}
$$

where the last two inequalities comes from that the error vanishing to zero as $|\mathcal{B}_k|$ reaches the upper bound $N$, and $\theta$ is sufficiently large depending on $\tau$ and $\mathcal{O}(N-K)$. $\square$

**Corollary 2.** *Lemma 7 further implies $x^*$ inhabits all subsequent $S_k$, i.e., $x^* \in S_k$ for any $k \geq K$.*

## B.5 The Initialization Stage

In previous parts, we show that the Half-Space Step guarantees to converge to the optimal solution, and ensures to recover the no-zero groups of the optimal solution under some assumptions with a "close-enough" initialization point $\boldsymbol{x}_{N_{\mathcal{P}}}$. To complete the story, in this part, we show that the iterate obtained from the *Subgradient Descent Update* in Algorithm 2 satisfies the "close-enough" condition with high probability. Remark here that the proximal methods, such as Prox-SG, Prox-SVRG and SAGA, may also serve in the initialization stage. However, for the general regularization $r(\boldsymbol{x})$, they may not have closed-form solution for the corresponding inherent subproblems, implying non-explicit update mechanism to the next iterate. Hence, people may have to inconveniently approximate the solutions of proximal operator by other techniques, whereas the sub-gradient method does not have these drawbacks. Therefore, for the generality of HSPG, we select the sub-gradient method in the Initialization Stage by default.

### B.5.1 Convergence Analysis of Initialization Stage

In this part, we show that the "close enough" condition

$$\|\boldsymbol{x}_k - \boldsymbol{x}^*\| \leq \frac{R}{2} \tag{80}$$

proposed in Theorem 1 can be achieved via the Initialization Stage (*Subgradient Descent Update*) in Algorithm 2 under the Assumption 5.

**Assumption 5.** *Assume the following assumptions hold.*

- **(A5-1).** $f : \mathbb{R}^n \mapsto \mathbb{R}$ *is differentiable and $\mu$-strongly convex.* $r : \mathbb{R}^n \mapsto \mathbb{R}$ *is convex.*

- **(A5-2).** *There exists an universal constant $M$ such that the stochastic gradient $\nabla f_{\mathcal{B}}(\boldsymbol{x})$ satisfies $\|\nabla f_{\mathcal{B}}(\boldsymbol{x})\|_2 \leq M$ for all $\boldsymbol{x} \in \mathbb{R}^d$ and mini-batch $\mathcal{B}$.*

- **(A5-3).** *The stochastic gradient $\nabla f_{\mathcal{B}}(\boldsymbol{x})$ satisfies $\mathbb{E}_{\mathcal{B}}[\nabla f_{\mathcal{B}}(\boldsymbol{x})|\boldsymbol{x}] = \nabla f(\boldsymbol{x})$ for all $\boldsymbol{x} \in \mathbb{R}^n$.*

**Proposition 2.** *Under Assumption 5, for any $R > 0$, any $\tau \in (0, 1)$, set*

$$N = \left\lceil \log\left(\frac{\tau R}{4\|\boldsymbol{x}_0 - \boldsymbol{x}^*\|_2^2}\right) \bigg/ \log\left(1 - \frac{\tau R}{4M}\right) \right\rceil, \tag{81}$$

$$\alpha_0 = \alpha_1 = \ldots = \alpha_{N_{\mathcal{P}}-1} = \frac{\tau \mu R}{4M^2}, \tag{82}$$

*where $R$ based on the setting of Theorem 1. We have the Algorithm 1 (Subgradient Descent Update) returns a solution $\boldsymbol{x}_{N_{\mathcal{P}}}$ that satisfies $\|\boldsymbol{x}_{N_{\mathcal{P}}} - \boldsymbol{x}^*\|_2 \leq R/2$ with probability $1 - \tau$.*

*Proof.* Let $\boldsymbol{x}^*$ be the global optimal solution of (2). Let $\nabla \psi(\boldsymbol{x}) = \nabla f(\boldsymbol{x}) + \lambda \zeta(\boldsymbol{x})$ and $\nabla \psi_{\mathcal{B}}(\boldsymbol{x}) = \nabla f_{\mathcal{B}}(\boldsymbol{x}) + \lambda \zeta(\boldsymbol{x})$ given any point $\boldsymbol{x} \in \mathbb{R}^n$ and mini-batch $\mathcal{B}$. Consider

$$\|\boldsymbol{x}_{k+1} - \boldsymbol{x}^*\|_2^2 = \|\boldsymbol{x}_k - \alpha_k \nabla \psi_{\mathcal{B}_k}(\boldsymbol{x}_k) - \boldsymbol{x}^*\|_2^2 \tag{83}$$

$$= \|\boldsymbol{x}_k - \boldsymbol{x}^*\|_2^2 - 2\alpha_k \langle \nabla \psi_{\mathcal{B}_k}(\boldsymbol{x}_k), \boldsymbol{x}_k - \boldsymbol{x}^* \rangle + \|\alpha_k \nabla \psi_{\mathcal{B}_k}(\boldsymbol{x}_k)\|_2^2. \tag{84}$$

Due to (A1) in Assumption 5, the $\mu$-strongly convexity of $f$ and the convexity of $r$ yields

$$\psi(\boldsymbol{x}^*) \geq \psi(\boldsymbol{x}_k) + \langle \nabla \psi(\boldsymbol{x}_k), \boldsymbol{x}^* - \boldsymbol{x}_k \rangle + \frac{\mu}{2}\|\boldsymbol{x}_k - \boldsymbol{x}^*\|_2^2. \tag{85}$$

Thus

$$\|\boldsymbol{x}_{k+1} - \boldsymbol{x}^*\|_2^2 \tag{86}$$

$$= \|\boldsymbol{x}_k - \boldsymbol{x}^*\|_2^2 - 2\alpha_k\langle\nabla\psi_{\mathcal{B}_k}(\boldsymbol{x}_k), \boldsymbol{x}_k - \boldsymbol{x}^*\rangle + \|\alpha_k\nabla\psi_{\mathcal{B}_k}(\boldsymbol{x}_k)\|_2^2 \tag{87}$$

$$= \|\boldsymbol{x}_k - \boldsymbol{x}^*\|_2^2 + 2\alpha_k\langle\nabla\psi_{\mathcal{B}_k}(\boldsymbol{x}_k), \boldsymbol{x}^* - \boldsymbol{x}_k\rangle + \|\alpha_k\nabla\psi_{\mathcal{B}_k}(\boldsymbol{x}_k)\|_2^2 \tag{88}$$

$$= \|\boldsymbol{x}_k - \boldsymbol{x}^*\|_2^2 + 2\alpha_k\langle\nabla\psi(\boldsymbol{x}_k) - \nabla\psi(\boldsymbol{x}_k) + \nabla\psi_{\mathcal{B}_k}(\boldsymbol{x}_k), \boldsymbol{x}^* - \boldsymbol{x}_k\rangle + \|\alpha_k\nabla\psi_{\mathcal{B}_k}(\boldsymbol{x}_k)\|_2^2 \tag{89}$$

$$\leq \|\boldsymbol{x}_k - \boldsymbol{x}^*\|_2^2 + 2\alpha_k\left(\psi(\boldsymbol{x}^*) - \psi(\boldsymbol{x}_k) - \frac{\mu}{2}\|\boldsymbol{x}_k - \boldsymbol{x}^*\|_2^2\right) \tag{90}$$

$$+ 2\alpha_k\langle\nabla\psi_{\mathcal{B}_k}(\boldsymbol{x}_k) - \nabla\psi(\boldsymbol{x}_k), \boldsymbol{x}^* - \boldsymbol{x}_k\rangle + \|\alpha_k\nabla\psi_{\mathcal{B}_k}(\boldsymbol{x}_k)\|_2^2 \tag{91}$$

$$\leq (1 - \alpha_k\mu)\|\boldsymbol{x}_k - \boldsymbol{x}^*\|_2^2 - 2\alpha_k(\psi(\boldsymbol{x}_k) - \psi(\boldsymbol{x}^*)) + \alpha_k^2\|\nabla\psi(\boldsymbol{x}_k)\|_2^2 \tag{92}$$

$$+ 2\alpha_k\langle\nabla\psi_{\mathcal{B}_k}(\boldsymbol{x}_k) - \nabla\psi(\boldsymbol{x}_k), \boldsymbol{x}^* - \boldsymbol{x}_k\rangle \tag{93}$$

$$\leq (1 - \alpha_k\mu)\|\boldsymbol{x}_k - \boldsymbol{x}^*\|_2^2 + \alpha_k^2 M^2 + 2\alpha_k\langle\nabla\psi_{\mathcal{B}_k}(\boldsymbol{x}_k) - \nabla\psi(\boldsymbol{x}_k), \boldsymbol{x}^* - \boldsymbol{x}_k\rangle. \tag{94}$$

Given $\boldsymbol{x}_k$, due to (A5-2) in Assumption 5, taking expectation over $\mathcal{B}_k$ yields

$$\mathbb{E}_{\mathcal{B}_k}[\|\boldsymbol{x}_{k+1} - \boldsymbol{x}^*\|_2^2 | \boldsymbol{x}_k] \leq (1 - \alpha_k\mu)\|\boldsymbol{x}_k - \boldsymbol{x}^*\|_2^2 + \alpha_k^2 M^2, \tag{95}$$

where the above inequality holds by (A5-3) in Assumption 5

$$\mathbb{E}_{\mathcal{B}_k}[\langle\nabla\psi_{\mathcal{B}_k}(\boldsymbol{x}_k) - \nabla\psi(\boldsymbol{x}_k), \boldsymbol{x}^* - \boldsymbol{x}_k\rangle | \boldsymbol{x}_k] = 0. \tag{96}$$

For any $k \in \mathbb{N}_+$, any constant $c > 0$, and initial point $\boldsymbol{x}_0$, setting $\alpha_k = \frac{\mu}{cM^2}$, apply above inequality recursively yields

$$\mathbb{E}_{\mathcal{H}}\left[\|\boldsymbol{x}_k - \boldsymbol{x}^*\|_2^2\right] \leq \left(1 - \frac{1}{cM^2}\right)^k \|\boldsymbol{x}_0 - \boldsymbol{x}^*\|_2^2 + \frac{1}{c}, \tag{97}$$

where $\mathcal{H} = \{\mathcal{B}_0, \ldots, \mathcal{B}_{k-1}\}$ denotes the whole history until step $k$.

**Non-asymptotic bounds.** Combine above together, given any $R/2 > 0$, for any $\tau \in (0, 1)$, set

$$N = \left\lceil \log\left(\frac{\tau R}{4\|\boldsymbol{x}_0 - \boldsymbol{x}^*\|_2^2}\right) \Big/ \log\left(1 - \frac{\tau R}{4M}\right) \right\rceil, \tag{98}$$

$$\alpha_0 = \alpha_1 = \ldots = \alpha_{N_\mathcal{P}-1} = \frac{\tau\mu R}{4M^2}, \tag{99}$$

by Markov inequality, we have

$$\|\boldsymbol{x}_k - \boldsymbol{x}^*\|_2 \leq R/2 \tag{100}$$

holds with probability $1 - \tau$. □

## C   Extensive Numerical Experiments

In this Appendix, we include extensive numerical experiments in the view of optimization to demonstrate the superiority of HSPG to other classical proximal methods on the sparsity exploration and the competitiveness on objective convergence in both convex and nonconvex settings. Particularly, in Appendix C.1, we provide convex experiments to *(i)* demonstrate the validness of group sparsity identification of HSPG; *(ii)* present comprehensive comparison to Prox-SG, RDA and Prox-SVRG on benchmark convex problems. In Appendix C.2, we show additional nonconvex experiments to reveal the superiority of HSPG to competitors on group sparsity exploration.

### C.1   Convex Experiments

**Linear Regression on Synthetic Data**   We numerically validate the proposed HSPG on group sparsity identification by linear regression problems with $\ell_1/\ell_2$ regularizations using synthetic data. Consider a data matrix $A \in \mathbb{R}^{N \times n}$ consisting of $N$ instances and the target variable $\boldsymbol{y} \in \mathbb{R}^N$, we are interested in the following problem:

$$\underset{\boldsymbol{x} \in \mathbb{R}^n}{\text{minimize}} \; \frac{1}{2N}\|A\boldsymbol{x} - \boldsymbol{y}\|^2 + \lambda\sum_{g \in \mathcal{G}}\|[\boldsymbol{x}]_g\|. \tag{101}$$

Our goal is to empirically show that HSPG is able to identify the ground truth zero groups with synthetic data. We conduct the experiments as follows: *(i)* generate the data matrix $A$ whose elements are uniformly distributed among $[-1, 1]$; *(ii)* generate a vector $\boldsymbol{x}^*$ working as the ground truth solution, where the elements are uniformly distributed among $[-1, 1]$ and the coordinates are equally divided into 10 groups ($|\mathcal{G}| = 10$); *(iii)* randomly set a number of groups of $\boldsymbol{x}^*$ to be 0 according to a pre-specified group sparsity ratio; *(iv)* compute the target variable $\boldsymbol{y} = A\boldsymbol{x}^*$; (v) solve the above problem (101) for $\boldsymbol{x}$ with $A$ and $\boldsymbol{y}$ only, and then evaluate the Intersection over Union (IoU) with respect to the identities of the zero groups between the computed solution estimate $\hat{\boldsymbol{x}}$ by HSPG and the ground truth $\boldsymbol{x}^*$.

We test HSPG on (101) under different problem settings. For a slim matrix $A$ where $N \geq n$, we test with various group sparsity ratios among $\{0.1, 0.3, 0.5, 0.7, 0.9\}$, and for a fat matrix $A$ where $N < n$, we only test with a certain group sparsity value since a recovery of $\boldsymbol{x}^*$ requires that the number of non-zero elements in $\boldsymbol{x}^*$ is bounded by $N$. Throughout the experiments, we set $\lambda$ to be $100/N$, the mini-batch size $|\mathcal{B}|$ to be 64, step size $\alpha_k$ to be 0.1 (constant), and fine-tune $\epsilon$ per problem. Based on a similar statistical test on objective function stationarity (98), we switch to Half-Space Step roughly after 30 epoches. Table 8 shows that under each setting, the proposed HSPG correctly identifies the groups of zeros as indicated by $\text{IoU}(\hat{\boldsymbol{x}}, \boldsymbol{x}^*) = 1.0$, which is a strong evidence to show the correctness of group sparsity identification of HSPG.

Table 8: Linear regression problem settings and IoU of the recovered solutions by HSPG.

| | $N$ | $n$ | Group sparsity ratio of $\boldsymbol{x}^*$ | $\text{IoU}(\hat{x}, x^*)$ |
|---|---|---|---|---|
| | 10000 | 1000 | {0.1, 0.3, 0.5, 0.7, 0.9} | 1.0 |
| Slim $A$ | 10000 | 2000 | {0.1, 0.3, 0.5, 0.7, 0.9} | 1.0 |
| | 10000 | 3000 | {0.1, 0.3, 0.5, 0.7, 0.9} | 1.0 |
| | 10000 | 4000 | {0.1, 0.3, 0.5, 0.7, 0.9} | 1.0 |
| | 200 | 1000 | 0.9 | 1.0 |
| Fat $A$ | 300 | 1000 | 0.8 | 1.0 |
| | 400 | 1000 | 0.7 | 1.0 |
| | 500 | 1000 | 0.6 | 1.0 |

**Logistic Regression** We then focus on the benchmark convex logistic regression problem with the mixed $\ell_1/\ell_2$-regularization given $N$ examples $(\boldsymbol{d}_1, l_1), \cdots, (\boldsymbol{d}_N, l_N)$ where $\boldsymbol{d}_i \in \mathbb{R}^n$ and $l_i \in \{-1, 1\}$ with the form

$$\underset{(\boldsymbol{x};b) \in \mathbb{R}^{n+1}}{\text{minimize}} \frac{1}{N} \sum_{i=1}^{N} \log(1 + e^{-l_i(\boldsymbol{x}^T \boldsymbol{d}_i + b)}) + \lambda \sum_{g \in \mathcal{G}} \|[\boldsymbol{x}]_g\|, \tag{102}$$

for binary classification with a bias $b \in \mathbb{R}$. We set the regularization parameter $\lambda$ as $100/N$ throughout the experiments since it yields high sparse solutions and low object value $f$'s, equally decompose the variables into 10 groups to form $\mathcal{G}$, and test problem (102) on 8 standard publicly available large-scale datasets from LIBSVM repository (6) as summarized in Table 9. All convex experiments are conducted on a 64-bit operating system with an Intel(R) Core(TM) i7-7700K CPU @ 4.20 GHz and 32 GB random-access memory.

We run the solvers with a maximum number of epochs as 60 following (8). The mini-batch size $|\mathcal{B}|$ is set to be $\min\{256, \lceil 0.01N \rceil\}$ similarly to (93). The step size $\alpha_k$ setting follows [Section 4](88). Particularly, we first compute a Lipschitz constant $L$ as $\max_i \|\boldsymbol{d}_i\|^2 / 4$, then fine tune and select constant $\alpha_k \equiv \alpha = 1/L$ to Prox-SG and Prox-SVRG since it exhibits the best results. For RDA, the step size parameter $\gamma$ is fined tuned as the one with the best performance among all powers of 10. For HSPG, we set $\alpha_k$ as the same as Prox-SG and Prox-SVRG in practice. We select two $\epsilon$'s as 0 and 0.8. The final objective value $\psi$ and group sparsity in the solutions are reported in Table 10-11, where we mark the best values as bold to facilitate the comparison. Furthermore, Figure 7 plots the relative runtime of these solvers for each dataset, scaled by the runtime of the most time-consuming solver.

Table 11 shows that our HSPG is definitely the best solver on exploring the group sparsity of the solutions. In fact, HSPG under $\epsilon = 0.8$ performs all the best except *ijcnn1*. Prox-SVRG is the second best solver on group sparsity exploration, which demonstrates that the variance reduction techniques works well in convex setting to promote sparsity, but not in non-convex settings. HSPG

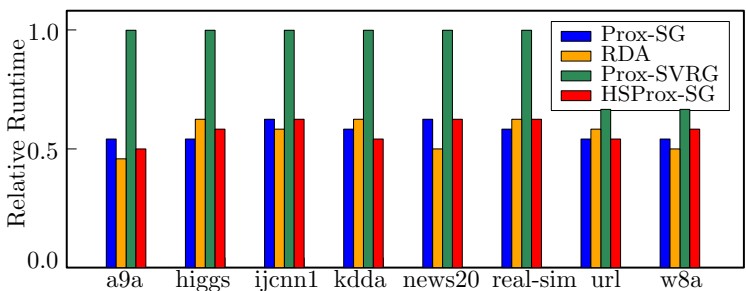

Figure 7: Relative runtime.

under $\epsilon = 0$ performs much better than Prox-SG which matches the better sparsity recovery property of HSPG even under $\epsilon$ as 0. Moreover, as shown in Table 10, we observe that all solvers perform quite competitively in terms of final objective values (round up to 3 decimals) except RDA, which demonstrates that HSPG reaches comparable convergence as Prox-SG and Prox-SVRG in practice. Finally, Figure 7 indicates that Prox-SG, RDA and HSPG have similar computational cost to proceed, except Prox-SVRG due to its periodical full gradient computation.

Table 9: Summary of datasets.

| Dataset | N | n | Attribute | Dataset | N | n | Attribute |
|---|---|---|---|---|---|---|---|
| a9a | 32561 | 123 | binary $\{0, 1\}$ | news20 | 19996 | 1355191 | unit-length |
| higgs | 11000000 | 28 | real $[-3, 41]$ | real-sim | 72309 | 20958 | real $[0, 1]$ |
| ijcnn1 | 49990 | 22 | real [-1, 1] | url_combined | 2396130 | 3231961 | real $[-4, 9]$ |
| kdda | 8407752 | 20216830 | real $[-1, 4]$ | w8a | 49749 | 300 | binary $\{0, 1\}$ |

Table 10: Final objective values $\psi$ for tested algorithms on convex problems.

| Dataset | Prox-SG | RDA | Prox-SVRG | HSPG | |
|---|---|---|---|---|---|
| | | | | $\epsilon$ as 0 | $\epsilon$ as 0.8 |
| a9a | **0.355** | 0.359 | **0.355** | **0.355** | **0.355** |
| higgs | **0.357** | 0.360 | 0.365 | 0.358 | 0.358 |
| ijcnn1 | **0.248** | 0.278 | **0.248** | **0.248** | **0.248** |
| kdda | **0.103** | 0.124 | **0.103** | **0.103** | **0.103** |
| news20 | **0.538** | 0.693 | **0.538** | **0.538** | **0.538** |
| real-sim | **0.242** | 0.666 | 0.244 | **0.242** | **0.242** |
| url_combined | 0.397 | 0.579 | **0.391** | 0.405 | 0.405 |
| w8a | **0.110** | 0.111 | 0.112 | **0.110** | **0.110** |

Table 11: Group sparsity for tested algorithms on convex problems.

| Dataset | Prox-SG | RDA | Prox-SVRG | HSPG | |
|---|---|---|---|---|---|
| | | | | $\epsilon$ as 0 | $\epsilon$ as 0.8 |
| a9a | 20% | **30%** | **30%** | **30%** | **30%** |
| higgs | 0% | 10% | 0% | 0% | **30%** |
| ijcnn1 | 50% | **70%** | 60% | 60% | 60% |
| kdda | 0% | 0% | 0% | 0% | **80%** |
| news20 | 20% | 80% | **90%** | 80% | **90%** |
| real-sim | 0% | 0% | **80%** | 0% | **80%** |
| url_combined | 0% | 0% | 0% | 0% | **90%** |
| w8a | **0%** | **0%** | **0%** | **0%** | **0%** |

## C.2 Nonconvex Experiments

To illustrate, among the state-of-the-art proximal stochastic optimizers, we exclude RDA because of no acceptable results attained during our following tests with the step size parameter $\gamma$ setting throughout all powers of 10 from $10^{-3}$ to $10^3$, and skip Prox-Spider and SAGA since Prox-SVRG has been a superb representative to the proximal incremental gradient methods. We consider the popular image classification tasks, with popular architectures, *i.e.*, VGG16 and ResNet18 on benchmark datasets CIFAR10 and Fashion-MNIST (86), where the group partition $\mathcal{G}$ is defined as 3D kernel following (14; 58), which are not ZIGs.

Table 12: Final $\psi$/group sparsity ratio/testing accuracy on non-convex problems over non-ZIGs.

| Backbone | Dataset | Prox-SG | Prox-SVRG | HSPG |
|---|---|---|---|---|
| VGG16 | CIFAR10 | 0.59 / 52.58% / 90.50% | 0.85 / 14.13% / 89.16% | **0.58** / **76.47%** / **91.93%** |
| | Fashion-MNIST | 0.52 / 12.31% / **92.83%** | 2.66 / 0.38% / 92.72% | **0.52** / **47.82%** / 92.87% |
| ResNet18 | CIFAR10 | **0.31** / 20.27% / 94.36% | 0.37 / 4.60% / 94.11% | **0.31** / **69.98%** / **94.40%** |
| | Fashion-MNIST | 0.14 / 0.00% / **94.94%** | 0.18 / 0.00% / 94.70% | **0.13** / **77.08%** / 94.61% |
| MobileNetV1 | CIFAR10 | **0.40** / 58.05% / 91.54% | 0.65 / 29.20% / 89.68% | **0.40** / **71.36%** / **92.04%** |
| | Fashion-MNIST | **0.22** / 62.62% / 94.22% | 0.40 / 41.99% / 94.19% | 0.26 / **84.26%** / **94.52%** |

Table 12 demonstrates the effectiveness and superiority of HSPG, where we mark the best values as bold, and the group sparsity ratio is defined as the percentage of zero groups. In particular, *(i)* HSPG computes remarkably higher group sparsity than other methods on all tests, of which the solutions are typically multiple times sparser in the manner of group than those of Prox-SG, while Prox-SVRG performs not comparably since the variance reduction techniques may not work as desired for deep learning applications (13); *(ii)* HSPG performs competitively with respect to the final objective values $\psi$. In addition, all the methods reach a comparable generalization performance on unseen test data. On the other hand, sparse regularization methods may yield solutions with entries that are not exactly zero but are very small. Sometimes all entries below certain threshold ($\mathcal{T}$) are set to zero (44; 20). However, such simple truncation mechanism is heuristic-rule based, hence may hurt convergence and accuracy. To illustrate this, we set the groups of the solutions of Prox-SG and Prox-SVRG to zero if the magnitudes of the group variables are less than some $\mathcal{T}$, and denote the corresponding solutions as Prox-SG* and Prox-SVRG*.

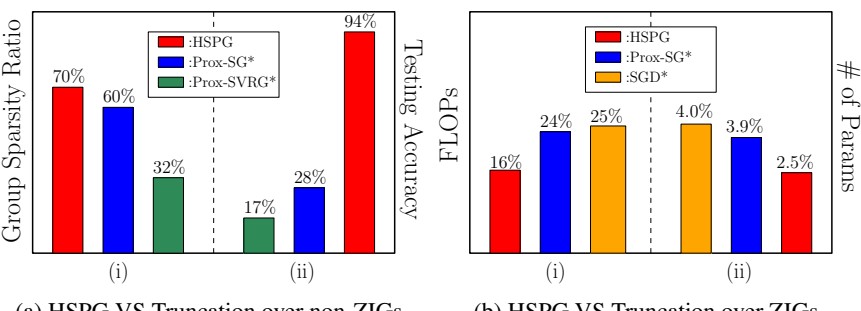

(a) HSPG VS Truncation over non-ZIGs.          (b) HSPG VS Truncation over ZIGs.

Figure 8: HSPG versus simple truncation. (a) On ResNet18 with CIFAR10 over non-ZIGs. (b) On VGG16 with CIFAR10 over ZIGs.

As shown in Figure 8a(i), under the $\mathcal{T}$ with no accuracy regression, Prox-SG* and Prox-SVRG* reach higher group sparsity ratio as 60% and 32% compared to Table 12, but still significantly lower than the 70% of HSPG without simple truncation. Under the $\mathcal{T}$ to reach the same group sparsity ratio as HSPG, the testing accuracy of Prox-SG* and Prox-SVRG* regresses drastically to 28% and 17% in Figure 8a(ii) respectively. Remark here that although further refitting the models from Prox-SG* and Prox-SVRG* on active (non-zero) groups of weights may recover the accuracy regression, it requires additional engineering efforts and training cost, which is less attractive and convenient than HSPG (with no need to refit). Similarly, as shown in Figure 8b, under the ZIG partition and the $\mathcal{T}$ without accuracy regression, the FLOPs and number of parameters reductions achieved by SGD* (subgradient descent with simple truncation) and Prox-SG* are not comparable with those achieve by HSPG, *i.e.*, HSPG achieves about $1.5\times$ fewer FLOPs and number of parameters.