# OpenReview forum: "Only Train Once: A One-Shot Neural Network Training And Pruning Framework"
_NeurIPS.cc/2021/Conference — NeurIPS 2021 Poster_

### Official Review · Reviewer_Echy · 2021-07-14

**Rating:** 7
**Confidence:** 4

**Summary:**

This paper proposes a structured pruning method called Only-Train-Once (OTO). OTO first partitions the trainable parameters into zero-invariant groups called ZIGs. This way allows one to prune without affecting the output and therefore it requires no additional training steps after pruning. Once they define ZIGs, they perform Half-space Stochastic Projected Gradient (HSPG), starting from a pre-trained estimate using SGD, it performs a form of projected gradient descent while gradually increasing group sparsity. The authors demonstrate the effectiveness of OTO on CIFAR, ImageNet, SQuAD by showing an impressive gain in terms of FLOPs while maintaining the performance of resulting slim network to the original dense counterparts.


**Limitations And Societal Impact:**

- Can the process of creating ZIGs be made automatic? I feel there should be a generalized way to perform the process. Discussing this may make the manuscript stronger.

- The idea of finishing the pruning step in one shot (and training only once) seems to be shared with other “single-shot” pruning methods either before such as SNIP by Lee et al. 2019, GraSP Wang et al. 2020, or any potential pruning method relying on a saliency criterion. While OTO is based on an optimization method with some convergence guarantee (I haven’t checked this in detail; is it applied for non-convex objectives?) and the above methods seem to rely on heuristics, crediting the former for algorithmic simplicity (L94: “OTO is simple”) over the others seems a bit stretched, considering that OTO requires additional step of creating ZIGs, performing a projected gradient descent, especially without any analysis on the computational cost of HSPG in comparison to others.

- Related to this, the paper defines pruning approaches to be those that mostly require fine-tuning (or re-training) and claims their contribution for removing it, and yet, this doesn’t seem to be a fair scope setting. Any sparsity inducing regularization based method (including those cited in this work such as [50]) can theoretically remove fine-tuning step.

- Discussing these properly in the paper seems necessary. Having said that, the title as well as the name of the method are not quite representative of the main idea.

- Any idea how to perform HSPG without ZIGs? Would it basically reduce to [58]?

- How do you control the level of sparsity (set hyperparameters in general) found by HSPG?

- Any analysis on the cost of HSPG and its comparison to other methods?


**Main Review:**

This work is motivated to remove the subsequent fine-tuning process after pruning (i.e., making some parameters zero-valued) that is often performed for various pruning approaches, in particular those in a similar fashion such as Group Lasso or any group sparsity inducing regularizer based methods. The idea of first creating zero invariant groups is interesting and new in the context of pruning as far as I know, and designing these groups for major components in modern neural networks is done in an easy and nice way. This can actually be very pragmatic in practice. Their actual pruning and training algorithm, HPGS, differs from rather aggressive filter pruning methods that work based on heuristics and lack theoretical guarantees on convergence, and combined with ZIGs, it seems that the idea is very effective as supported by experimental evidence. Overall, the paper is written with clarity and organized well.


**Time Spent Reviewing:**

6

---

> ### Author Response · Authors · 2021-08-10
> **Author response for Reviewer Echy**
>
> Thank you for the appreciation of our work and the in-depth reviews and suggestions. We have prepared our responses to each of your point.
>
> - **Q1: Can the process of creating ZIGs be made automatic? I feel there should be a generalized way to perform the process. Discussing this may make the manuscript stronger.**
>
> 	A1: This is a very insightful question. Yes, the process of constructing ZIGs can be definitely made automatically. Indeed, after submitting the manuscript, we have begun to study the automatic ZIG construction given an arbitrary DNN architecture and made promising progress.
>
> 	The key is to figure out the connections between the layers in order to properly tag the type of each layers, i.e., Conv-BN, Residual Connection, Multi-head attention, Fully Connected and so on, then constructing ZIGs in the ways described in Section 3.1. Therefore, the ultimate OTO framework is able to train and compress an arbitrary DNN from scratch into a slimmer sub-DNN with high performance by only-train-once. We will discuss the automatic process in the revision and publish the code into the official repository after the review process.
>
> - **Q2: Is HSPG applied for non-convex objectives?**
>
> 	A2: Yes, the proposed HSPG is applicable for non-convex objectives. In Appendix B.2, we provide the assumptions used throughout the convergence analysis and do not assume the convexity of the loss function $f$. The regualizer needs to be convex, which is naturally satisfied by the definition of the popular group-sparsity inducing regularization, such as the popular mixed $\ell_1/\ell_p$  norm $(p\geq 1)$ and group minmax concave penalty.
>
> - **Q3: Although the author claims that “OTO is simple” on L94, it seems not considering that  OTO requires additional step of creating ZIGs, performing a projected gradient descent, especially without any analysis on the computational cost of HSPG in comparison to others.**
>
> 	A3: This is an insightful comment. We clarify that the L94, ``OTO is simple" is more referred from the user perspective. Since once the ZIGs can be automatically generated, the engineering requirements to get a slimmer pruned model from the user side is reduced dramatically. On the other hand, the additional operations in HSPG are not expensive and can be completed very efficiently by the processors. To support it, in Figure 7 of Appendix C, we provided the relative runtime of different optimization algorithms for solving the same group-sparsity problem, where HSPG is competitive to other gradient-based methods in terms of the computational cost. We will incorporate the above discussion into the revision.
>
> - **Q4:  Since SNIP and GraSP also do not require fine-tuning before OTO, claim removing fine-tuning as contribution is not proper.  Discussing them properly is necessary.**
>
> 	A4: This is a great suggestion. To address it, we will discuss these related one-shot pruning methods in the revision, adjust corresponding statement, and make the title to be more representative to ZIG and HSPG.
>
> - **Q5: Any idea how to perform HSPG without ZIGs? Would it basically reduce to [NO2006Springer]?**
>
> 	A5: Our proposed HSPG is a novel optimization algorithm for general group-sparsity regularization problem, hence it is applicable to non-ZIGs as well. In Appendix C, we tested HSPG on both convex and nonconvex problems where most of these experiments were defined over non-ZIGs, and compared it with other SOTA optimization algorithms. The results on non-ZIGs are consistent with the results on ZIGs, i.e., the HSPG can achieve higher group sparsity and maintain competitive convergence property.
>
>     Furthermore, there is no similar projected optimization algorithm in [NO2006Springer] to HSPG, since the proposed half-space projection operator in HSPG is a non-Elucidean projection operator (L215 in the main body) which is not covered by [NO2006Springer].
>
> 	**Reference**
>
> 	[NO2006Springer] Jorge Nocedal and Stephen Wright. Numerical optimization. Springer Science & Business Media. 2006.
>
> - **Q6: How do you control the level of sparsity (set hyperparameters in general) found by HSPG?**
>
> 	A6: This is a great question. In general, we have two hyperparameters to control the level of sparsity and might include more mechanism in the near future.
>
> 	- **$\epsilon$ in Half-Space projection:** A larger $\epsilon$ refers to a more aggressive Half-Space projection. As Appendix A.2, the $\epsilon$ was set as zero by default. We provided an adaptive mechanism to incrementally increase its value if no group sparsity was identified under default $0$. For CNN pruning experiments, this mechanism was not triggerred thereby $\epsilon$ was fixed as zero during the whole training/pruning procedure. For Bert experiments, the mechanism was triggerred.
>
> 	- **Upper bound of group sparsity:** The upper bound of group sparsity was added to cooperate with the adaptive mechanism of $\epsilon$ if it was trigerred to avoid too aggressive group sparsity exploration. There was no upper bound of the sparsity level for $\epsilon$ as zero. That being said, no sparsity upper bound were used for CNN pruning experiments. In the Bert pruning experiments, as Appendix A.2, we set 10%, 30%, 50%, 70%, 90% for the layers that were involved in the group sparsity penalization, i.e., multi-head attention layer and fully connected layer. Remark here that the results reported in Table 4 also took account the embedding layers which were not penalized by group regularizer. We will add proper description into the revision.
>
> - **Q7: Any analysis on the cost of HSPG and its comparison to other methods?**
>
> 	A7: Yes, we do. Comparing to other SOTA optimization algorithms, Figure 7 in Appendix C shows that HSPG has similar computational cost to proceed as Prox-SG and RDA, and Prox-SVRG is the most expensive due to its periodical full gradient computation. This is because (i) each epoch/iteration of HSPG is polynomial for the computational complexity; and (ii) in practice, the additional operations in HSPG are not expensive and can be completed very efficiently by the processors.

---

### Official Review · Reviewer_DjRg · 2021-07-16

**Rating:** 5
**Confidence:** 3

**Summary:**

This paper proposes a one-shot pruning strategy, in which the pruned subnetwork achieves good performance on several benchmarks without post procedures or iterations. They make use of the ZIG and propose HSPG to enhance the group sparsity exploration. Good performance across several benchmarks supports  the validity of OTO.



**Limitations And Societal Impact:**

See the weakness.

**Main Review:**

Strength

1. The pruning method no longer requires fine-tuning and achieves impressive results on several benchmarks, little performance drop with high compression ratio.

2.  The compression experiments are designed for both vision  and NLP tasks.

Weakness
1.  Missing one important baseline ResRep[1],  which can make this work weaker if included. Note that [1] is a lossless pruning method.

2.  The paper claims that it achieves SOTAs compression benchmark such as R50. However, [1] also prunes without fine-tuning. And the latter slims down a standard ResNet-50 with 76.15% accuracy on ImageNet to a narrower one with only 45% FLOPs.  Morever, ResRep[1] achives 75.3% under similar FLOPS reduction (OTO 74.7%).


Others

1. It's interesting to see the performance of OTO on vision transformers such as ViT.

Reference

[1]Lossless CNN Channel Pruning via Decoupling Remembering and Forgetting, https://arxiv.org/abs/2007.03260 （ICCV21）

Post:

Thanks for the reply. My concern is not well addressed. And I have carefully read the reviews of other reviewers.

I still feel that ignoring some strong baselines  can be misleading and greatly weaken the contribution of this paper. Therefore, I lower my score to below the acceptance threshold.

**Time Spent Reviewing:**

4

---

> ### Author Response · Authors · 2021-08-10
> **Author response for Reviewer DjRg**
>
> Thank you for the constructive comments which help us improve the paper. Please see our following responses.
>
> - **Q1: Missing one important baseline [ResRep2020Arxiv].**
>
> 	A1: We apologize for missing this interesting and important baseline paper, and will include it in the revision.
>
> 	**Reference**
>
> 	[ResRep2020Arxiv] Ding, Xiaohan, et al. Lossless CNN Channel Pruning via Decoupling Remembering and Forgetting. arXiv. 2020.
>
> - **Q2: ResRep achieves better result than OTO on ResNet50 ImageNet.**
>
> 	A2: This is a great comment. We will add this baseline into the revision and adjust the statement regarding the results of OTO on ResNet50 on ImageNet.
>
> - **Q3: It's interesting to see the performance of OTO on vision transformers such as ViT.**
>
> 	A3: This is a good suggestion. We will apply OTO onto vision-transformer and include it into the future revision.

---

> ### Author Response · Authors · 2021-08-23
> **Author response for the post of Reviewer DjRg**
>
> Thank you for the post review. Please see our follow-up response to address the concerns.
>
> Since given an untrained full model, OTO trains it only once from scratch to achieve both high accuracy and slim architecture, it might be not fair enough to compare the result achieved by OTO under fewer epochs to other pruning methods, which in contrast either require pretraining the model, or fine-tuning after pruning, or both. In our experiments, if employing more epochs, the top-1 accuracy can be further improved to be more competitive to these strong baselines.
>
> We provided an updated table to include all the mentioned baselines by the reviewers and will include it into the revision. In the revision, we will adjust our statement regarding the experimental results accordingly.
>
> **Table. ResNet50 for ImageNet.**
>
> |   Method  | FLOPs | # of Params | Top-1 Acc. | Top-5 Acc.|
> | ------------------ | -------- | ------- | ------------------- | ----------- |
> |   Baseline | 100% | 100% | 76.1% | 92.9% |
> |   DDS-26 | 57.0% | 61.2% | 71.8% | 91.9% |
> |   CP | 66.7% | - | 72.3% | 90.8% |
> |   ThiNet-50 | 44.2% | 48.3% | 71.0% | 90.0% |
> |   RBP | 43.5% | 48.0% | 71.1% | 90.0% |
> |   RRBP | 45.4% | - | 73.0% | 91.0% |
> |   SFP | 41.8% | - | 74.6% | 92.1% |
> |   Hinge | 46.6% | - | 74.7% | - |
> |   **OTO** | 34.5% | **35.5%** | 74.7% | 92.1% |
> |   **GBN-50** | 44.9% | 46.6% | 75.2% | 92.4% |
> |   **GBN-60** | 59.5% | 68.2% | 76.2% | 92.8% |
> |   **Group-HS 2e-5** | **32.4%** | - | 75.2% | 92.5% |
> |   **Group-HS 1e-5** | 52.9% | - | **76.4%** | **93.1%** |
> |   **ResRep** | 45.5% | - | 76.2% | 92.9% |
>
> **ResRep**: Lossless CNN Channel Pruning via Decoupling Remembering and Forgetting. ICCV 2021.
>
> **GBN**: Gate Decorator: Global Filter Pruning Method for Accelerating Deep Convolutional Neural Networks. NeurIPS 2020.
>
> **Group-HS**: DeepHoyer: Learning Sparser Neural Network with Differentiable Scale-Invariant Sparsity Measures. ICLR. 2020.

---

### Official Review · Reviewer_7J6K · 2021-07-16

**Rating:** 6
**Confidence:** 3

**Summary:**

This paper proposes a group sparsity optimisation approach for training pruned models. This is done by splitting the optimisation into an initial phase where the non-differentiable regularisation objective is optimised using subgradients, followed by a second stage where channels/units (plus their corresponding bias/BN/residual path) are mapped to exact zero by augmenting the sub-gradients with an indicator function.  For each parameter group, this function compares the direction of subgradient with the direction you'd get as a result of pruning that group, and if the angle between the two is larger than 90 it prunes them away.

**Main Review:**

I think this is an interesting work and the results seem significant but I have a few concerns and suggestions before I can increase my score.

* I feel the way OTO and ZIG are presented as novel ideas is a bit misleading. In the list of contributions OTO is presented as a separate thing in addition to ZIG and HSPG, but it's essentially just the combination of two? The idea of pruning corresponding parameters attached to a channel/weight/unit is also not exactly novel and in some cases (for instance with residuals) is necessary to get shapes right. Similarly, framing this work as "one-shot" pruning is a little bit misleading. The sparsity-inducing objective is present in both initialisation and the projection steps for the entirety of the training so I'm not sure what's "one-shot" about this approach.
* At the beginning of Section 3.1, the authors claim that the _root cause_ that other pruning methods fail is the fact that saliency scores are computed for units/channels in isolation, and their corresponding bias/BN/residual terms are not included. I think this is a bold claim and is not supported by any evidence in the paper. Many pruning objectives are trivial to expand to include the corresponding parameters so if authors want to make this claim this claim I suggest they support it with some ablation experiments with some baselines where this change can be easily tested.
* Section 3.3:
    * I found this section difficult to read and follow. Unless I missed something I felt this section explains the same idea three times with slightly different notations. The beginning of this section includes an "intuition" for doing HSPG but after reading that I was expecting to see the half-space _perpendicular_ to the direction of the gradient that takes you from $x_k$ to $x_{k+1}$. But instead, the half-spaced is based on the origin. Could the authors pehaps make this bit clearer?
    * I was also a bit confused by Fig3a as I was expecting to see projections of $x_{k+1}$ on [x]1 and [x]2 axes?
    * Am I right in thinking that the optimization using subgradient stage doesn't actually stop, but is simply augmented with this additional projection? If yes I think the writing could be improved to include this fact early on.
    * if epsilon is non-zero in all of the experiments why is mentioned at all?
* Section 3.4
    * I'm not sure what's the point of this section. Surely by the end of the optimisation the zero groups have either emerged or not. So I'm not sure what's left to do and what authors mean by "full model" vs "pruned model". Are you talking about physically removing those rows/columns from weight/bias tensors?

Other questions and suggestions
* I'd be curious to see an ablation study to analyse the effect of initialization stage vs the projection stage. What's the split (in terms of the number of epochs) between the two and how did the authors choose these numbers? what is the performance of the neural network _before_ the projection phase?
* I don't find the ResNet-50 experiments with CIFAR-10 convincing. Why have you decided on two baselines (AMC and ANNC) that achieve such low pruning rates (and report no FLOPS numbers)? The fact that these two methods also use quantisation only muddies the waters furthermore. I suggest including some good old pruning baselines with similar pruning ratios.
* how do you control the sparsity target?
* using $x$ to denote parameters is kind of confusing. Perhaps use $\theta$?


**Time Spent Reviewing:**

3

---

> ### Author Response · Authors · 2021-08-10
> **Author response for Reviewer 7J6K**
>
> Thank you for the elaborate reviews and suggestions which are very helpful for us to further improve the paper. Please find the comments duplicated and rephrased for ease of reference. We hope our responses could well address the questions and concerns.
>
> - **Q1: The way OTO and ZIG are presented as novel ideas is a bit misleading, since OTO is essentially the combination of ZIG and HSPG.**
>
> 	A1: This is a great comment. Yes, we proposed a new concept ZIG to partition trainable variables of DNNs and a novel group-sparisty optimization algorithm HSPG, both of which are combined and constructed as the OTO framework. Rather than being two separate ideas, OTO framework is the result of two novelties working together. We will reorganize the list of contributions in Introduction Section to make them more clear.
>
>     In addition, ZIG as Definition 1 is a new general concept for DNNs used to partition the variables. It has an attractive property that if one ZIG equals to zero, then removing its corresponding structure does not affect the model output. This property is leveraged to combine with the proposed HSPG to avoid the accuracy regression during constructing pruned model then remove fine-tuning.
>
>     To the best of our knowledge, no related work has formalized a similar concept and partitions the DNNs as the proposed ZIG. In Section 3.1 and Appendix A.1, we provided a few concrete examples for illustrating the ZIG concept from Conv-BN to Residual Blocks and Multi-head attention. Furthermore, as a general concept, ZIG is not limited to the illustrating examples shown in this paper, it has varying possibilities to different architectures.
>
> - **Q2: Not sure what the one-shot means about this approach.**
>
> 	A2: One-Shot is a description from a practical point of view when OTO is being used. It refers to that given a full untrained model, our framework only trains once to get a slimmer model of high accuracy. Although the HSPG is a two-stage optimization algorithm, since it has been packaged as an optimization algorithm, the users do not have to pay too much attention to when the stage switches, see more in Q9/A9. We will add more descriptive languages to provide more clarity.
>
>
> - **Q3: Many pruning methods, e.g., the ones use saliency scores, are trivial to expand to include ZIG. so if authors want to make this claim I suggest they support it with some ablation experiments with some baselines where this change can be easily tested.**
>
>    A3: This is a good comment. We have general perspectives that (i) ZIG might not be trivially integrated with the majority of pruning methods; and (ii) some structured-sparsity pruning methods might be able to naturally leverage ZIG, while their utility of ZIG would be not as adequate as OTO to combine ZIG with the proposed HSPG.
>
>     In details, as mentioned in Q1/A1, ZIG has an attractive property that if one zero-invariant group equals to zero, then removing its corresponding structure does not affect the model output. That being said, this key property might be only leveraged by the pruning methods that identify redundant structures via group-sparsity identification. Therefore, based on this criteria, the related structured-sparsity pruning methods might be able to leverage this crucial property of ZIG, while others might not, e.g., the ones using saliency scores, since they do not generate zero groups unless equip with additional mechanisms,  .
>
>     On the other hand, to adequately make use of ZIG for pruning, identifying sufficiently many ZIGs as zero is necessity. But the existing structured-sparsity pruning methods do not work well enough because of the limitation of the standard proximal method and ADMM on group sparsity exploration, and typically requires a heuristic post-processing to generate exact group sparsity. To overcome the limitations, we proposed the HSPG to do a better group sparsity exploration with comparable objective convergence, and demonstrated its superiority comprehensively from intuitive explanation, theory, to numerical experiments in both the main body with Hinge and ProxSSI and Appendix C.
>
>     Based on the above, the combination of ZIG and HSPG as OTO might be the optimal at this point. Other pruning methods might not utilize ZIG to the largest extent. To verify, in Appendix C.2, we conducted an ablation study to apply standard SGD and Prox-SG on ZIG, where as shown in Figure 8(b), both FLOPs and number of parameters reduction via SGD and Prox-SG are worse than HSPG. We will add more ablation studies in the revision.
>
> - **Q4: Section 3.3: I was expecting to see the half-space perpendicular to the direction of the gradient. But instead, the half-spaced is based on the origin. Could the authors perhaps make this bit clearer?**
>
> 	A4: This is a good question. In general, the half-space defined as $\mathcal{S}(x_k)$ in the main body describes a region that if a group $g$ of variables in the trial iterate $[\tilde{x}_{k+1}]_g:=[x_k-\alpha_k \nabla \psi(x_k)]_g$ conducted by sub-gradient descent falls outside $\mathcal{S}(x_k)$, then projecting this group of variables to zero, i.e., $-[x_k]_g$, is a descent direction to $\psi(x_k)$. Otherwise if the trial iterate falls into $\mathcal{S}(x_k)$, the $-[x_k]_g$ is not necessarily to be a descent direction; then no projection should be proceeded.
>
>     Based on the construction of $\mathcal{S}(x_k)$, we established a novel Half-Space projection to effectively project groups of variables to zero without deteriorating the progress to the optimality. Furthermore,  sufficient decrease property and objective convergence can be achieved by HSPG as drawn in Lemma 1 and Theorem 1.
>
>     We will add more description to provide more clarity in the revision.
>
> - **Q5: I was expecting to see projections of  on $[x]_1$ and $[x]_2$ axes in Fig3a**
>
> 	A5: In the caption of Figure 3, we provided the group partition $\mathcal{G}=${{1,2}}, which indicates that $[x]_1, [x]_2$ belongs to the same group. Therefore, the Half-Space projection aims at projecting the entire group {1,2} to zero, not the individual element onto either $[x]_1$ or $[x]_2$ axis. As a result, if projection on this group happens, the iterate is projected onto the origin point $\mathcal{O}$.
>
> - **Q6: If subgradient descent doesn't actually stop, the writing could be improved to include this fact early on.**
>
> 	A6: Yes, a great suggestion. We will highlight it early on accordingly.
>
> - **Q7: Why not mentioned the setting of $\epsilon$ if it is not zero?**
>
> 	A7: As described in Appendix A.2 Implementation Details, the $\epsilon$ is set as zero by default. If no group sparsity explored under $\epsilon$ as zero, an adaptive strategy is implemented to incrementally increase its value. For all CNN pruning experiments in the main body, the $\epsilon$ was fixed as zero during the whole procedure without triggering the adaptive mechanism. For the Bert pruning experiments, the $\epsilon$ started from zero then incrementally increased as no group sparsity was identified under $\epsilon=0$. We will incorporate the above into the main body of the revision.
>
> - **Q8: Section 3.4 is redundant. Does pruned model refer to removing the rows/columns from tensors?**
>
> 	A8: This is a great suggestion. Yes, they refer to physically removing the structures corresponding to zero groups identified by HSPG. We added Section 3.4 for the description completeness of the main Algorithm 1 and agree to remove it to give more space to incorporate the other discussions.
>
> Please see more Q/A in the second official reply.

---

> > ### Author Response · Authors · 2021-08-10
> > **Author response for Reviewer 7J6K Part 2**
> >
> > - **Q9: I'd be curious to see an ablation study to analyse the effect of initialization stage vs the projection stage. What's the split (in terms of the number of epochs) between the two and how did the authors choose these numbers? what is the performance of the neural network before the projection phase?**
> >
> > 	A9: This is a great question. We will add an ablation study regarding the impact the switch between initialization stage and projection stage. In theory, as shown in Theorem 1, the projection stage should start when the iterate falls nearby a group sparse local minimizer. In practice, we relaxed it to start projection stage once the iterate falling into some stationary status regarding the validation accuracy. As described in Appendix A.2, we proceeded a similar test as [SALSA2020Arxiv] but on the validation accuracy and found that the validation accuracy fell into stationarity on the late epochs of each period. (Recall that we periodically decayed the learning rate from $10^{-1}$ to $10^{-4}$ every $T$ epochs.) Therefore, in our pruning experiments, we switched to projection stage from initialization stage right after the first $T$ epochs.
> >
> > 	As shown in the below table, in general, the DNN performances before projection stage had relatively good validation accuracy, but not as high as that after projection stage because they were only trained by SGD $T$ early epochs. And the initialization stage did not identify any zero group out, with the group sparsity ratio as zero, which is defined as
> > $$
> > \text{group sparsity ratio}= \frac{\text{the number of zero groups}}{\text{the total number of groups}},
> > $$
> >     while the group sparsity stage explored group sparsity much more effectively. These results implies the effectiveness of Half-Space step on group sparsity identification with comparable objective convergence.
> >
> >   **Table. Validation accuracy and group sparsity ratio at the end of each stage**
> >
> >   |   Experiment  | Initialization Stage | Initialization Stage | Group Sparsity Stage | Group Sparsity Stage |
> > | --- | -------- | ------- | ------------------- | ----------- |
> > |  | Validation ACC1 | Group Sparsity Ratio | Validation ACC1 | Group Sparsity Ratio |
> > | VGG16 CIFAR10 | 75.2% | 0% | 91.0% | 80.4% |
> > | VGG16-BN CIFAR10 | 76.9% | 0% | 93.3% | 75.8% |
> > | ResNet50 CIFAR10 | 86.1% | 0% | 94.4% | 73.9% |
> > | ResNet50 ImageNet | 63.2% | 0% | 74.7% | 65.8% |
> >
> >
> >     In our experiments, starting the projection stage right after $2T$ or $3T$ epochs did not result in significantly different pruning results because of the existence of the numerous local minimizers to the nonconvex DNN training problems. We will incorporate the above into the revision.
> >
> > 	**Reference:**
> >
> > 	[SALSA2020Arxiv] Pengchuan Zhang, Hunter Lang, Qiang Liu, and Lin Xiao. Statistical adaptive stochastic gradient methods. arXiv preprint arXiv:2002.10597 (2020).
> >
> > - **Q10: For ResNet50 on Cifar10, why  choose AMC and ANNC as baselines?**
> >
> > 	A10: This is a great comment. In general, we selected [ANNC2020CVPR] and [AMC2018ECCV] to be compared because of two reasons.
> >
> > 	- [ANNC2020CVPR] and [AMC2018ECCV] are two SOTA automatic DNN compression frameworks. Since OTO is also able to automatically compress DNNs, we compared them on their shared experiment and included all the results shown in [Table 3, ANNC2020CVPR], but excluded the numbers conducted with quantization.
> >
> > 	- [ANNC2020CVPR] also studies pruning from the view of sparsity and uses ADMM as the optimization algorithm. ADMM is an alternative optimizer to the standard proximal method which has been compared in both main body and Appendix C.
> >
> > 	We will incorporate the above discussion into the revision. Additionally, we have found more baseline literatures which conduct the same experiment but achieve similar pruning performance (though a little worse) compared to OTO, and will add these baselines into the revision.
> >
> >     **Reference:**
> >
> >     [AMC2018ECCV] AMC: AutoML for model compression and acceleration on mobile devices. ECCV. 2018.
> >
> >     [ANNC2020CVPR] Automatic neural network compression by sparsity-quantization joint learning: A constrained optimization-based approach. CVPR. 2020.
> >
> >     [PruneTrain2019AMCSC] PruneTrain: fast neural network training by dynamic sparse model reconfiguration. AMCSC. 2019.
> >
> >     [N2NSkip2020BMVC] N2NSkip: Learning Highly Sparse Networks using Neuron-to-Neuron Skip Connections. BMVC. 2020.
> >
> > - **Q11: How do you control the sparsity target?**
> >
> > 	A11: Currently, we provided two mechanisms to control the group sparsity target (i) $\epsilon$ in Half-Space projection, and (ii) upper bound for group sparsity. The $\epsilon$ has been discussed in Q7/A7. The upper bound of group sparsity was added to cooperate with the adaptive mechanism of $\epsilon$ if trigerred to avoid too aggressive group sparsity exploration. There was no upper bound of the sparsity level for $\epsilon$ as zero. That being said, no sparsity upper bound were used for CNN pruning experiments. In the Bert pruning experiments, we set the upper bounds as 10\%, 30\%, 50\%, 70\% and 90\% for the layers that were involved during pruning, i.e., multi-head attention layer and fully connected layer. Remark here that the results reported in Table 4 took account the embedding layers which were not penalized by group regularizer. We will add more clarification into the revision.
> >
> > - **Q12: Using $x$ to denote parameters is kind of confusing. Perhaps use $\theta$?**
> >
> > 	A12: We will switch $x$ to $\theta$ in the revision.

---

### Official Review · Reviewer_N3gq · 2021-07-17

**Rating:** 6
**Confidence:** 4

**Summary:**

This paper first proposes the zero-invariant group for neural network. Zero-invariant group is the disjoint group of trainable parameters which results in corresponding output to be zero. Then, this paper formulates a optimization problem with group sparsity regularizer on the zero-invariant groups. Then, authors alternate SGD and Half-space projection to optimize the problem. This optimization method is called Half-Space Stochastic Projected Gradient (HSPG) which benefits from the larger projection region. This method does not require fine-tuning  and pruned model is directly used on the inference. On various dataset (Cifar-10 and ImageNet), this paper achieves state-of-the-art performance.


**Ethics Review Area:**

["I don’t know"]

**Limitations And Societal Impact:**

Yes

**Main Review:**

Weakness
1.  I believe the novelty of the group sparsity regularizer on the zero-invariant group is not enough. The idea of a group sparsity regularizer has received attention for a long time. Please refer to [1], [2]. The only extension of this paper is to leverage the group sparsity regularizer on the zero-invariant group which includes the parameter in the batch normalization. Therefore, authors need to compare these related works and this paper, then show the superiority of the proposed method with the experiments.

2. The paper insists on the superiority of the HSPG method. However, there is no intuitive explanation for this. The only explanation is the larger projection space. But, readers might not agree that this point directly contributes to the performance of the pruning method. The authors need to provide the intuitive explanation that most of the readers and reviewers agree with. I suggest the authors reduce the ZIG section and add an ablation study and explanation from the appendix.

3. Table1 is quite weird. It seems that the most of FLOPs reduction comes from the first channel reduction (64->3). There should be sufficient analysis and explanation for this.

4. The baselines on the experiment results are outdated and there are not enough baselines in Tables 2 and 3. Authors need to compare the proposed method with recent pruning papers. For example, GBN [3] achieves 75.18 with 50% FLOPS.

Therefore, I conclude that this paper is not ready for submission. The novelty of the proposed method is hard to find and the superiority of the proposed method is hard to believe since there is no intuitive explanation and not enough experiment results.

[1] Learning Structured Sparsity in Deep Neural Networks, NeurIPS 16
[2] DeepHoyer: Learning Sparser Neural Network with Differentiable Scale-Invariant Sparsity Measures, ICLR 20
[3] Gate Decorator: Global Filter Pruning Method for Accelerating Deep Convolutional Neural Networks, NeurIPS 20.


Post :

I have read all the feedback. Some issues are resolved. However, still other critical issues are not fully resolved. Also, I could not agree with the novelty from ZIG and do not fully agree with importance of ZIG. Furthermore, I still could not fully understand the correlation between the large projection area and group sparsity. Furthermore, I could not see the updated experiment results of important baselines in the main table. Therefore, I keep my score.


Post Post:

I have read authors' responses. Most of critical issues which I pointed are resolved. Please update the experiment results (after fine-tuning) + other baselines. I wish to see them in the final version. Some parts are still hard to understand (HSPG) and I'm not sure that claims on ZIG are impressive. However, clear experiment results support the novelty and importance of the proposed method. There is no reason to reject this paper. Therefore, I raise my score.


**Time Spent Reviewing:**

6 hours

---

> ### Author Response · Authors · 2021-08-10
> **Author response for Reviewer N3gq**
>
> Thank you for the reviews and suggestions. Below, please find the comments duplicated and rephrased for the ease of reference. We hope the responses can improve the assessment of our paper.
>
> - **Q1: The only novelty/extension of this paper is to leverage the group sparsity regularizer on the zero-invariant group which includes the parameters in the batch normalization.**
>
>   A1: We disagree that our novelty is restricted to proposing zero-invariant group which includes parameters of batch normalization and then applying group sparsity regularizer. In fact, as L113-L115, it serves as a special example for illustrating the proposed generalized concept ZIG as Definiton 1, and accounts to a small portion to our main contributions.
>
>   Particularly, as summarized in Introduction, the novelty contains the following two aspects:
>
>   - **Zero-Invariant Group (ZIG)**
>
>   	- **Unique and novel:** We studied  how  to  properly  partition  the  variables  of  DNNs  into  a  set  of  groups to  avoid  multiple  training  steps  during  the  whole  train/pruning  procedure,  which is rarely  explored.  To the best of our knowledge, none related work partitions the variables of DNNs as ZIG. In  fact,  the  existing structured-sparsity pruning methods focus on a shared group setting, i.e.,the row of each filter matrices, which is significantly different from the proposed ZIG partition.
>
>   	- **Generality:** The ZIG is proposed as a general concept and applicable to various DNN architectures. The groups including parameters of batch normalization was used only as a straightforward example for illustrating the ZIG concept. Beyond it, we further provided more illustrating examples, such as residual block and multi-head attention in Section 3.1 and Appendix A.1. Furthermore, remark here that as a general concept for DNNs, the variety of ZIG is also not limited to the illustrated examples shown in the paper.
>
>
>   - **Half-Space Stochatic Projected Gradient (HSPG) Method**
>
> 	  To promote the group sparsity effectively, we proposed a novel HSPG optimization algorithm, which is
>
> 	  - **Unique and novel:** To the best of our knowledge, there is no similar method as ours. As the main-stream works on (group) sparsity have focused on using proximal method, our method solve the group sparsity problem in a brand new way.
>
> 	  - **Superiority in theory:** The proposed HSPG is superior to the standard proximal gradient methods on group sparsity exploration and maintain comparable convergence property. To demonstrate it,  we provided
>
> 	       - **Intuitive illustration:** On L221-233, we provided an intuitive illustration that HSPG enjoys a lager projection region associate with mathematical proof in Appendix B.3. Hence HSPG can promote group sparsity more effectively than others.
>
> 	  	- **Theoretical guarantee:** On L233-235, we explicitly pointed out that we provided Theorem 2 to reveal that HSPG has a better group sparsity identification property comparing to the standard proximal method since it requires a better $\ell_2$-ball radius, see Appendix B.5.
>
> 	  - **Superiority in numerical experiments**: We provided comprehensive numerical experiments to demonstrate the superiority of HSPG on group sparsity exploration. See more in Q2/A2.
>
> - **Q2: No evidence to demonstrate the superiority of HSPG.**
>
> 	A2: We did provide comprehensive evidence from various aspects to demonstrate the superiority of HSPG to the standard commonly used proximal method in the structured-sparsity pruning methods.
>
> 	- **Intuitive Explanation:** On L221-L232 and Figure 3b, we provided an intuitive explanation to reveal the superiority that HSPG enjoys a larger projection region compared to the standard proximal gradient methods. To support it in more depth, as pointed out in L230-L231, we also provided a rigorous mathematical derivation of this projection region in Appendix B, see the Proposition 1 in Appendix B.3 along with the detailed explanations.
>
> 	- **Theoretical demonstration with explanation:** On L233-L235, from a high level perspective, we summarized the superiority of HSPG in the view of theory, and explicitly pointed out that HSPG has a better group sparsity recovery property than the proximal methods as drawn in Theorem 2. We also provided its detailed proofs in Appendix B.5, associated with a Remark section on L986-L994 (in Appendix) to intuitively explain the superiority from the view of group sparsity recovery.
>
> 	- **Numerical verification with explanation:** We explicitly presented sufficient numerical verifications as evidence to demonstrate the superiority of HSPG from three aspects:
>
> 		- **Comparison with SOTA structured-sparsity pruning methods:** In Section 4, we compared OTO with various pruning methods on the pruning experiments from computer vision to NLP. Among these competitors, we paid special attention the recent SOTA structured-sparsity pruning methods [Hinge2020CVPR] and [ProxSSI2021Arxiv]. As L296-L302 and L343-L349, we carefully explained that OTO outperforms these methods significantly on the compression performance because of the superiority of HSPG on the group sparsity exploration to the proximal methods used in [Hinge2020CVPR] and [ProxSSI2021Arxiv].
>
> 		- **Comparison with SOTA proximal methods in the view of optimization:** As we explicitly pointed out on L236, we provided extensive numerical experiments along with detailed explanations in Appendix C to demonstrate the superiority of HSPG. In these extensive numerical experiments, we carefully compared HSPG with several SOTA proximal methods in the view of optimization on various settings from convex to nonconvex, from ZIG to non-ZIG.
>
> 			The test models and datasets in these extensive experiments cover the benchmark problems in classical machine learning as well additional DNN structures. Many of them did not appear in the main body. In particular,
>
> 			- **Covered Models in Appendix C:** (i) Linear Regression, (ii) Logistic Regression, (iii) MobileNetV1, (iv) ResNet18 and (v) VGG16.
>
> 			- **Covered Datasets in Appendix C:** (i) Eight large-scale datasets from [LIBSVM2011ACM], (ii) Synthetic data, (iii) Fashion-MNIST and (iv) CIFAR10.
>
> 		- **Ablation Study:** As shown in Figure 8b and the description on L1125-L1128 in Appendix C.2, we provided an ablation study to show that if replaced HSPG with other standard optimization algorithms and applied on the ZIG partition,  both FLOPs and number of parameters reduction are not competitive to the ones achieved by HSPG.
>
> 	**Reference:**
>
> 	[Hinge2020CVPR] Group sparsity: The hinge between filter pruning and decomposition for network compression. CVPR 2020.
>
> 	[ProxSSI2021Arxiv] Tristan Deleu, Yoshua Bengio. Structured sparsity inducing adaptive optimizers for deep
>  learning. Arxiv. 2021.
>
> 	[LIBSVM2011ACM] Chih-Chung Chang and Chih-Jen Lin. Libsvm: A library for support vector machines. ACM transactions on intelligent systems and technology. 2011.
>
> - **Q3: Authors need to refer to related structured sparsity pruning methods in the paper such as [SSL2016NIPS] and [DeepHoyer2020ICLR]**
>
> 	A3: Thanks for pointing out two related references. But in L87-L92 Section 2 of our main paper, we actually had referred and discussed
>  [SSL2016NeurIPS] ,i.e., the [72] in the main body, along with a fair amount of discussion to other structured sparsity pruning methods. In Section 4, we had compared with the recent SOTA  structured sparsity pruning methods such as [Hinge2020CVPR] and [ProxSSI2021Arxiv]. We will refer, discuss, and compare [DeepHoyer2020ICLR] in the revision.
>
> 	**Reference:**
>
>     [SSL2016NeurIPS] Learning Structured Sparsity in Deep Neural Networks. NeurIPS. 2016.
>
>     [DeepHoyer2020ICLR] DeepHoyer: Learning Sparser Neural Network with Differentiable Scale-Invariant Sparsity Measures. ICLR. 2020.
>
>
> - **Q4: In Table 1, it seems that the most of FLOPs reduction comes from the first channel reduction. There should be sufficient analysis and explanation for this.**
>
> 	A4: For VGG16, the majority of the FLOPs reduction via OTO comes from a few middle ConvLayers (over 10% to the overall FLOPs reductions) instead of the first ConvLayer (0.45% to the overall FLOPs reduction), see the Table below. In general, the distribution of FLOPs reduction per Layer of OTO is similar to other pruning baselines.
>
> 	**Table. FLOPs Reduction Breakdown for the ConvLayers of VGG16 on CIFAR10**
>
> |   ConvLayer Index  | Original Output Channel | Pruned Output Channel | FLOPs Reduction Quantity (Million) | FLOPs Reduction Percentage (%)|
> | ------------------ | -------- | ------- | ------------------- | ----------- |
> |   1 | 64 | 21 | 1.19M | 0.45% |
> |   2 | 64 | 45| 29.04M | 11.07% |
> |   3 | 128 | 82 | 10.47M | 3.99% |
> |   4 | 128 | 110 | 17.22M | 6.57% |
> |   5 | 256 | 109 | 11.97M | 4.56% |
> |   6 | 256 | 68 | 33.48M | 12.77% |
> |   7 | 256 | 37 | 36.30M | 13.84% |
> |   8 | 512 | 13 | 18.81M | 7.17% |
> |   9 | 512 | 9 | 37.73M | 14.38% |
> |   10 | 512 | 7 | 37.74M | 14.39% |
> |   11 | 512 | 3 | 9.44M | 3.60% |
> |   12 | 512 | 5 | 9.44M | 3.60% |
> |   13 | 512 | 8 | 9.44M | 3.60% |
>
>    Remark here that 3 is the number of channel for the input image, which will be excluded from Table 1 in the revision since it is not affected during pruning. Therefore, 21 and 22 are the actual first output channel after pruning VGG16 and VGG16-BN respectively.
>
>    Thanks for the good suggestion. We will provide layer-wise FLOP reduction breakdown in the revision.
>
> - **Q5: The baselines on the experiment results are outdated and there are not enough baselines in Tables 2 and 3.**
>
> 	A5: This is a good suggestion, we will incorporate more recent baselines into the revision.

---

> ### Author Response · Authors · 2021-08-23
> **Author response for the post of Reviewer N3gq**
>
> Thank you for the post review. Please see our follow-up responses. We hope the responses can further address your concerns.
>
> - **Q6: I could not agree with the novelty from ZIG and do not fully agree with importance of ZIG.**
>
>   A6: The novelty and importance of ZIG are explained in details as follows:
>
>    - **Novelty**: Zero-Invariant Group (ZIG) is a dedicatedly designed new concept used for partitioning the variables of DNNs. It has a variety of constructions, including but not limited to Conv-BN, Residual Block and Multi-Head Attention. As L111-115 on the main body, ZIG has in a very attractive property.
>
> 	  **Property**: If one zero-invariant group equals to zero, then removing its corresponding structure does not affect the model output.
>
> 	 To the best of our knowledge, no related work has studied similar group partition associated with this property. We kindly ask the reviewer to point out the missing literature which studied the group partition in the way as ZIG.
>
> 	- **Importance**: The above property is one of the fundementals to the proposed OTO to achieve that given an untrained full model, **only train it once** to achieve both high accuracy and slimmer architecture.
>
> 	  In sharp contrast, the existing structured-sparsity pruning methods, including the mentioned [SSL2016NeurIPS, DeepHoyer2020ICLR, GD2020NeurlPS], have to train/fine-tuning the untrained full model multiple times to achieve a slimmer model of high accuracy. They all require multiple-time trainings because their partitioned groups of DNNs are not ZIGs, thus pruning the full model results in significant accuracy regression and has to equip with a followup fine-tuning step to regain the accuracy. Such procedure might be complicated, time-consuming and not user-friendly.
>
>          By using ZIG associated with HSPG, the drawbacks of the existing structured-sparsity pruning methods can be resolved. Therefore, the proposed ZIG is important.
>
>    **Reference**:
>
>    [SSL2016NeurIPS] Learning Structured Sparsity in Deep Neural Networks. NeurIPS 2016.
>
>    [DeepHoyer2020ICLR] DeepHoyer: Learning Sparser Neural Network with Differentiable Scale-Invariant Sparsity Measures. ICLR 2020.
>
>    [GD2020NeurlPS] Gate Decorator: Global Filter Pruning Method for Accelerating Deep Convolutional Neural Networks. NeurIPS 2020.
>
> - **Q7: I still could not fully understand the correlation between the large projection area and group sparsity.**
>
> 	A7: We here further explain the correlation between projection region and group sparsity.
>
> 	In general, as shown in Table 10 and 11 of Appendix C, the proximal method performs not well on the group sparsity exploration in the stochastic settings and even returns fully dense solutions unless equipped with additional post-processing mechanism. As L221-L232 and Figure 3b in the main body, we intuitively highlighted that the reason is largely due to its limited projection region.
>
> 	In details, as L223-L225, the projection region refers to the area that map a group of variables on the trivial iterate to zero, which is the fundamental mechanism of proximal method to generate zero groups and yield group sparsity. In particular, at $k$th iteration, proximal gradient method computes a trial iterate $x':=x_k-\alpha_k \nabla f(x_k)$, where $\alpha_k$ is the step size (learning rate), and $f$ is the loss function. Suppose the regularizer is the popular mixed $\ell_1/\ell_2$ norm, the next iterate $x_{k + 1}$ is then computed as follows that for each group $g\in\mathcal{G}$,
>
>   $[x_{k+1}]_g=\text{max}${$0,1-\frac{\alpha_k\lambda}{||[x']_g||_2}$ }$[x']_g$.
>
> 	Based on the above equation, it is clear to see that one group $g$ is projected to zero if and only if the trial iterate $[x']_g$ falls into an $\ell_2$-ball with radius as $\alpha_k\lambda$. Therefore, the area of this $\ell_2$-ball represents the likelihood that one group $g$ can be mapped to zero. A larger projection region typically means more easily to yield zero groups, hence may achieve higher group sparsity.
>
> 	Furthermore, as L225-L229, in nonconvex DNNs applications, the step size (learning rate) $\alpha_k$ is typically selected as small as $10^{-1}$ to $10^{-4}$ for convergence. Together with the common setting $\lambda\ll 1$, the projection region of proximal method, i.e., the $\ell_2$-ball with radius $\alpha_k\lambda$, is too small to project groups of variables onto zeros effectively. In contrast, as L229-L232, the projection region of HSPG does not vanish as the proximal method thereby can produce group sparsity much more effectively.
>
>
> - **Q8: I could not see the updated experiment results of important baselines in the main table.**
>
>   A8: We provided the revised Table including all the mentioned important baselines from the reviewers. We will include it into the revision and adjust our statements regarding the experimental results accordingly.
>
>   **Table. ResNet50 for ImageNet.**
>
>   |   Method  | FLOPs | # of Params | Top-1 Acc. | Top-5 Acc.|
>   | ------------------ | -------- | ------- | ------------------- | ----------- |
>   |   Baseline | 100% | 100% | 76.1% | 92.9% |
>   |   DDS-26 | 57.0% | 61.2% | 71.8% | 91.9% |
>   |   CP | 66.7% | - | 72.3% | 90.8% |
>   |   ThiNet-50 | 44.2% | 48.3% | 71.0% | 90.0% |
>   |   RBP | 43.5% | 48.0% | 71.1% | 90.0% |
>   |   RRBP | 45.4% | - | 73.0% | 91.0% |
>   |   SFP | 41.8% | - | 74.6% | 92.1% |
>   |   Hinge | 46.6% | - | 74.7% | - |
>   |   **OTO** | 34.5% | **35.5%** | 74.7% | 92.1% |
>   |   **GBN-50** | 44.9% | 46.6% | 75.2% | 92.4% |
>   |   **GBN-60** | 59.5% | 68.2% | 76.2% | 92.8% |
>   |   **Group-HS 2e-5** | **32.4%** | - | 75.2% | 92.5% |
>   |   **Group-HS 1e-5** | 52.9% | - | **76.4%** | **93.1%** |
>   |   **ResRep** | 45.5% | - | 76.2% | 92.9% |
>
>
>   **GBN**: Gate Decorator: Global Filter Pruning Method for Accelerating Deep Convolutional Neural Networks. NeurIPS 2020.
>
>   **Group-HS**: DeepHoyer: Learning Sparser Neural Network with Differentiable Scale-Invariant Sparsity Measures. ICLR. 2020.
>
>    **ResRep**: Lossless CNN Channel Pruning via Decoupling Remembering and Forgetting. ICCV 2021.
>
>   Given an untrained full model, OTO trains it only once from scratch to achieve both high accuracy and slim architecture. The results achieved by OTO are obtained under fewer epochs to other pruning methods, which in contrast either require pretraining the model, or fine-tuning after pruning, or both. In fact, if employing more epochs onto OTO, the top-1 accuracy can be further improved to be more competitive to these strong baselines.

---

> > ### Comment · Reviewer_N3gq · 2021-08-26
> > **Additional question on ResNet50 experiment in ImageNet**
> >
> > Thank you for the author responses. In the table, OTO did not outperform other baselines. Authors insist that employing more epochs onto OTO leads to the further improvement on Top-1 accuracy. Then, would you share me the improved accuracy?
> >
> > Also, authors need to compare with [1] since [1] does not require finetuning. Top-1 accuracy is 75.26 when FLOPs is reduced to  45.7%, please refer to [1]. For the fair comparison, authors need to compare with these baselines under the same flops constraint.
> >
> > For the issue of importance of ZIG and HSPG, there are some dissensions between me and other reviewers. Authors should show the importance of ZIG and HSPG with experiment results on network pruning. For example,
> > 1) performance of the pruned network with sparsity regularizer on ZIG and with sparisty regualrizer on batchnorm,
> > 2) OTO and OTO without HSPG (other projection technique) on ResNet 50, ImageNet or other dataset.
> >
> > [1] Operation-Aware Soft Channel Pruning using Differentiable Masks, ICML2020

---

> > > ### Author Response · Authors · 2021-08-27
> > > **Author response for the additional questions of Reviewer N3gq**
> > >
> > > Thank you for the response. Please find the below duplicated and rephrased comments along with our responses for addressing the concerns.
> > >
> > >
> > > - **Q9: Authors insist that employing more epochs onto OTO leads to the further improvement on Top-1 accuracy. Then, would you share me the improved accuracy?**
> > >
> > >   A9: We do have some preliminary experimental results regarding keep training the last checkpoint achieved by OTO by more epochs under additional training tricks, such as smaller learning rate and fine-tuning the momentum etc. In particular, the Top 1 accuracy can be further improved in general. So far, the best top-1 accuracy achieved after making additional training efforts is 75.1% which becomes more competitive to these strong baselines. The corresponding checkpoint has been uploaded onto https://tinyurl.com/nips20212271, i.e., the highly group sparse full model resnet50\_imagenet\_acc1\_75.1.pt associated with the pruned slimmer model resnet50\_imagenet\_pruned\_acc1\_75.1.pt with the same accuracy.
> > >
> > > 	In the revision, we will include these extensively post-training experiments after the results are stabilized.
> > >
> > > - **Q10: [SCP2020ICML] also does not require fine-tuning. It considers batchnorm. Authors need to compare with it**
> > >
> > > 	A10: Thanks for pointing out this related literature. [SCP2020ICML] avoids fine-tuning by involving the batchnorm (BN) into the pruning consideration. This idea is similar to our first illustrating example of ZIG in Section 3.1, i.e., the Conv-BN.
> > >
> > > 	But we claim that ZIG is a more general concept and serves as a superset of the ideas shown in [SCP2020ICML] due to the following reasons:
> > >
> > > 	- [SCP2020ICML] considers only batchnorm, and does not formalized a general concept as our proposed ZIG to be extended for general DNNs.
> > > 	- In contrast, ZIG is a well-defined and general concept for various DNNs. It supports not only the batchnorm but also the general DNN architectures.
> > >
> > >        For examples, besides Conv-BN, we provided the illustrating examples such as residual blocks and multi-head attention, both of which the ZIG are dedicately and non-trivially constructed comparing with batchnorm because of the more sophasiticated architecture, see Section 3.1 and Appendix A.1 for more details.
> > >
> > >       Furthermore, the ZIG is not limited to the shown illustrating examples in the main body. It has more varieties to broader DNN architectures.
> > >
> > > 	To the best of our knowledge, no related work systematically formulizes, studies, and utilizes the ZIG as we did in OTO.
> > >
> > > 	To compare with [SCP2020ICML], we present the below tables on two shared experiments, i.e., VGG16-BN on CIFAR10 and ResNet50 on ImageNet. We can see that OTO outperforms SCP on most of the evaluaton metrics except the Top-1 ACC on VGG16 for CIFAR10.
> > >
> > >   **Reference**:
> > >
> > > 	[SCP2020ICML] Operation-Aware Soft Channel Pruning using Differentiable Masks, ICML2020
> > >
> > > 	We will include the above discussion into the revision.
> > >
> > > 	**Table. VGG16 for CIFAR10.**
> > >
> > > 	|   Method  | FLOPs | # of Params | Top-1 Acc. |
> > > 	| ---------- | -------- | ------- | ----- |
> > > 	| **OTO** | **26.8%** | **5.5%** | 93.3% |
> > > 	| SCP | 33.8% | 7.0% | **93.8%** |
> > >
> > >
> > > 	**Table. ResNet50 for ImageNet.**
> > >
> > > 	|   Method  | FLOPs | # of Params | Top-1 Acc. | Top-5 Acc.|
> > > 	| ------ | -------- | ------- | ----- | ------ |
> > > 	|   **OTO** | **34.5%** | **35.5%** | **74.7%** | **92.1%** |
> > > 	| SCP | 45.7% | - | 74.2% | 92.0% |
> > >
> > >
> > >
> > >
> > > - **Q11: Authors should show performance of the pruned network with sparsity regularizer on ZIG and with sparisty regualrizer on batchnorm.**
> > >
> > >   A11: If we understand correctly, you suggest to employ OTO on the pruned model again or maybe even more times to see if further compression could be achieved.
> > >
> > >   We will study the phenomenon of iteratively employing OTO on one model and include it into the Appendix of revision. In the main body, we would like to focus more on discussing an only-train-once DNN training and pruning framework.
> > >
> > >    Furthermore, batchnorm is typically included into the ZIG construction so that there is no need to penalize them via sparsity regularizer separately.
> > >
> > >
> > > - **Q12: Authors should show OTO and OTO without HSPG (other projection technique) on ResNet 50, ImageNet or other dataset.**
> > >
> > >   A12: This is a good suggestion. Similarly to the ablation study as VGG16 on CIFAR10 shown in Figure 8(b) of Appendix C.2, we are conducting an extensive experiment on ResNet50 for CIFAR10 under ZIG but training by SGD and ProxSG rather than by HSPG. We will update the results in the revision.

---

### Decision · Program_Chairs · 2021-09-27

**Decision:**

Accept (Poster)

**Comment:**

The authors propose a fine-tuning free structured pruning method (OTO). The idea is to first partition the parameters into zero-invariant groups, pruning the zero groups, and solving for a structured sparsity optimization problem with projections. The experiment results on CIFAR10, ImageNet, and SQuAD show the competitive performance on FLOPs and number of parameters reduction.

The paper initially received a split review. I would like to thank the time and effort the authors and reviewers spent engaging in the active discussion during the rebuttal phase.

I strongly recommend the authors to revise the submission and include the ablation study and the baseline comparisons the reviewers requested.

Specifically, please include

1. The ablation study of initialization stage vs projection stage

2. Baseline comparison with related methods [1,2,3,4,5].

[1] Learning Structured Sparsity in Deep Neural Networks, NeurIPS 16
[2] DeepHoyer: Learning Sparser Neural Network with Differentiable Scale-Invariant Sparsity Measures, ICLR 20
[3] Gate Decorator: Global Filter Pruning Method for Accelerating Deep Convolutional Neural Networks, NeurIPS 20.
[4] Operation-Aware Soft Channel Pruning using Differentiable Masks, ICML2020
[5] Lossless CNN Channel Pruning via Decoupling Remembering and Forgetting. ICCV 2021.